# Ordered Momentum for Asynchronous SGD

**Chang-Wei Shi**     **Yi-Rui Yang**     **Wu-Jun Li**[*]
National Key Laboratory for Novel Software Technology,
School of Computer Science, Nanjing University, Nanjing, China
{shicw, yangyr}@smail.nju.edu.cn, liwujun@nju.edu.cn

## Abstract

Distributed learning is essential for training large-scale deep models. Asynchronous SGD (ASGD) and its variants are commonly used distributed learning methods, particularly in scenarios where the computing capabilities of workers in the cluster are heterogeneous. Momentum has been acknowledged for its benefits in both optimization and generalization in deep model training. However, existing works have found that naively incorporating momentum into ASGD can impede the convergence. In this paper, we propose a novel method called ordered momentum (OrMo) for ASGD. In OrMo, momentum is incorporated into ASGD by organizing the gradients in order based on their iteration indexes. We theoretically prove the convergence of OrMo with both constant and delay-adaptive learning rates for non-convex problems. To the best of our knowledge, this is the first work to establish the convergence analysis of ASGD with momentum without dependence on the maximum delay. Empirical results demonstrate that OrMo can achieve better convergence performance compared with ASGD and other asynchronous methods with momentum.

## 1   Introduction

Many machine learning problems can be formulated as optimization problems of the following form:

$$\min_{\mathbf{w} \in \mathbb{R}^d} F(\mathbf{w}) = \mathbb{E}_{\xi \sim \mathcal{D}} \left[ f(\mathbf{w}; \xi) \right], \tag{1}$$

where $\mathbf{w}$ denotes the model parameter, $d$ is the dimension of the parameter, $\mathcal{D}$ represents the distribution of the training instances and $f(\mathbf{w}; \xi)$ denotes the loss on the training instance $\xi$.

Stochastic gradient descent (SGD) [28] and its variants [6, 12] are widely employed to solve the problem in (1). At each iteration, SGD uses one stochastic gradient or a mini-batch of stochastic gradients as an estimate of the full gradient to update the model parameter. In practice, momentum [26, 36, 24, 33] is often incorporated into SGD as a crucial technique for faster convergence and better generalization performance. Many popular machine learning libraries, such as TensorFlow [1] and PyTorch [25], include SGD with momentum (SGDm) as one of the optimizers.

Due to the rapid increase in the sizes of both models and datasets in recent years, a single machine is often insufficient to complete the training task of machine learning models within a reasonable time. Distributed learning [42, 37] aims to distribute the computations across multiple machines (workers) to accelerate the training process. Because of its necessity for training large-scale machine learning models, distributed learning has become a hot research topic in recent years. Existing distributed learning methods can be categorized into two main types: synchronous distributed learning (SDL) methods [19, 35, 29, 38, 44, 39] and asynchronous distributed learning (ADL) methods [2, 27, 8, 47, 17, 43]. In SDL methods, faster workers that have completed the computation must wait idly for

---

[*]Corresponding author.

the other slower workers in each communication round. Hence, the speed of SDL methods is often hindered by slow workers. In contrast, faster workers do not necessarily wait idly for the other slower workers in ADL methods, because ADL methods require aggregating information from only one worker or a subset of workers in each communication round. Representative ADL methods include asynchronous SGD (ASGD) and its variants [2, 8, 46, 30, 49, 48, 7, 3, 31, 22, 13]. In ASGD, once a worker finishes its gradient computation, the parameter (typically on the server) is immediately updated using this gradient through an SGD step, without waiting for other workers.

Momentum has been acknowledged for its benefits in both optimization and generalization in deep model training [33]. In SDL methods, momentum is extensively utilized across various domains, including decentralized algorithms [18, 45], communication compression algorithms [19, 29, 38, 40, 34, 41], infrequent communication algorithms [44, 39, 40], and federated learning algorithms [21, 32]. However, in ADL methods, some works [23, 9] have found that naively incorporating momentum into ASGD may decrease the convergence rate or even result in divergence. To tackle this challenge, some more sophisticated methods have been proposed to incorporate momentum into ASGD. The works in [23, 9] recommend tuning the momentum coefficient to enhance convergence performance when naively incorporating momentum into ASGD. The work in [9] proposes shifted momentum, which maintains local momentum on each worker. Inspired by Nesterov's accelerated gradient, the work in [4] proposes SMEGA$^2$, which leverages the momentum to estimate the future parameter. However, the process of tuning the momentum coefficient in [23, 9] is time-consuming and yields limited improvement in practice. Although shifted momentum and SMEGA$^2$ can achieve better empirical convergence performance than the method which naively incorporates momentum into ASGD, both of them lack theoretical convergence analysis.

In this paper, we propose a novel method, called ordered momentum (OrMo), for asynchronous SGD. The main contributions of this paper are outlined as follows:

- OrMo incorporates momentum into ASGD by organizing the gradients in order based on their iteration indexes.

- We theoretically prove the convergence of OrMo with both constant and delay-adaptive learning rates for non-convex problems. To the best of our knowledge, this is the first work to establish the convergence analysis for ASGD with momentum without dependence on the maximum delay.

- Empirical results demonstrate that OrMo can achieve better convergence performance compared with ASGD and other asynchronous methods with momentum.

## 2  Preliminary

In this paper, we use $\| \cdot \|$ to denote the $L_2$ norm. For a positive integer $n$, we use $[n]$ to denote the set $\{0, 1, 2, \ldots, n-1\}$. $\nabla f(\mathbf{w}; \xi)$ denotes the stochastic gradient computed over the training instance $\xi$ and model parameter $\mathbf{w}$. In this paper, we focus on the widely used Parameter Server framework [15], where the server is responsible for storing and updating the model parameter and the workers are responsible for sampling training instances and computing stochastic gradients. For simplicity, we assume that each worker samples one training instance for gradient computation each time. The analysis of mini-batch sampling on each worker follows a similar approach.

One of the most representative methods for distributing SGD across multiple workers is Synchronous SGD (SSGD) [20, 10]. Distributed SGD (DSGD), as presented in Algorithm 1, unifies SSGD and ASGD within a single framework [13]. The waiting set $\mathcal{C}$ in Algorithm 1 is a collection of workers (indexes) that are awaiting the server to send the latest parameter. The only difference between SSGD and ASGD is the communication scheduler associated with the waiting set. SSGD corresponds to DSGD with a synchronous communication scheduler, while ASGD corresponds to DSGD with an asynchronous communication scheduler. We use $\mathbf{g}^{k_t}_{ite(k_t,t)}$ to denote the stochastic gradient $\nabla f(\mathbf{w}_{ite(k_t,t)}; \xi^{k_t})$, where $k_t$ is the index of the worker whose gradient participates in the parameter update at iteration $t$ and $\xi^{k_t}$ denotes a training instance sampled on worker $k_t$. The function $ite(k,t)$ denotes the iteration index of the latest parameter sent to worker $k$ before iteration $t$, where $k \in [K]$ and $t \in [T]$. The delay of the gradient $\mathbf{g}^{k_t}_{ite(k_t,t)}$ is defined as $\tau_t = t - ite(k_t,t)$. When $K = 1$, DSGD degenerates to vanilla SGD, i.e., $ite(k_t,t) \equiv t$.

**Algorithm 1** Distributed SGD

---

 1: **Server:**
 2: **Input**: number of workers $K$, number of iterations $T$, learning rate $\eta$;
 3: **Initialization**: initial parameter $\mathbf{w}_0$, waiting set $\mathcal{C} = \emptyset$;
 4: Send the initial parameter $\mathbf{w}_0$ to all workers;
 5: **for** $t = 0$ **to** $T - 1$ **do**
 6:     Receive a stochastic gradient $\mathbf{g}_{ite(k_t,t)}^{k_t}$ from some worker $k_t$;
 7:     Update the parameter $\mathbf{w}_{t+1} = \mathbf{w}_t - \eta \mathbf{g}_{ite(k_t,t)}^{k_t}$;
 8:     Add the worker $k_t$ to the waiting set $\mathcal{C} = \mathcal{C} \cup \{k_t\}$;
 9:     Execute the communication scheduler:
        Option I: (Synchronous) only when all the workers are in the waiting set, i.e., $\mathcal{C} = [K]$, send the parameter $\mathbf{w}_{t+1}$ to the workers in $\mathcal{C}$ and set $\mathcal{C}$ to $\emptyset$;
        Option II: (Asynchronous) once the waiting set is not empty, i.e., $\mathcal{C} \neq \emptyset$, immediately send the parameter $\mathbf{w}_{t+1}$ to the worker in $\mathcal{C}$ and set $\mathcal{C}$ to $\emptyset$;
10: **end for**
11: Notify all workers to stop;
12: **Worker** $k : (k \in [K])$
13: **repeat**
14:     Wait until receiving the parameter $\mathbf{w}$ from the server;
15:     Randomly sample $\xi^k \sim \mathcal{D}$ and then compute the stochastic gradient $\mathbf{g}^k = \nabla f(\mathbf{w}; \xi^k)$;
16:     Send the stochastic gradient $\mathbf{g}^k$ to the server;
17: **until** receive server's notification to stop

---

In ASGD, the latest parameter $\mathbf{w}_{t+1}$ will be immediately sent back to the worker after the server updates the parameter at each iteration. The function $ite(k,t)$ in ASGD can be formulated as follows:

$$
ite(k,t) = \begin{cases} 0 & t = 0, k \in [K], \\ t & t > 0, k = k_{t-1}, \\ ite(k, t-1) & t > 0, k \neq k_{t-1}, \end{cases}
$$

where $k \in [K], t \in [T]$.

In SSGD, there is a barrier in the synchronous communication scheduler since the latest parameter $\mathbf{w}_{t+1}$ will be sent back to the workers only when all the workers are in the waiting set. The function $ite(k,t)$ in SSGD can be formulated as $ite(k,t) = \lfloor \frac{t}{K} \rfloor K$, where $k \in [K], t \in [T]$ and $\lfloor \cdot \rfloor$ is the floor function.

**Remark 1.** *In existing works [10, 22], SSGD is often presented in the form of mini-batch SGD:*

$$
\tilde{\mathbf{w}}_{s+1} = \tilde{\mathbf{w}}_s - \frac{\tilde{\eta}}{K} \sum_{k \in [K]} \nabla f(\tilde{\mathbf{w}}_s; \xi^k), \tag{2}
$$

*where $s \in [S]$ and $S$ denotes the number of iterations. Here, all workers aggregate their stochastic gradients to obtain the mini-batch gradient $\frac{1}{K} \sum_{k \in [K]} \nabla f(\tilde{\mathbf{w}}_s; \xi^k)$, which is then used to update the parameter. To unify SSGD and ASGD into a single framework in Algorithm 1, we reformulate SSGD in the form of mini-batch SGD in (2). Specifically, one update using a mini-batch gradient computed over $K$ training instances in (2) is split into $K$ updates, each using a stochastic gradient over a single training instance. Letting $\eta = \frac{\tilde{\eta}}{K}, T = KS$ and $\mathbf{w}_0 = \tilde{\mathbf{w}}_0$, the sequence $\{\mathbf{w}_{sK}\}_{s \in [S]}$ in SSGD in Algorithm 1 matches $\{\tilde{\mathbf{w}}_s\}_{s \in [S]}$ in (2).*

## 3   Ordered Momentum

In this section, we first propose a new reformulation of SSGD with momentum, which inspires the design of ordered momentum (OrMo) for ASGD. Then, we present the details of OrMo, including the algorithm and convergence analysis.

### 3.1 Reformulation of SSGD with Momentum

The widely used SGD with momentum (SGDm) [26] can be expressed as follows:

$$\tilde{\mathbf{w}}_{s+1} = \tilde{\mathbf{w}}_s - \beta\tilde{\mathbf{u}}_s - \frac{\tilde{\eta}}{|\mathcal{B}_s|}\sum_{\xi\in\mathcal{B}_s}\nabla f\left(\tilde{\mathbf{w}}_s;\xi\right), \tag{3}$$

$$\tilde{\mathbf{u}}_{s+1} = \beta\tilde{\mathbf{u}}_s + \frac{\tilde{\eta}}{|\mathcal{B}_s|}\sum_{\xi\in\mathcal{B}_s}\nabla f\left(\tilde{\mathbf{w}}_s;\xi\right), \tag{4}$$

where $\tilde{\mathbf{u}}_0 = \mathbf{0}, \beta \in [0,1), s \in [S]$ and $S$ denotes the number of iterations. $\beta$ is the momentum coefficient. $\tilde{\mathbf{u}}_s$ represents the Polyak's momentum. $\frac{1}{|\mathcal{B}_s|}\sum_{\xi\in\mathcal{B}_s}\nabla f\left(\tilde{\mathbf{w}}_s;\xi\right)$ denotes the stochastic gradient computed over the sampled training instance set $\mathcal{B}_s$, which contains either a single training instance or a mini-batch of training instances sampled from $\mathcal{D}$. (3) denotes the parameter update step and (4) denotes the momentum update step. When $\beta = 0$, SGDm degenerates to (mini-batch) SGD.

Since SSGD can be presented in the form of mini-batch SGD as depicted in Remark 1, it's straightforward to implement SSGD with momentum (SSGDm) as follows:

$$\tilde{\mathbf{w}}_{s+1} = \tilde{\mathbf{w}}_s - \beta\tilde{\mathbf{u}}_s - \frac{\tilde{\eta}}{K}\sum_{k\in[K]}\nabla f(\tilde{\mathbf{w}}_s;\xi^k),$$

$$\tilde{\mathbf{u}}_{s+1} = \beta\tilde{\mathbf{u}}_s + \frac{\tilde{\eta}}{K}\sum_{k\in[K]}\nabla f(\tilde{\mathbf{w}}_s;\xi^k), \tag{5}$$

where $\tilde{\mathbf{u}}_0 = \mathbf{0}, \beta \in [0,1), s \in [S]$ and $S$ denotes the number of iterations. Here, the server aggregates the stochastic gradients from all the workers to obtain the mini-batch gradient $\frac{1}{K}\sum_{k\in[K]}\nabla f(\tilde{\mathbf{w}}_s;\xi^k)$, which is then used to update both the parameter and the momentum in (5).

To gain insights from SSGDm on incorporating momentum into ASGD, we reformulate SSGDm in (5) to fit into the framework of Algorithm 1. Similar to the reformulation in Remark 1, the updates using a mini-batch gradient computed over $K$ training instances in (5) are split into $K$ updates, each using a stochastic gradient over a single training instance. The corresponding implementation details of SSGDm are presented in Algorithm 3 in Appendix B. In this way, the update rules of SSGDm in (5) can be reformulated as follows:

$$\mathbf{w}_{t+\frac{1}{2}} = \begin{cases} \mathbf{w}_t - \beta\mathbf{u}_t & K\mid t, \\ \mathbf{w}_t & K\nmid t, \end{cases}$$

$$\mathbf{u}_{t+\frac{1}{2}} = \begin{cases} \beta\mathbf{u}_t & K\mid t, \\ \mathbf{u}_t & K\nmid t, \end{cases} \tag{6}$$

$$\mathbf{w}_{t+1} = \mathbf{w}_{t+\frac{1}{2}} - \eta\mathbf{g}^{k_t}_{\lfloor\frac{t}{K}\rfloor K},$$

$$\mathbf{u}_{t+1} = \mathbf{u}_{t+\frac{1}{2}} + \eta\mathbf{g}^{k_t}_{\lfloor\frac{t}{K}\rfloor K},$$

where $\mathbf{u}_0 = \mathbf{0}, \mathbf{g}^{k_t}_{\lfloor\frac{t}{K}\rfloor K} = \nabla f(\mathbf{w}_{\lfloor\frac{t}{K}\rfloor K};\xi^{k_t})$ and $t \in [T]$. We give the following proposition about the relationship between the sequences in (5) and those in (6). The proof details can be found in Appendix C.1.1.

**Proposition 1.** *Letting $\eta = \frac{\tilde{\eta}}{K}, T = KS$ and $\mathbf{w}_0 = \tilde{\mathbf{w}}_0$, the sequences $\{\mathbf{w}_{sK}\}_{s\in[S]}$ and $\{\mathbf{u}_{sK}\}_{s\in[S]}$ in (6) are equivalent to $\{\tilde{\mathbf{w}}_s\}_{s\in[S]}$ and $\{\tilde{\mathbf{u}}_s\}_{s\in[S]}$ in (5), respectively.*

We investigate how the momentum term $\mathbf{u}_{t+1}$ evolves during the iterations in (6). For $t \geq K$ and $t \in [T]$, $\mathbf{u}_{t+1}$ can be formulated as:

$$\mathbf{u}_{t+1} = \sum_{i=0}^{\lfloor\frac{t}{K}\rfloor-1}\left(\beta^{\lfloor\frac{t}{K}\rfloor-i}\times\sum_{k\in[K]}\eta\mathbf{g}^k_{iK}\right) + \beta^0\times\sum_{j=\lfloor\frac{t}{K}\rfloor K}^{t}\eta\mathbf{g}^{k_j}_{\lfloor\frac{t}{K}\rfloor K},$$

where the superscript of the scalar $\beta$ indicates the exponent. For $t < K$, $\mathbf{u}_{t+1} = \beta^0\times\sum_{j=0}^{t}\eta\mathbf{g}^{k_j}_0$. Figure 1 shows $\mathbf{u}_{10}$ as an example when $K = 4$. We define $\{\eta\mathbf{g}^0_{iK}, \eta\mathbf{g}^1_{iK}, \cdots, \eta\mathbf{g}^{K-1}_{iK}\}$ as the $i$-th

Figure 1: An example of the momentum term $\mathbf{u}_{10}$ in SSGDm when $K = 4$. The gradients shown in red indicate those having not arrived at the server. In this case, $\mathbf{u}_{10} = \beta^2 \times \left( \eta \mathbf{g}_0^0 + \eta \mathbf{g}_0^1 + \eta \mathbf{g}_0^2 + \eta \mathbf{g}_0^3 \right) + \beta^1 \times \left( \eta \mathbf{g}_4^0 + \eta \mathbf{g}_4^1 + \eta \mathbf{g}_4^2 + \eta \mathbf{g}_4^3 \right) + \beta^0 \times \left( \eta \mathbf{g}_8^0 + \eta \mathbf{g}_8^3 \right)$.

(scaled) gradient group, which contains $K$ gradients scaled by the learning rate $\eta$. The order of the gradient groups is based on the iteration indexes of their corresponding gradients. Though some gradients may be missing because they have not yet arrived at the server, the momentum is a weighted sum of the gradients from the first several gradient groups. Hence, the momentum in SSGDm is referred to as an *ordered momentum*. Specifically, the gradients in the $i$-th gradient group are weighted by $\beta^{\lfloor \frac{t}{K} \rfloor - i}$ in the momentum $\mathbf{u}_{t+1}$, where $i \in [\lfloor \frac{t}{K} \rfloor + 1]$. We refer to the gradient group whose gradients are weighted by $\beta^0$ as the latest gradient group, which contains the latest gradients. For $\mathbf{u}_{t+1}$ in SSGDm, the latest gradient group corresponds to the $\lfloor \frac{t}{K} \rfloor$-th gradient group.

Due to the barrier in the synchronous communication scheduler in SSGDm as presented in Algorithm 3, the gradients in SSGDm consistently arrive at the server in the order of their iteration indexes. The arriving gradient always belongs to the latest gradient group at each iteration. Thus, maintaining such an ordered momentum in SSGDm is straightforward. As shown in line 13 of Algorithm 3, the scaled gradient $\eta \mathbf{g}_{ite(k_t,t)}^{k_t}$ is always added to the momentum with a weight of $\beta^0$ at each iteration. However, for ASGD, since the gradients arrive at the server out of order, it's not trivial to incorporate such an ordered momentum. To address this problem, we propose a solution in the following subsection.

## 3.2 OrMo for ASGD

In this subsection, we introduce our novel method called ordered momentum (OrMo) for ASGD, and present it in Algorithm 2.

Firstly, we define the (scaled) gradient groups in OrMo for ASGD. Due to the differences in the communication scheduler, the iteration indexes of the parameters used to compute the gradients in ASGD differ from those in SSGD (SSGDm). Specifically, the sequence of gradients computed in SSGD (SSGDm) can be formulated as:

$$\mathbf{g}_0^0, \mathbf{g}_0^1, \cdots, \mathbf{g}_0^{K-1}, \mathbf{g}_K^0, \mathbf{g}_K^1, \cdots, \mathbf{g}_K^{K-1}, \mathbf{g}_{2K}^0, \mathbf{g}_{2K}^1, \cdots, \mathbf{g}_{2K}^{K-1}, \cdots. \tag{7}$$

In contrast, the sequence of gradients computed in ASGD is given by:

$$\mathbf{g}_0^0, \mathbf{g}_0^1, \cdots, \mathbf{g}_0^{K-1}, \mathbf{g}_1^{k_0}, \mathbf{g}_2^{k_1}, \cdots, \mathbf{g}_K^{k_{K-1}}, \mathbf{g}_{K+1}^{k_K}, \mathbf{g}_{K+2}^{k_{K+1}}, \cdots, \mathbf{g}_{2K}^{k_{2K-1}}, \cdots. \tag{8}$$

Thus, the $i$-th (scaled) gradient group in OrMo for ASGD is defined as:

$$\left\{ \eta \mathbf{g}_{(i-1)K+1}^{k_{(i-1)K}}, \eta \mathbf{g}_{(i-1)K+2}^{k_{(i-1)K+1}}, \cdots, \eta \mathbf{g}_{iK}^{k_{iK-1}} \right\},$$

where $i \geq 1$. The 0-th (scaled) gradient group in OrMo is $\{\eta \mathbf{g}_0^0, \eta \mathbf{g}_0^1, \cdots, \eta \mathbf{g}_0^{K-1}\}$. Despite the difference in the gradients' iteration indexes, each gradient group in OrMo for ASGD also contains $K$ gradients scaled by the learning rate $\eta$, similar to that in SSGDm as discussed in Subsection 3.1.

We use $I_{t+1}$ to denote the index of the latest gradient group of $\mathbf{u}_{t+1}$ in OrMo. The iteration index of the latest gradient in $\mathbf{u}_{t+1}$ can be $t$ at most. Since the gradient with iteration index $t$ belongs to the $\lceil \frac{t}{K} \rceil$-th gradient group, the latest gradient group for $\mathbf{u}_{t+1}$ should be the $\lceil \frac{t}{K} \rceil$-th gradient group, i.e., $I_{t+1} \equiv \lceil \frac{t}{K} \rceil, \forall t \in [T]$. $\mathbf{u}_{t+1}$ is the weighted sum of the gradients from the first $I_{t+1} + 1$ gradient groups, where some gradients may be missing because they have not yet arrived at the server. The gradients in the $i$-th gradient group are weighted by $\beta^{\lceil \frac{t}{K} \rceil - i}$ in the momentum $\mathbf{u}_{t+1}$, where $i \in [I_{t+1} + 1]$ and the superscript of the scalar $\beta$ indicates the exponent. Figure 2 shows an example of $\mathbf{u}_{10}$ in OrMo when $K = 4$.

For the $t$-th iteration in OrMo, the server performs the following operations:

**Algorithm 2** OrMo

1: **Server:**
2: **Input**: number of workers $K$, number of iterations $T$, learning rate $\eta$, momentum coefficient $\beta \in [0,1)$;
3: **Initialization**: initial parameter $\mathbf{w}_0$, momentum $\mathbf{u}_0 = \mathbf{0}$, index of the latest gradient group $I_0 = 0$, waiting set $\mathcal{C} = \emptyset$;
4: Send the initial parameter $\mathbf{w}_0$ and its iteration index 0 to all workers;
5: **for** $t = 0$ **to** $T - 1$ **do**
6:      **if** the waiting set $\mathcal{C}$ is empty and $\lceil \frac{t}{K} \rceil > I_t$ **then**
7:          $\mathbf{w}_{t+\frac{1}{2}} = \mathbf{w}_t - \beta \mathbf{u}_t, \mathbf{u}_{t+\frac{1}{2}} = \beta \mathbf{u}_t, I_{t+1} = I_t + 1$;
8:      **else**
9:          $\mathbf{w}_{t+\frac{1}{2}} = \mathbf{w}_t, \mathbf{u}_{t+\frac{1}{2}} = \mathbf{u}_t, I_{t+1} = I_t$;
10:      **end if**
11:      Receive a stochastic gradient $\mathbf{g}^{k_t}_{ite(k_t,t)}$ and its iteration index $ite(k_t,t)$ from some worker $k_t$ and then calculate $\lceil \frac{ite(k_t,t)}{K} \rceil$ (i.e., the index of the gradient group that $\mathbf{g}^{k_t}_{ite(k_t,t)}$ belongs to);
12:      Update the momentum $\mathbf{u}_{t+1} = \mathbf{u}_{t+\frac{1}{2}} + \beta^{I_{t+1} - \lceil \frac{ite(k_t,t)}{K} \rceil} \times \left( \eta \mathbf{g}^{k_t}_{ite(k_t,t)} \right)$;
13:      Update the parameter $\mathbf{w}_{t+1} = \mathbf{w}_{t+\frac{1}{2}} - \frac{1 - \beta^{I_{t+1} - \lceil \frac{ite(k_t,t)}{K} \rceil + 1}}{1 - \beta} \times \left( \eta \mathbf{g}^{k_t}_{ite(k_t,t)} \right)$;
14:      Add the worker $k_t$ to the waiting set $\mathcal{C} = \mathcal{C} \cup \{k_t\}$;
15:      Execute the asynchronous communication scheduler: once the waiting set is not empty, i.e., $\mathcal{C} \neq \emptyset$, immediately send the parameter $\mathbf{w}_{t+1}$ and its iteration index $t+1$ to the worker in $\mathcal{C}$ and set $\mathcal{C}$ to $\emptyset$;
16: **end for**
17: Notify all workers to stop;
18: **Worker $k$ :** $(k \in [K])$
19: **repeat**
20:      Wait until receiving the parameter $\mathbf{w}_{t'}$ and its iteration index $t'$ from the server;
21:      Randomly sample $\xi^k \sim \mathcal{D}$ and then compute the stochastic gradient $\mathbf{g}^k_{t'} = \nabla f(\mathbf{w}_{t'}; \xi^k)$;
22:      Send the stochastic gradient $\mathbf{g}^k_{t'}$ and its iteration index $t'$ to the server;
23: **until** receive server's notification to stop

Figure 2: An example of the momentum term $\mathbf{u}_{10}$ in OrMo when $K = 4$. The gradients shown in red indicate those having not arrived at the server. In this case, $\mathbf{u}_{10} = \beta^3 \times \left( \eta \mathbf{g}^0_0 + \eta \mathbf{g}^1_0 + \eta \mathbf{g}^2_0 + \eta \mathbf{g}^3_0 \right) + \beta^2 \times \left( \eta \mathbf{g}^{k_0}_1 + \eta \mathbf{g}^{k_1}_2 + \eta \mathbf{g}^{k_2}_3 \right) + \beta^1 \times \left( \eta \mathbf{g}^{k_5}_6 + \eta \mathbf{g}^{k_7}_8 \right) + \beta^0 \times \left( \eta \mathbf{g}^{k_8}_9 \right)$.

- If the parameter with iteration index $t$ that satisfies $\lceil \frac{t}{K} \rceil > I_t$ has been sent to some worker, update the parameter using the momentum and multiply the momentum with $\beta$: $\mathbf{w}_{t+\frac{1}{2}} = \mathbf{w}_t - \beta \mathbf{u}_t, \mathbf{u}_{t+\frac{1}{2}} = \beta \mathbf{u}_t, I_{t+1} = I_t + 1$.

  In this way, the momentum changes the index of its latest gradient group to $\lceil \frac{t}{K} \rceil$ and gets ready to accommodate the new gradient with iteration index $t$.

- Receive a stochastic gradient $\mathbf{g}^{k_t}_{ite(k_t,t)}$ and its iteration index $ite(k_t,t)$ from some worker $k_t$ and calculate $\lceil \frac{ite(k_t,t)}{K} \rceil$, which is the index of the gradient group that $\mathbf{g}^{k_t}_{ite(k_t,t)}$ belongs to.

- Update the momentum: $\mathbf{u}_{t+1} = \mathbf{u}_{t+\frac{1}{2}} + \beta^{I_{t+1} - \lceil \frac{ite(k_t,t)}{K} \rceil} \times \left( \eta \mathbf{g}^{k_t}_{ite(k_t,t)} \right)$.

  Since the weight of the scaled gradients from the latest gradient group in the momentum is $\beta^0$, the weight of the gradients from the $\lceil \frac{ite(k_t,t)}{K} \rceil$-th gradient group should be $\beta^{I_{t+1} - \lceil \frac{ite(k_t,t)}{K} \rceil}$.

OrMo updates the momentum by adding the scaled gradient $\eta\mathbf{g}_{ite(k_t,t)}^{k_t}$ into the momentum with a weight of $\beta^{I_{t+1}-\lceil\frac{ite(k_t,t)}{K}\rceil}$.

- Update the parameter: $\mathbf{w}_{t+1} = \mathbf{w}_{t+\frac{1}{2}} - \frac{1-\beta^{I_{t+1}-\lceil\frac{ite(k_t,t)}{K}\rceil+1}}{1-\beta} \times \left(\eta\mathbf{g}_{ite(k_t,t)}^{k_t}\right)$.

  The update rule of the parameter in OrMo is motivated by that in SSGDm, as presented in Algorithm 3. In SSGDm, the scaled gradient $\eta\mathbf{g}_{ite(k_t,t)}^{k_t}$ is always added to the momentum with a weight of $\beta^0$. At the current iteration, this scaled gradient updates the parameter with a coefficient $-\beta^0$. In subsequent iterations, this scaled gradient in the momentum updates the parameter with the coefficients $-\beta, -\beta^2, -\beta^3, \cdots$. By the time this scaled gradient is weighted by $\beta^{I_{t+1}-\lceil\frac{ite(k_t,t)}{K}\rceil}$ in the momentum of SSGDm, it has already updated the parameter for $I_{t+1} - \lceil\frac{ite(k_t,t)}{K}\rceil + 1$ steps, with a total coefficient $-\sum_{j=0}^{I_{t+1}-\lceil\frac{ite(k_t,t)}{K}\rceil}\beta^j$. In OrMo, for the scaled gradient $\eta\mathbf{g}_{ite(k_t,t)}^{k_t}$ which is added to the momentum with a weight of $\beta^{I_{t+1}-\lceil\frac{ite(k_t,t)}{K}\rceil}$, we compensate for the missed $I_{t+1} - \lceil\frac{ite(k_t,t)}{K}\rceil + 1$ steps compared with SSGDm and update the parameter with the coefficient $-\frac{1-\beta^{I_{t+1}-\lceil\frac{ite(k_t,t)}{K}\rceil+1}}{1-\beta}$ at the current iteration. The design of the parameter update rule is crucial for the derivation of Lemma 2, which is further supported by the ablation study in Appendix A.3.

- Add the worker $k_t$ to the waiting set $\mathcal{C} = \mathcal{C} \cup \{k_t\}$ and execute the asynchronous communication scheduler.

**Remark 2.** *Compared to ASGD, the additional communication overhead introduced by the iteration index in OrMo is negligible since the iteration index is only a scalar.*

**Remark 3.** *When the momentum coefficient $\beta$ is set to $0$, OrMo degenerates to ASGD in Algorithm 1. If the asynchronous communication scheduler in line 15 of Algorithm 2 is replaced by a synchronous communication scheduler: only when all the workers are in the waiting set, i.e., $\mathcal{C} = [K]$, send the parameter $\mathbf{w}_{t+1}$ and the iteration index $t + 1$ to the workers in $\mathcal{C}$ and set $\mathcal{C}$ to $\emptyset$, OrMo degenerates to SSGDm in Algorithm 3.*

### 3.3 Convergence Analysis

In this section, we prove the convergence of OrMo in Algorithm 2 for non-convex problems. We only present the main results here. The proof details can be found in Appendix C.

We make the following assumptions, which are widely used in distributed learning [47, 43, 41, 22].

**Assumption 1.** *For any stochastic gradient $\nabla f(\mathbf{w}; \xi)$, we assume that it satisfies:*

$$\mathbb{E}_{\xi\sim\mathcal{D}}[\nabla f(\mathbf{w}; \xi)] = \nabla F(\mathbf{w}), \mathbb{E}_{\xi\sim\mathcal{D}}\|\nabla f(\mathbf{w}; \xi) - \nabla F(\mathbf{w})\|^2 \leq \sigma^2, \forall\mathbf{w} \in \mathbb{R}^d.$$

**Assumption 2.** *For any stochastic gradient $\nabla f(\mathbf{w}; \xi)$, we assume that it satisfies:*

$$\mathbb{E}_{\xi\sim\mathcal{D}}\|\nabla f(\mathbf{w}; \xi)\|^2 \leq G^2, \forall\mathbf{w} \in \mathbb{R}^d.$$

**Assumption 3.** *$F(\mathbf{w})$ is $L$-smooth ($L > 0$):*

$$F(\mathbf{w}) \leq F(\mathbf{w}') + \nabla F(\mathbf{w}')^T(\mathbf{w} - \mathbf{w}') + \frac{L}{2}\|\mathbf{w} - \mathbf{w}'\|^2, \forall\mathbf{w}, \mathbf{w}' \in \mathbb{R}^d.$$

**Assumption 4.** *The objective function $F(\mathbf{w})$ is lower bounded by $F^*$: $F(\mathbf{w}) \geq F^*, \forall\mathbf{w} \in \mathbb{R}^d$.*

Firstly, we define the auxiliary sequence $\{\hat{\mathbf{u}}_t\}_{t\geq1}$ for the momentum: $\hat{\mathbf{u}}_1 = \sum_{k\in[K]}\eta\mathbf{g}_0^k$, and

$$\hat{\mathbf{u}}_{t+1} = \begin{cases} \beta\hat{\mathbf{u}}_t + \eta\mathbf{g}_t^{k_{t-1}} & K \mid (t-1), \\ \hat{\mathbf{u}}_t + \eta\mathbf{g}_t^{k_{t-1}} & K \nmid (t-1), \end{cases}$$

for $t \geq 1$.

**Lemma 1.** *For any $t \geq 0$, the gap between $\mathbf{u}_{t+1}$ and $\hat{\mathbf{u}}_{t+1}$ can be formulated as follows:*

$$\hat{\mathbf{u}}_{t+1} - \mathbf{u}_{t+1} = \sum_{k\in[K], k\neq k_t}\beta^{\lceil\frac{t}{K}\rceil-\lceil\frac{ite(k,t)}{K}\rceil}\eta\mathbf{g}_{ite(k,t)}^k. \tag{9}$$

Then, we define the auxiliary sequence $\{\hat{\mathbf{w}}_t\}_{t\geq 1}$ for the parameter: $\hat{\mathbf{w}}_1 = \mathbf{w}_0 - \sum_{k\in[K]}\eta\mathbf{g}_0^k$, and

$$\hat{\mathbf{w}}_{t+1} = \begin{cases} \hat{\mathbf{w}}_t - \beta\hat{\mathbf{u}}_t - \eta\mathbf{g}_t^{k_{t-1}} & K \mid (t-1), \\ \hat{\mathbf{w}}_t - \eta\mathbf{g}_t^{k_{t-1}} & K \nmid (t-1), \end{cases}$$

for $t \geq 1$.

**Lemma 2.** *For any $t \geq 0$, the gap between $\mathbf{w}_{t+1}$ and $\hat{\mathbf{w}}_{t+1}$ can be formulated as follows:*

$$\hat{\mathbf{w}}_{t+1} - \mathbf{w}_{t+1} = - \sum_{k\in[K], k\neq k_t} \frac{1 - \beta^{\lceil \frac{t}{K}\rceil - \lceil\frac{ite(k,t)}{K}\rceil + 1}}{1 - \beta}\eta\mathbf{g}_{ite(k,t)}^k. \tag{10}$$

Then, we define another auxiliary sequence $\{\hat{\mathbf{y}}_t\}_{t\geq 1}$: $\hat{\mathbf{y}}_1 = \frac{\hat{\mathbf{w}}_1 - \beta\mathbf{w}_0}{1-\beta}$, and $\hat{\mathbf{y}}_{t+1} = \hat{\mathbf{y}}_t - \frac{\eta}{1-\beta}\mathbf{g}_t^{k_{t-1}}$, for $t \geq 1$.

**Lemma 3.** *For any $t \geq 1$, the gap between $\hat{\mathbf{y}}_t$ and $\hat{\mathbf{w}}_t$ can be formulated as follows:*

$$\hat{\mathbf{y}}_t - \hat{\mathbf{w}}_t = -\frac{\beta}{1-\beta}\hat{\mathbf{u}}_t. \tag{11}$$

**Theorem 1.** *With Assumptions 1, 2, 3 and 4, letting $\eta = \min\{\frac{1-\beta}{2KL}, \frac{(1-\beta)\Delta^{\frac{1}{2}}}{(LT)^{\frac{1}{2}}\sigma}, \frac{(1-\beta)^{\frac{5}{3}}\Delta^{\frac{1}{3}}}{(LKG)^{\frac{2}{3}}T^{\frac{1}{3}}}\}$, Algorithm 2 has the following convergence rate:*

$$\frac{1}{T}\sum_{t=1}^T \mathbb{E}\|\nabla F(\mathbf{w}_t)\|^2 \leq \mathcal{O}\left(\sqrt{\frac{L\sigma^2}{T}} + \left(\frac{KLG}{T}\right)^{\frac{2}{3}} + \frac{KL}{T}\right),$$

*where $\Delta = F(\mathbf{w}_0) - F^*$ and $T \geq K$.*

Many works [46, 30, 7, 13, 22] consider delay-adaptive methods for ASGD. The key insight of these methods is to penalize the gradients with large delays and reduce their contribution to the parameter update. OrMo is orthogonal to these delay-adaptive methods. Concretely, we can replace the constant learning rate $\eta$ in Algorithm 2 with a delay-adaptive learning rate $\eta_t$, which is dependent on the delay of the gradient $\tau_t$. Inspired by [13], we adopt the following delay-adaptive learning rate $\eta_t$:

$$\eta_t = \begin{cases} \eta & \tau_t \leq 2K, \\ \min\{\eta, \dfrac{1}{4L\tau_t}\} & \tau_t > 2K. \end{cases}$$

The convergence of OrMo with the above delay-adaptive learning rate (called OrMo-DA) is guaranteed by Theorem 2.

**Theorem 2.** *With Assumptions 1, 3 and 4, letting $\eta = \min\{\frac{(1-\beta)^2}{8KL}, \sqrt{\frac{(1-\beta)^3\Delta}{TL\sigma^2}}\}$, OrMo-DA has the following convergence rate:*

$$\mathbb{E}\|\nabla F(\bar{\mathbf{w}}_T)\|^2 \leq \mathcal{O}\left(\sqrt{\frac{L\sigma^2}{T}} + \frac{KL}{T}\right),$$

*where $\Delta = F(\mathbf{w}_0) - F^*$ and $\bar{\mathbf{w}}_T$ is randomly chosen from $\{\mathbf{w}_0, \mathbf{w}_1, \cdots, \mathbf{w}_{T-1}\}$ according to a probability distribution which is related to the delay-adaptive learning rates.*

The proof details can be found in Appendix C.3. Compared with Theorem 1, Theorem 2 removes the dependence on Assumption 2 (bounded gradient) and provides a better convergence bound.

**Remark 4.** *We focus on the scenario where the training instances across all workers are independent and identically distributed (i.i.d.) from $\mathcal{D}$. This scenario commonly appears in the data-center setup for distributed training [5], where all workers have access to the full training dataset. Our analysis for the i.i.d. scenario can also provide insights into the analysis in a non-i.i.d. scenario [22], which will be studied in future work.*

**Remark 5.** *Most existing theoretical analyses of ASGD [16, 49, 3, 20] rely on the maximum delay $\tau_{max}$ (e.g., $\mathcal{O}(\sqrt{\frac{L\sigma^2}{T}} + \frac{\tau_{max}L}{T})$ in [20]), where $\tau_{max} = \max_{t\in[T]}\tau_t$. However, since ASGD can still perform well even when the maximum delay is extremely large ($\tau_{max} \gg K$) in practice, these theoretical analyses don't accurately reflect the true behavior of ASGD. The most closely related works to this work are [13, 22], which analyze ASGD without relying on the maximum delay. But the works in [13, 22] do not consider momentum. To the best of our knowledge, this is the first work to establish the convergence guarantee of ASGD with momentum without relying on the maximum delay.*

Table 1: Empirical results of different methods on CIFAR10 dataset.

| Number of Workers | 16 (hom.) | | 64 (hom.) | | 16 (het.) | | 64 (het.) | |
|---|---|---|---|---|---|---|---|---|
| Methods | Training Loss | Test Accuracy | Training Loss | Test Accuracy | Training Loss | Test Accuracy | Training Loss | Test Accuracy |
| ASGD | $0.06 \pm 0.00$ | $89.77 \pm 0.11$ | $0.40 \pm 0.02$ | $83.14 \pm 0.55$ | $0.06 \pm 0.00$ | $89.73 \pm 0.19$ | $0.38 \pm 0.01$ | $83.94 \pm 0.21$ |
| naive ASGDm | $0.20 \pm 0.07$ | $88.15 \pm 1.70$ | $0.44 \pm 0.06$ | $82.39 \pm 1.79$ | $0.58 \pm 0.86$ | $73.23 \pm 31.61$ | $0.78 \pm 0.77$ | $68.75 \pm 29.51$ |
| shifted momentum | $0.08 \pm 0.01$ | $90.23 \pm 0.27$ | $0.38 \pm 0.00$ | $83.72 \pm 0.29$ | $0.10 \pm 0.02$ | $89.95 \pm 0.32$ | $0.37 \pm 0.01$ | $83.99 \pm 0.23$ |
| SMEGA$^2$ | $0.05 \pm 0.01$ | $90.60 \pm 0.42$ | $0.23 \pm 0.04$ | $86.82 \pm 0.69$ | $0.04 \pm 0.01$ | $90.88 \pm 0.25$ | $0.22 \pm 0.07$ | $86.89 \pm 1.42$ |
| OrMo | $0.04 \pm 0.01$ | $90.95 \pm 0.27$ | $\mathbf{0.15 \pm 0.02}$ | $\mathbf{88.03 \pm 0.28}$ | $0.04 \pm 0.00$ | $91.01 \pm 0.10$ | $0.16 \pm 0.03$ | $87.76 \pm 0.57$ |
| OrMo-DA | $\mathbf{0.03 \pm 0.01}$ | $\mathbf{91.17 \pm 0.18}$ | $0.16 \pm 0.02$ | $88.03 \pm 0.33$ | $\mathbf{0.03 \pm 0.01}$ | $\mathbf{91.28 \pm 0.37}$ | $\mathbf{0.15 \pm 0.02}$ | $\mathbf{88.08 \pm 0.38}$ |

Table 2: Empirical results of different methods on CIFAR100 dataset.

| Number of Workers | 16 (hom.) | | 64 (hom.) | | 16 (het.) | | 64 (het.) | |
|---|---|---|---|---|---|---|---|---|
| Methods | Training Loss | Test Accuracy | Training Loss | Test Accuracy | Training Loss | Test Accuracy | Training Loss | Test Accuracy |
| ASGD | $0.51 \pm 0.01$ | $66.16 \pm 0.36$ | $0.96 \pm 0.03$ | $61.61 \pm 0.59$ | $0.51 \pm 0.01$ | $65.94 \pm 0.39$ | $0.95 \pm 0.01$ | $61.74 \pm 0.30$ |
| naive ASGDm | $0.54 \pm 0.01$ | $65.46 \pm 0.20$ | $1.03 \pm 0.05$ | $59.96 \pm 0.90$ | $0.53 \pm 0.00$ | $65.69 \pm 0.42$ | $0.97 \pm 0.06$ | $61.13 \pm 1.02$ |
| shifted momentum | $0.47 \pm 0.01$ | $66.37 \pm 0.14$ | $0.82 \pm 0.01$ | $63.55 \pm 0.32$ | $0.47 \pm 0.00$ | $66.28 \pm 0.14$ | $0.82 \pm 0.04$ | $63.28 \pm 0.66$ |
| SMEGA$^2$ | $0.41 \pm 0.00$ | $67.32 \pm 0.22$ | $0.69 \pm 0.00$ | $64.16 \pm 0.12$ | $0.40 \pm 0.01$ | $67.29 \pm 0.16$ | $0.68 \pm 0.02$ | $64.12 \pm 0.53$ |
| OrMo | $0.41 \pm 0.01$ | $67.56 \pm 0.34$ | $0.56 \pm 0.00$ | $65.48 \pm 0.17$ | $0.40 \pm 0.01$ | $67.71 \pm 0.33$ | $0.58 \pm 0.02$ | $65.43 \pm 0.35$ |
| OrMo-DA | $\mathbf{0.40 \pm 0.00}$ | $\mathbf{67.72 \pm 0.21}$ | $\mathbf{0.56 \pm 0.01}$ | $\mathbf{65.79 \pm 0.12}$ | $\mathbf{0.04 \pm 0.00}$ | $\mathbf{67.82 \pm 0.20}$ | $\mathbf{0.57 \pm 0.01}$ | $\mathbf{65.82 \pm 0.30}$ |

## 4    Experiments

In this section, we evaluate the performance of OrMo, OrMo-DA and other baseline methods. All the experiments are implemented based on the Parameter Server framework [15]. Our distributed platform is conducted with Docker. Each Docker container corresponds to either a server or a worker. All the methods are implemented with PyTorch 1.3.

The baseline methods include ASGD, *naive ASGDm* which naively incorporates momentum into ASGD [23], shifted momentum [9] and SMEGA$^2$ [4]. The details of naive ASGDm are shown in Algorithm 4. In OrMo-DA, for a gradient with a large delay satisfying $\tau_t > 2K$, its corresponding learning rate will be multiplied by $\frac{1}{\tau_t}$. We evaluate these methods by training ResNet20 model [11] on CIFAR10 and CIFAR100 datasets [14]. The number of workers is set to 16 and 64. The batch size on each worker is set to 64. The momentum coefficient is set to 0.9. Each experiment is repeated 5 times. The experiments are conducted under two settings:

- Setting I [homogeneous (hom.)]: each worker has similar computing capabilities, which ensures comparable average time for gradient computations.
- Setting II [heterogeneous (het.)]: some workers ($\frac{1}{16}$ of all) are designated as slow workers, with an average computation time that is 10 times longer than that of the others.

For the CIFAR10 dataset, the weight decay is set to 0.0001 and the model is trained with 160 epochs. The learning rate is multiplied by 0.1 at the 80-th and 120-th epoch, as suggested in [11]. The experiments for the CIFAR10 dataset are conducted on NVIDIA RTX 2080 Ti GPUs. For the CIFAR100 dataset, the weight decay is set to 0.0005 and the model is trained with 200 epochs. The learning rate is multiplied by 0.2 at the 60-th, 120-th and 160-th epoch, as suggested in [40]. The experiments for the CIFAR100 dataset are conducted on NVIDIA V100 GPUs.

Table 1 and Table 2 show the empirical results of different methods. Figure 3 and Figure 4 show the test accuracy curves of different methods. We also present the test accuracy curves of SSGDm for reference in Figure 3 and Figure 4. Training loss curves can be found in Appendix A.1. Compared with other asynchronous methods, OrMo and OrMo-DA can achieve better training loss and test accuracy. As shown in Figure 3, naive ASGDm occasionally fails to converge under Setting II. We can find that naively incorporating momentum into ASGD will impede its convergence. Due to the existence of slow workers, the maximum delay under Setting II is far greater than that under Setting I. For example, when training on the CIFAR10 dataset with $K = 64$, the maximum delay under Setting II is about 30000, while it's around 300 under Setting I. OrMo and OrMo-DA perform well under both settings, which aligns with our theoretical results without dependence on the maximum delay.

Figure 5 presents the training curves of OrMo and SSGDm with respect to wall-clock time. As shown in Figure 5(b), OrMo can be 8 times faster than SSGDm under Setting II since the training speed of SSGDm is hindered by slow workers. In contrast, slow workers have a limited impact on OrMo's training speed. Even under Setting I where each worker possesses similar computing

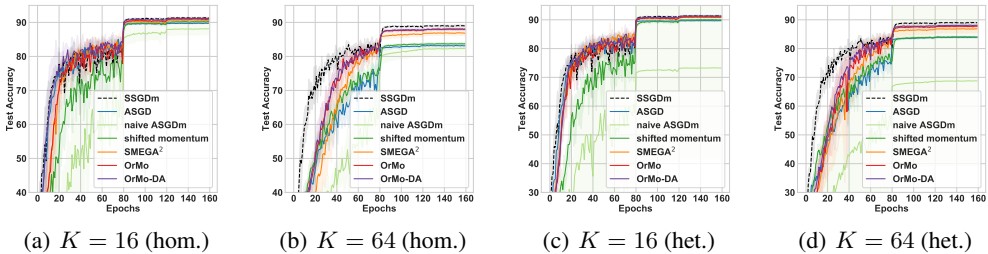

(a) $K = 16$ (hom.)  (b) $K = 64$ (hom.)  (c) $K = 16$ (het.)  (d) $K = 64$ (het.)

Figure 3: Test accuracy curves on CIFAR10 with different numbers of workers.

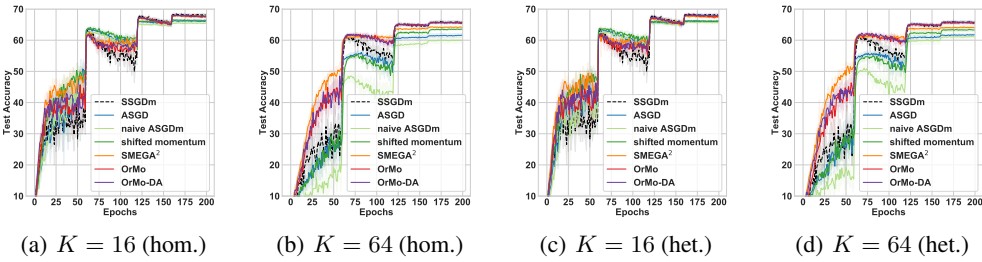

(a) $K = 16$ (hom.)  (b) $K = 64$ (hom.)  (c) $K = 16$ (het.)  (d) $K = 64$ (het.)

Figure 4: Test accuracy curves on CIFAR100 with different numbers of workers.

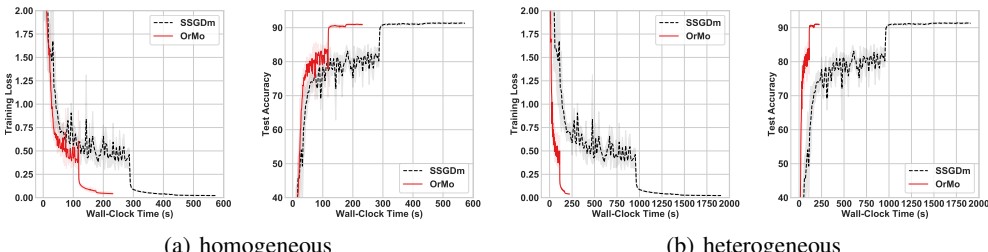

(a) homogeneous  (b) heterogeneous

Figure 5: Training curves with respect to wall-clock time on CIFAR10 when $K = 16$.

capability, OrMo can still be more than twice as fast as SSGDm, as shown in Figure 5(a). This advantage arises because the computation time of each worker varies within a certain range even under the homogeneous setting and some workers must wait for others to finish gradient computations in SSGDm.

## 5 Conclusion

In this paper, we propose a novel method named ordered momentum (OrMo) for asynchronous SGD. We theoretically prove the convergence of OrMo with both constant and delay-adaptive learning rates for non-convex problems. To the best of our knowledge, this is the first work to establish the convergence analysis of ASGD with momentum without dependence on the maximum delay. Empirical results demonstrate that OrMo can achieve state-of-the-art performance.

## Acknowledgment

This work is supported by National Key R&D Program of China (No. 2020YFA0713900), NSFC Project (No. 12326615), Major Key Project of Pengcheng Laboratory (No. PCL2024A06) and Key R&D Project of Jiangsu Province (No. BE2023652).

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

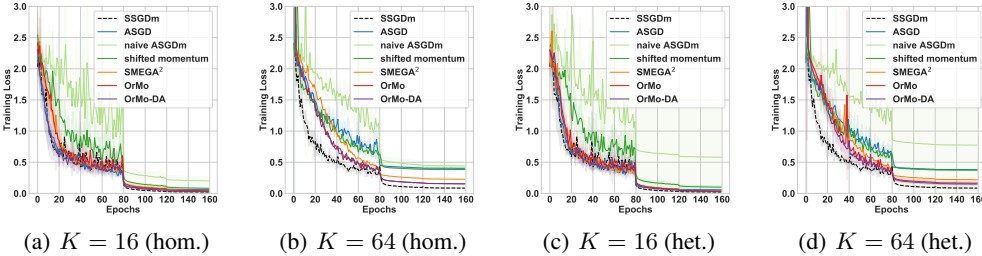

(a) $K = 16$ (hom.)     (b) $K = 64$ (hom.)     (c) $K = 16$ (het.)     (d) $K = 64$ (het.)

Figure 6: Training loss curves of different methods on CIFAR10 dataset with different numbers of workers.

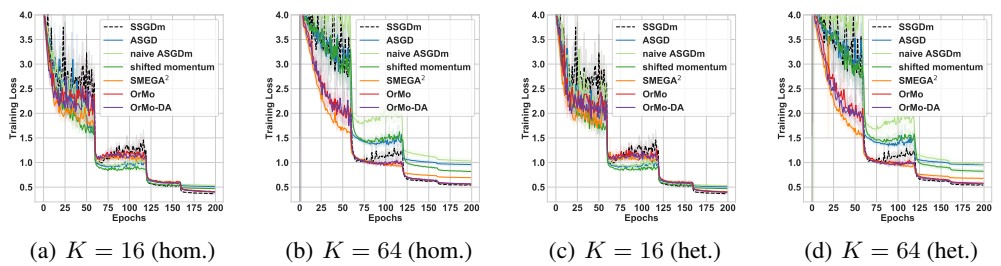

(a) $K = 16$ (hom.)     (b) $K = 64$ (hom.)     (c) $K = 16$ (het.)     (d) $K = 64$ (het.)

Figure 7: Training loss curves of different methods on CIFAR100 dataset with different numbers of workers.

Table 3: Empirical results of naive ASGDm with different $\beta$ when training ResNet20 on CIFAR10 dataset.

| Number of workers | 16 (hom.) | | 64 (hom.) | | 16 (het.) | | 64 (het.) | |
| Algorithm ($\beta$) | Training Loss | Test Accuracy | Training Loss | Test Accuracy | Training Loss | Test Accuracy | Training Loss | Test Accuracy |
|---|---|---|---|---|---|---|---|---|
| naive ASGDm (0.1) | $0.06 \pm 0.01$ | $89.85 \pm 0.24$ | $0.38 \pm 0.01$ | $83.93 \pm 0.25$ | $0.06 \pm 0.00$ | $89.95 \pm 0.19$ | $0.38 \pm 0.01$ | $83.76 \pm 0.34$ |
| naive ASGDm (0.3) | $0.06 \pm 0.00$ | $89.91 \pm 0.20$ | $0.36 \pm 0.02$ | $84.23 \pm 0.49$ | $0.05 \pm 0.01$ | $90.26 \pm 0.05$ | $0.35 \pm 0.02$ | $84.43 \pm 0.22$ |
| naive ASGDm (0.6) | $0.07 \pm 0.00$ | $90.39 \pm 0.24$ | $0.38 \pm 0.02$ | $83.87 \pm 0.28$ | $0.06 \pm 0.01$ | $90.56 \pm 0.13$ | $0.37 \pm 0.02$ | $84.07 \pm 0.38$ |
| naive ASGDm (0.9) | $0.20 \pm 0.07$ | $88.15 \pm 1.70$ | $0.44 \pm 0.06$ | $82.39 \pm 1.79$ | $0.58 \pm 0.86$ | $73.23 \pm 31.61$ | $0.78 \pm 0.77$ | $68.75 \pm 29.51$ |
| OrMo (0.9) | $\mathbf{0.04 \pm 0.01}$ | $\mathbf{90.95 \pm 0.27}$ | $\mathbf{0.15 \pm 0.02}$ | $\mathbf{88.03 \pm 0.28}$ | $\mathbf{0.04 \pm 0.00}$ | $\mathbf{91.01 \pm 0.10}$ | $\mathbf{0.16 \pm 0.03}$ | $\mathbf{87.76 \pm 0.57}$ |

# A  More Experimental Results

## A.1  Loss Curves

Figure 6 and Figure 7 show the training loss curves of ResNet20 model.

## A.2  Tuning $\beta$ for Naive ASGDm

Following the suggestion in [23, 9], we conduct experiments to tune the momentum coefficient $\beta$ for naive ASGDm and present the results when training ResNet20 on CIFAR10 in Table 3. While tuning the momentum coefficient can enhance the performance of naive ASGDm, hyperparameter tuning is quite time-consuming and costly. In contrast, OrMo achieves better performance using the commonly used momentum value of 0.9, without requiring extensive tuning.

## A.3  Ablation Study

An ablation study is also conducted to justify the parameter update rule in line 13 of Algorithm 2. We replace the update rule in line 13 of Algorithm 2 with a vanilla SGD step, $\mathbf{w}_{t+1} = \mathbf{w}_{t+\frac{1}{2}} - \eta \mathbf{g}_{ite(k_t,t)}^{k_t}$, and name it OrMo (vanilla SGD step). The comparison between the experimental results of OrMo and OrMo (vanilla SGD step) are presented in Table 4 and Figure 8.

Table 4: Empirical results of OrMo and OrMo (vanilla SGD step) when training ResNet20 on CIFAR10 dataset.

| Number of Workers | 16 (hom.) | | 64 (hom.) | | 16 (het.) | | 64 (het.) | |
|---|---|---|---|---|---|---|---|---|
| Methods | Training Loss | Test Accuracy | Training Loss | Test Accuracy | Training Loss | Test Accuracy | Training Loss | Test Accuracy |
| OrMo (vanilla SGD step) | $0.07 \pm 0.02$ | $90.32 \pm 0.45$ | $0.27 \pm 0.07$ | $86.08 \pm 1.33$ | $0.07 \pm 0.01$ | $90.23 \pm 0.32$ | $0.27 \pm 0.07$ | $86.10 \pm 1.71$ |
| OrMo | $\mathbf{0.04 \pm 0.01}$ | $\mathbf{90.95 \pm 0.27}$ | $\mathbf{0.15 \pm 0.02}$ | $\mathbf{88.03 \pm 0.28}$ | $\mathbf{0.04 \pm 0.00}$ | $\mathbf{91.01 \pm 0.10}$ | $\mathbf{0.16 \pm 0.03}$ | $\mathbf{87.76 \pm 0.57}$ |

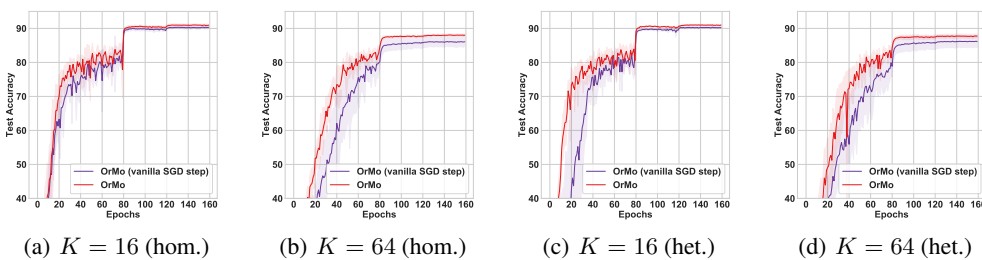

(a) $K = 16$ (hom.)  (b) $K = 64$ (hom.)  (c) $K = 16$ (het.)  (d) $K = 64$ (het.)

Figure 8: Test accuracy curves when training ResNet20 model on CIFAR10 dataset with different numbers of worker number.

Table 5: Test accuracy of different methods when training ResNet18 on CIFAR10 dataset.

| | homogeneous | heterogeneous |
|---|---|---|
| ASGD | 91.45 | 91.52 |
| naive ASGDm | 93.74 | 93.10 |
| shifted momentum | 94.02 | 94.20 |
| SMEGA$^2$ | 93.72 | 93.36 |
| OrMo | 94.32 | **94.26** |
| OrMo-DA | **94.50** | 94.03 |

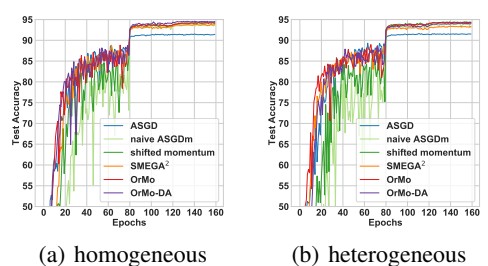

(a) homogeneous  (b) heterogeneous

Figure 9: Test accuracy curves when training ResNet18 model on CIFAR10 dataset.

## A.4 Experimental Results on ResNet18 Model

Table 5 and Figure 9 show the performance of different methods when training the ResNet18 model on the CIFAR10 dataset. The number of workers is set to 8. In the homogeneous setting, each worker has similar computing capabilities. In the heterogeneous setting, one worker is designated as the slow worker, whose average computation time is 10 times longer than that of the others.

## B  Algorithm Details

Algorithm 3 and Algorithm 4 show the details of SSGDm and naive ASGDm.

---
**Algorithm 3** SSGDm
---
1: **Server:**
2: **Input**: number of workers $K$, number of iterations $T$, learning rate $\eta$, momentum coefficient $\beta \in [0, 1)$;
3: **Initialization**: initial parameter $\mathbf{w}_0$, momentum $\mathbf{u}_0 = \mathbf{0}$, waiting set $\mathcal{C} = \emptyset$;
4: Send the initial parameter $\mathbf{w}_0$ to all workers;
5: **for** $t = 0$ **to** $T - 1$ **do**
6:     **if** the waiting set $\mathcal{C}$ is empty **then**
7:         $\mathbf{w}_{t+\frac{1}{2}} = \mathbf{w}_t - \beta\mathbf{u}_t$, $\mathbf{u}_{t+\frac{1}{2}} = \beta\mathbf{u}_t$;
8:     **else**
9:         $\mathbf{w}_{t+\frac{1}{2}} = \mathbf{w}_t$, $\mathbf{u}_{t+\frac{1}{2}} = \mathbf{u}_t$;
10:     **end if**
11:     Receive a stochastic gradient $\mathbf{g}^{k_t}_{ite(k_t,t)}$ from some worker $k_t$;
12:     Update the parameter $\mathbf{w}_{t+1} = \mathbf{w}_{t+\frac{1}{2}} - \eta\mathbf{g}^{k_t}_{ite(k_t,t)}$;
13:     Update the momentum $\mathbf{u}_{t+1} = \mathbf{u}_{t+\frac{1}{2}} + \eta\mathbf{g}^{k_t}_{ite(k_t,t)}$;
14:     Add the worker $k_t$ to the waiting set $\mathcal{C} = \mathcal{C} \cup \{k_t\}$;
15:     Execute the synchronous communication scheduler: only when all the workers are in the waiting set, i.e., $\mathcal{C} = [K]$, send the parameter $\mathbf{w}_{t+1}$ to the workers in $\mathcal{C}$ and set $\mathcal{C}$ to $\emptyset$;
16: **end for**
17: Notify all workers to stop;
18: **Worker** $k$ : $(k \in [K])$
19: **repeat**
20:     Wait until receiving the parameter $\mathbf{w}$ from the server;
21:     Randomly sample $\xi^k \sim \mathcal{D}$ and then compute the stochastic gradient $\mathbf{g}^k = \nabla f(\mathbf{w}; \xi^k)$;
22:     Send the stochastic gradient $\mathbf{g}^k$ to the server;
23: **until** receive server's notification to stop
---

---
**Algorithm 4** naive ASGDm
---
1: **Server:**
2: **Input**: number of workers $K$, number of iterations $T$, learning rate $\eta$, momentum coefficient $\beta \in [0, 1)$;
3: **Initialization**: initial parameter $\mathbf{w}_0$, momentum $\mathbf{u}_0 = \mathbf{0}$, waiting set $\mathcal{C} = \emptyset$;
4: Send the initial parameter $\mathbf{w}_0$ to all workers;
5: **for** $t = 0$ **to** $T - 1$ **do**
6:     Receive a stochastic gradient $\mathbf{g}^{k_t}_{ite(k_t,t)}$ from some worker $k_t$;
7:     Update the momentum $\mathbf{u}_{t+1} = \beta\mathbf{u}_t + \eta\mathbf{g}^{k_t}_{ite(k_t,t)}$
8:     Update the parameter $\mathbf{w}_{t+1} = \mathbf{w}_t - \mathbf{u}_{t+1}$
9:     Add the worker $k_t$ to the waiting set $\mathcal{C} = \mathcal{C} \cup \{k_t\}$;
10:     Execute the asynchronous communication scheduler: once the waiting set is not empty, i.e., $\mathcal{C} \neq \emptyset$, immediately send the parameter $\mathbf{w}_{t+1}$ to the worker in $\mathcal{C}$ and set $\mathcal{C}$ to $\emptyset$;
11: **end for**
12: Notify all workers to stop;
13: **Worker** $k$ : $(k \in [K])$
14: **repeat**
15:     Wait until receiving the parameter $\mathbf{w}$ from the server;
16:     Randomly sample $\xi^k \sim \mathcal{D}$ and then compute the stochastic gradient $\mathbf{g}^k = \nabla f(\mathbf{w}; \xi^k)$;
17:     Send the stochastic gradient $\mathbf{g}^k$ to the server;
18: **until** receive server's notification to stop
---

## C  Proof Details

### C.1  Reformulation of SSGDm

#### C.1.1  Proof of Proposition 1

*Proof.* It's easy to verify that $\{k_t, k_{t+1}, \cdots, k_{t+K-1}\} = [K]$ in (6), where $K \mid t$.

Base case: for s = 0, $\tilde{\mathbf{w}}_0 = \mathbf{w}_0$ and $\tilde{\mathbf{u}}_0 = \mathbf{u}_0$.

Inductive hypothesis: for some arbitrary integer $s' \geq 0$, assume that $\tilde{\mathbf{w}}_{s'} = \mathbf{w}_{s'K}, \tilde{\mathbf{u}}_{s'} = \mathbf{u}_{s'K}$.

Inductive step:

$$\tilde{\mathbf{u}}_{s'+1} = \beta\tilde{\mathbf{u}}_{s'} + \frac{\tilde{\eta}}{K} \sum_{k \in [K]} \nabla f(\tilde{\mathbf{w}}_{s'}; \xi^k) = \beta\mathbf{u}_{s'K} + \eta \sum_{k \in [K]} \nabla f(\mathbf{w}_{s'K}; \xi^k) = \mathbf{u}_{(s'+1)K},$$

$$\tilde{\mathbf{w}}_{s'+1} = \tilde{\mathbf{w}}_{s'} - \beta\tilde{\mathbf{u}}_{s'} - \frac{\tilde{\eta}}{K} \sum_{k \in [K]} \nabla f(\tilde{\mathbf{w}}_{s'}; \xi^k) = \mathbf{w}_{s'K} - \beta\mathbf{u}_{s'K} - \eta \sum_{k \in [K]} \nabla f(\mathbf{w}_{s'K}; \xi^k)$$

$$= \mathbf{w}_{(s'+1)K}.$$

We can conclude that $\tilde{\mathbf{w}}_s = \mathbf{w}_{sK}$ and $\tilde{\mathbf{u}}_s = \mathbf{u}_{sK}$ for any $s \in [S]$. $\qquad\square$

### C.2  OrMo

#### C.2.1  Proof of Lemma 1

Lemma 1 can be viewed as a special case of Lemma 5, and its proof is completed by substituting each delay-adaptive learning rate $\hat{\eta}_{k,t}$ in the proof of Lemma 5 with $\eta$ for all $k \in [K]$ and $t \in [T]$.

#### C.2.2  Proof of Lemma 2

Lemma 2 can be viewed as a special case of Lemma 6, and its proof is completed by substituting each delay-adaptive learning rate $\hat{\eta}_{k,t}$ in the proof of Lemma 6 with $\eta$ for all $k \in [K]$ and $t \in [T]$.

#### C.2.3  Proof of Lemma 3

Lemma 3 can be viewed as a special case of Lemma 7, and its proof is completed by substituting each delay-adaptive learning rate $\hat{\eta}_{k,t}$ in the proof of Lemma 7 with $\eta$ for all $k \in [K]$ and $t \in [T]$.

**Lemma 4.** *With Assumption 2, the gap between $\hat{\mathbf{y}}_t$ and $\hat{\mathbf{w}}_t$ can be bounded:*

$$\mathbb{E}\|\hat{\mathbf{y}}_t - \hat{\mathbf{w}}_t\|^2 \leq \frac{\beta^2\eta^2 K^2 G^2}{(1-\beta)^4}, \forall t \geq 1. \qquad (12)$$

*Proof.* For any $t \geq 1$, $\hat{\mathbf{u}}_t$ can be formulated as follows:

$$\hat{\mathbf{u}}_t = \beta^{\lfloor\frac{t+K-2}{K}\rfloor}\left(\sum_{k \in [K]} \eta\mathbf{g}_0^k\right) + \sum_{s=1}^{\lfloor\frac{t+K-2}{K}\rfloor} \beta^{\lfloor\frac{t+K-2}{K}\rfloor-s}\left(\sum_{j=(s-1)K+1}^{\min\{sK, t-1\}} \eta\mathbf{g}_j^{k_{j-1}}\right).$$

$$\|\hat{\mathbf{y}}_t - \hat{\mathbf{w}}_t\|^2 = \frac{\beta^2}{(1-\beta)^2}\|\hat{\mathbf{u}}_t\|^2$$

$$= \frac{\beta^2}{(1-\beta)^2}\left\|\beta^{\lfloor\frac{t+K-2}{K}\rfloor}\left(\sum_{k \in [K]} \eta\mathbf{g}_0^k\right) + \sum_{s=1}^{\lfloor\frac{t+K-2}{K}\rfloor} \beta^{\lfloor\frac{t+K-2}{K}\rfloor-s}\left(\sum_{j=(s-1)K+1}^{\min\{sK, t-1\}} \eta\mathbf{g}_j^{k_{j-1}}\right)\right\|^2$$

Let $q_t = \sum_{k \in [K]} \beta^{\lfloor \frac{t+K-2}{K} \rfloor} + \sum_{s=1}^{\lfloor \frac{t+K-2}{K} \rfloor} \sum_{j=(s-1)K+1}^{\min\{sK,t-1\}} \beta^{\lfloor \frac{t+K-2}{K} \rfloor - s}$, then we have

$$\|\hat{\mathbf{y}}_t - \hat{\mathbf{w}}_t\|^2 = \frac{\beta^2}{(1-\beta)^2} \left\| \beta^{\lfloor \frac{t+K-2}{K} \rfloor} \left( \sum_{k \in [K]} \eta \mathbf{g}_0^k \right) + \sum_{s=1}^{\lfloor \frac{t+K-2}{K} \rfloor} \beta^{\lfloor \frac{t+K-2}{K} \rfloor - s} \left( \sum_{j=(s-1)K+1}^{\min\{sK,t-1\}} \eta \mathbf{g}_j^{k_{j-1}} \right) \right\|^2$$

$$= \frac{\beta^2 q_t^2}{(1-\beta)^2} \left\| \sum_{k \in [K]} \frac{\beta^{\lfloor \frac{t+K-2}{K} \rfloor}}{q_t} \eta \mathbf{g}_0^k + \sum_{s=1}^{\lfloor \frac{t+K-2}{K} \rfloor} \sum_{j=(s-1)K+1}^{\min\{sK,t-1\}} \frac{\beta^{\lfloor \frac{t+K-2}{K} \rfloor - s}}{q_t} \eta \mathbf{g}_j^{k_{j-1}} \right\|^2$$

$$\leq \frac{\beta^2 q_t}{(1-\beta)^2} \left( \sum_{k \in [K]} \beta^{\lfloor \frac{t+K-2}{K} \rfloor} \left\| \eta \mathbf{g}_0^k \right\|^2 + \sum_{s=1}^{\lfloor \frac{t+K-2}{K} \rfloor} \sum_{j=(s-1)K+1}^{\min\{sK,t-1\}} \beta^{\lfloor \frac{t+K-2}{K} \rfloor - s} \left\| \eta \mathbf{g}_j^{k_{j-1}} \right\|^2 \right).$$

$$\mathbb{E}\|\hat{\mathbf{y}}_t - \hat{\mathbf{w}}_t\|^2 \leq \frac{\beta^2 q_t}{(1-\beta)^2} \left( \sum_{k \in [K]} \beta^{\lfloor \frac{t+K-2}{K} \rfloor} \mathbb{E}\|\eta \mathbf{g}_0^k\|^2 + \sum_{s=1}^{\lfloor \frac{t+K-2}{K} \rfloor} \sum_{j=(s-1)K+1}^{\min\{sK,t-1\}} \beta^{\lfloor \frac{t+K-2}{K} \rfloor - s} \mathbb{E}\|\eta \mathbf{g}_j^{k_{j-1}}\|^2 \right)$$

$$\leq \frac{\beta^2 \eta^2 G^2 q_t}{(1-\beta)^2} \left( \sum_{k \in [K]} \beta^{\lfloor \frac{t+K-2}{K} \rfloor} + \sum_{s=1}^{\lfloor \frac{t+K-2}{K} \rfloor} \sum_{j=(s-1)K+1}^{\min\{sK,t-1\}} \beta^{\lfloor \frac{t+K-2}{K} \rfloor - s} \right)$$

$$= \frac{\beta^2 \eta^2 G^2 q_t^2}{(1-\beta)^2} \leq \frac{\beta^2 \eta^2 K^2 G^2}{(1-\beta)^4}$$

$\square$

### C.2.4 Proof of Theorem 1

*Proof.*

$$\mathbb{E}F(\hat{\mathbf{y}}_{t+1}) \leq F(\hat{\mathbf{y}}_t) - \frac{\eta}{1-\beta} \langle \nabla F(\hat{\mathbf{y}}_t), \mathbb{E}\mathbf{g}_t^{k_{t-1}} \rangle + \frac{L\eta^2}{2(1-\beta)^2} \mathbb{E}\|\mathbf{g}_t^{k_{t-1}}\|^2$$

$$\leq F(\hat{\mathbf{y}}_t) - \frac{\eta}{1-\beta} \langle \nabla F(\hat{\mathbf{y}}_t), \nabla F(\mathbf{w}_t) \rangle + \frac{L\eta^2}{2(1-\beta)^2} \mathbb{E}\|\mathbf{g}_t^{k_{t-1}} - \nabla F(\mathbf{w}_t)\|^2$$

$$+ \frac{L\eta^2}{2(1-\beta)^2} \|\nabla F(\mathbf{w}_t)\|^2$$

$$\leq F(\hat{\mathbf{y}}_t) - \frac{\eta}{1-\beta} \langle \nabla F(\hat{\mathbf{y}}_t), \nabla F(\mathbf{w}_t) \rangle + \frac{L\eta^2 \sigma^2}{2(1-\beta)^2} + \frac{L\eta^2}{2(1-\beta)^2} \|\nabla F(\mathbf{w}_t)\|^2$$

$$-\langle \nabla F(\hat{\mathbf{y}}_t), \nabla F(\mathbf{w}_t) \rangle = -\langle \nabla F(\hat{\mathbf{y}}_t) - \nabla F(\mathbf{w}_t) + \nabla F(\mathbf{w}_t), \nabla F(\mathbf{w}_t) \rangle$$

$$= -\langle \nabla F(\hat{\mathbf{y}}_t) - \nabla F(\mathbf{w}_t), \nabla F(\mathbf{w}_t) \rangle - \|\nabla F(\mathbf{w}_t)\|^2$$

$$\leq \frac{1}{2} \|\nabla F(\hat{\mathbf{y}}_t) - \nabla F(\mathbf{w}_t)\|^2 + \frac{1}{2} \|\nabla F(\mathbf{w}_t)\|^2 - \|\nabla F(\mathbf{w}_t)\|^2$$

$$\leq \frac{L^2}{2} \|\hat{\mathbf{y}}_t - \mathbf{w}_t\|^2 - \frac{1}{2} \|\nabla F(\mathbf{w}_t)\|^2$$

$$\mathbb{E}\|\hat{\mathbf{y}}_t - \mathbf{w}_t\|^2 \leq 2\mathbb{E}\|\hat{\mathbf{y}}_t - \hat{\mathbf{w}}_t\|^2 + 2\mathbb{E}\|\hat{\mathbf{w}}_t - \mathbf{w}_t\|^2 \leq \frac{2\beta^2 \eta^2 K^2 G^2}{(1-\beta)^4} + \frac{2\eta^2 K^2 G^2}{(1-\beta)^2} \leq \frac{2\eta^2 K^2 G^2}{(1-\beta)^4}$$

$$\mathbb{E}F(\hat{\mathbf{y}}_{t+1}) \leq \mathbb{E}F(\hat{\mathbf{y}}_t) - \frac{\eta}{1-\beta} \mathbb{E}\langle \nabla F(\hat{\mathbf{y}}_t), \nabla F(\mathbf{w}_t) \rangle + \frac{L\eta^2 \sigma^2}{2(1-\beta)^2} + \frac{L\eta^2}{2(1-\beta)^2} \mathbb{E}\|\nabla F(\mathbf{w}_t)\|^2$$

$$\leq \mathbb{E}F(\hat{\mathbf{y}}_t) + \frac{\eta L^2}{2(1-\beta)} \mathbb{E}\|\hat{\mathbf{y}}_t - \mathbf{w}_t\|^2 + \left( \frac{L\eta^2}{2(1-\beta)^2} - \frac{\eta}{2(1-\beta)} \right) \mathbb{E}\|\nabla F(\mathbf{w}_t)\|^2 + \frac{L\eta^2 \sigma^2}{2(1-\beta)^2}$$

$$\overset{\eta \leq \frac{1-\beta}{2KL}}{\leq} \mathbb{E}F(\hat{\mathbf{y}}_t) + \frac{\eta L^2}{2(1-\beta)} \mathbb{E}\|\hat{\mathbf{y}}_t - \mathbf{w}_t\|^2 - \frac{\eta}{4(1-\beta)} \mathbb{E}\|\nabla F(\mathbf{w}_t)\|^2 + \frac{L\eta^2 \sigma^2}{2(1-\beta)^2}$$

$$\leq \mathbb{E}F(\hat{\mathbf{y}}_t) - \frac{\eta}{4(1-\beta)} \mathbb{E}\|\nabla F(\mathbf{w}_t)\|^2 + \frac{L\eta^2 \sigma^2}{2(1-\beta)^2} + \frac{\eta^3 K^2 G^2 L^2}{(1-\beta)^5}$$

Summing up the above equation from $t = 1$ to $t = T$, we can get that

$$\frac{1}{T} \sum_{t=1}^{T} \mathbb{E} \|\nabla F(\mathbf{w}_t)\|^2 \leq \frac{4(1-\beta) \left[\mathbb{E}F(\hat{\mathbf{y}}_1) - F^*\right]}{T\eta} + \frac{2L\eta\sigma^2}{1-\beta} + \frac{4\eta^2 K^2 G^2 L^2}{(1-\beta)^4}.$$

Since $\hat{\mathbf{y}}_1 - \mathbf{w}_0 = \frac{1}{1-\beta}(\hat{\mathbf{w}}_1 - \mathbf{w}_0) = -\frac{\eta}{1-\beta} \sum_{k \in [K]} \mathbf{g}_0^k$, we have

$$\mathbb{E}F(\hat{\mathbf{y}}_1) \leq F(\mathbf{w}_0) - \frac{\eta}{1-\beta} \mathbb{E}\langle \nabla F(\mathbf{w}_0), \sum_{k \in [K]} \mathbf{g}_0^k \rangle + \frac{L\eta^2}{2(1-\beta)^2} \mathbb{E} \left\| \sum_{k \in [K]} \mathbf{g}_0^k \right\|^2$$

$$\leq F(\mathbf{w}_0) - \frac{K\eta}{1-\beta} \|\nabla F(\mathbf{w}_0)\|^2 + \frac{L\eta^2}{2(1-\beta)^2} \mathbb{E} \left\| \sum_{k \in [K]} \mathbf{g}_0^k \right\|^2$$

$$\leq F(\mathbf{w}_0) - \frac{K\eta}{1-\beta} \|\nabla F(\mathbf{w}_0)\|^2 + \frac{L\eta^2}{2(1-\beta)^2} \mathbb{E} \left\| \sum_{k \in [K]} \mathbf{g}_0^k - K\nabla F(\mathbf{w}_0) + K\nabla F(\mathbf{w}_0) \right\|^2$$

$$\leq F(\mathbf{w}_0) + \left( \frac{LK^2\eta^2}{2(1-\beta)^2} - \frac{K\eta}{1-\beta} \right) \|\nabla F(\mathbf{w}_0)\|^2 + \frac{KL\sigma^2\eta^2}{2(1-\beta)^2}$$

$$\leq F(\mathbf{w}_0) + \frac{KL\sigma^2\eta^2}{2(1-\beta)^2}.$$

The last inequality above holds because $\eta \leq \frac{1-\beta}{2KL}$.

Combining the above equations, we can get that

$$\frac{1}{T} \sum_{t=1}^{T} \mathbb{E} \|\nabla F(\mathbf{w}_t)\|^2 \leq \frac{4(1-\beta)\left[F(\mathbf{w}_0) - F^*\right]}{T\eta} + \frac{2L\eta\sigma^2}{1-\beta} + \frac{2LK\sigma^2\eta}{(1-\beta)T} + \frac{4\eta^2 L^2 K^2 G^2}{(1-\beta)^4}$$

$$\overset{T \geq K}{\leq} \frac{4(1-\beta)\left[F(\mathbf{w}_0) - F^*\right]}{T\eta} + \frac{4L\eta\sigma^2}{1-\beta} + \frac{4\eta^2 L^2 K^2 G^2}{(1-\beta)^4}.$$

Let $\eta = \min\{\frac{1-\beta}{2KL}, \frac{(1-\beta)[F(\mathbf{w}_0)-F^*]^{\frac{1}{2}}}{(LT)^{\frac{1}{2}}\sigma}, \frac{(1-\beta)^{\frac{5}{3}}[F(\mathbf{w}_0)-F^*]^{\frac{1}{3}}}{(LKG)^{\frac{2}{3}}T^{\frac{1}{3}}}\}$, then we can get that

$$\frac{1}{T} \sum_{t=1}^{T} \mathbb{E}\|\nabla F(\mathbf{w}_t)\|^2 \leq \mathcal{O}\left( \sqrt{\frac{L\sigma^2}{T}} + \left(\frac{KLG}{T}\right)^{\frac{2}{3}} + \frac{KL}{T} \right).$$

$\square$

## C.3 OrMo with Delay-Adaptive Learning Rate

### C.3.1 Algorithm

The details of OrMo with delay-adaptive learning rate (OrMo-DA) are presented in Algorithm 5.

### C.3.2 Notation

For a positive integer $n$, $[n] = \{0, 1, 2, \cdots, n-1\}$. $[0]$ is defined as $\emptyset$.

The function $ite(k, t)$ denotes the iteration index of the latest parameter sent to worker $k$ before iteration $t$, which can be formulated as

$$ite(k, t) = \begin{cases} 0 & t = 0, k \in [K], \\ t & t > 0, k = k_{t-1}, \\ ite(k, t-1) & t > 0, k \neq k_{t-1}, \end{cases}$$

where $k \in [K], t \in [T+1]$.

**Algorithm 5** OrMo-DA

1: **Server:**
2: **Input**: number of workers $K$, number of iterations $T$, momentum coefficient $\beta \in [0,1)$;
3: **Initialization**: initial parameter $\mathbf{w}_0$, momentum $\mathbf{u}_0 = \mathbf{0}$, index of the latest gradient group $I_0 = 0$, waiting set $\mathcal{C} = \emptyset$;
4: Send the initial parameter $\mathbf{w}_0$ and its iteration index 0 to all workers;
5: **for** $t = 0$ **to** $T-1$ **do**
6:     **if** the waiting set $\mathcal{C}$ is empty and $\lceil \frac{t}{K} \rceil > I_t$ **then**
7:         $\mathbf{w}_{t+\frac{1}{2}} = \mathbf{w}_t - \beta\mathbf{u}_t$, $\mathbf{u}_{t+\frac{1}{2}} = \beta\mathbf{u}_t$, $I_{t+1} = I_t + 1$;
8:     **else**
9:         $\mathbf{w}_{t+\frac{1}{2}} = \mathbf{w}_t$, $\mathbf{u}_{t+\frac{1}{2}} = \mathbf{u}_t$, $I_{t+1} = I_t$;
10:     **end if**
11:     Receive a stochastic gradient $\mathbf{g}_{ite(k_t,t)}^{k_t}$ and its iteration index $ite(k_t,t)$ from some worker $k_t$ and then calculate $\lceil \frac{ite(k_t,t)}{K} \rceil$ (i.e., the index of the gradient group that $\mathbf{g}_{ite(k_t,t)}^{k_t}$ belongs to);
12:     Update the momentum $\mathbf{u}_{t+1} = \mathbf{u}_{t+\frac{1}{2}} + \beta^{I_{t+1} - \lceil \frac{ite(k_t,t)}{K} \rceil} \times \left( \eta_t \mathbf{g}_{ite(k_t,t)}^{k_t} \right)$;
13:     Update the parameter $\mathbf{w}_{t+1} = \mathbf{w}_{t+\frac{1}{2}} - \frac{1 - \beta^{I_{t+1} - \lceil \frac{ite(k_t,t)}{K} \rceil + 1}}{1 - \beta} \times \left( \eta_t \mathbf{g}_{ite(k_t,t)}^{k_t} \right)$;
14:     Add the worker $k_t$ to the waiting set $\mathcal{C} = \mathcal{C} \cup \{k_t\}$;
15:     Execute the asynchronous communication scheduler: once the waiting set is not empty, i.e., $\mathcal{C} \neq \emptyset$, immediately send the parameter $\mathbf{w}_{t+1}$ and its iteration index $t+1$ to the worker in $\mathcal{C}$ and set $\mathcal{C}$ to $\emptyset$;
16: **end for**
17: Notify all workers to stop;
18: **Worker** $k : (k \in [K])$
19: **repeat**
20:     Wait until receiving the parameter $\mathbf{w}_{t'}$ and its iteration index $t'$ from the server;
21:     Randomly sample $\xi^k \sim \mathcal{D}$ and then compute the stochastic gradient $\mathbf{g}_{t'}^k = \nabla f(\mathbf{w}_{t'}; \xi^k)$;
22:     Send the stochastic gradient $\mathbf{g}_{t'}^k$ and its iteration index $t'$ to the server;
23: **until** receive server's notification to stop

$\eta_t$ is the learning rate at iteration $t$ and satisfies that

$$
\eta_t = \begin{cases} \eta & \tau_t \leq 2K, \\ \min\{\eta, \dfrac{1}{4L\tau_t}\} & \tau_t > 2K, \end{cases}
$$

where $t \in [T]$.

The function $next(k,t)$ denotes the index of the next iteration that the gradient from worker $k$ will participate in the parameter update after iteration $t$ (including iteration $t$), which can be formulated as

$$
next(k,t) = \begin{cases} \min\{j \geq t : k_j = k\} & \exists j \in [T] \setminus [t], k_j = k, \\ T & \forall j \in [T] \setminus [t], k_j \neq k, \end{cases}
$$

where $k \in [K]$, $t \in [T]$. It's easy to verify that $ite(k, next(k,t)) = ite(k,t)$, $next(k, ite(k,t)) = next(k,t)$, where $k \in [K], t \in [T]$.

We define $\hat{\tau}_{k,t} = t - ite(k,t)$, where $k \in [K]$ and $t \in [T+1]$. $\hat{\tau}_{k,t}$ denotes the current delay of the gradient $\mathbf{g}_{ite(k,t)}^k$ at iteration $t$, which is the number of iterations that have happened since $ite(k,t)$. It's easy to verify that $\hat{\tau}_{k_t,t} = t - ite(k_t,t) = \tau_t$, where $t \in [T]$.

We also define an auxiliary sequence $\hat{\eta}_{k,t}$ for the adaptive learning rates. $\hat{\eta}_{k,t}$ denotes the learning rate corresponding to the gradient $\mathbf{g}_{ite(k,t)}^k$.

$$
\hat{\eta}_{k,t} = \begin{cases} \eta & \hat{\tau}_{k,next(k,t)} \leq 2K, \\ \min\{\eta, \dfrac{1}{4L\hat{\tau}_{k,next(k,t)}}\} & \hat{\tau}_{k,next(k,t)} > 2K, \end{cases}
$$

where $k \in [K]$ and $t \in [T]$. Since $next(k,t) = next(k, ite(k,t))$, we can have that $\hat{\eta}_{k,t} = \hat{\eta}_{k,ite(k,t)}$, where $k \in [K], t \in [T]$.

If $next(k,t) \in [T]$, $\hat{\tau}_{k,next(k,t)} = \hat{\tau}_{k_{next(k,t)},next(k,t)} = \tau_{next(k,t)}$ and then we have that

$$\hat{\eta}_{k,t} = \eta_{next(k,t)} = \begin{cases} \eta & \tau_{next(k,t)} \le 2K, \\ \min\{\eta, \dfrac{1}{4L\tau_{next(k,t)}}\} & \tau_{next(k,t)} > 2K. \end{cases}$$

It's easy to verify that $\hat{\eta}_{k_t,t} = \eta_{next(k_t,t)} = \eta_t$, where $t \in [T]$.

If $next(k,t) = T$,

$$\hat{\eta}_{k,t} = \begin{cases} \eta & \hat{\tau}_{k,T} \le 2K, \\ \min\{\eta, \dfrac{1}{4L\hat{\tau}_{k,T}}\} & \hat{\tau}_{k,T} > 2K. \end{cases}$$

### C.3.3 Convergence Analysis for OrMo-DA

Firstly, we define one auxiliary sequence $\{\hat{\mathbf{u}}_t\}_{t \ge 1}$ for the momentum: $\hat{\mathbf{u}}_1 = \sum_{k \in [K]} \hat{\eta}_{k,0} \mathbf{g}_0^k$, and

$$\hat{\mathbf{u}}_{t+1} = \begin{cases} \beta \hat{\mathbf{u}}_t + \hat{\eta}_{k_{t-1},t} \mathbf{g}_t^{k_{t-1}} & K \mid (t-1), \\ \hat{\mathbf{u}}_t + \hat{\eta}_{k_{t-1},t} \mathbf{g}_t^{k_{t-1}} & K \nmid (t-1), \end{cases} \tag{13}$$

for $t \ge 1$.

**Lemma 5.** *For any $t \ge 0$, the gap between $\mathbf{u}_{t+1}$ and $\hat{\mathbf{u}}_{t+1}$ can be formulated as follows:*

$$\hat{\mathbf{u}}_{t+1} - \mathbf{u}_{t+1} = \sum_{k \in [K], k \ne k_t} \beta^{\lceil \frac{t}{K} \rceil - \lceil \frac{ite(k,t)}{K} \rceil} \hat{\eta}_{k,ite(k,t)} \mathbf{g}_{ite(k,t)}^k. \tag{14}$$

*Proof.* Base case: for $t = 0$, $\hat{\mathbf{u}}_1 = \sum_{k \in [K]} \hat{\eta}_{k,0} \mathbf{g}_0^k$, $\mathbf{u}_1 = \hat{\eta}_{k_0,0} \mathbf{g}_0^{k_0}$, then we have

$$\begin{aligned} \hat{\mathbf{u}}_1 - \mathbf{u}_1 &= \sum_{k \in [K], k \ne k_0} \beta^{\lceil \frac{0}{K} \rceil - \lceil \frac{0}{K} \rceil} \hat{\eta}_{k,0} \mathbf{g}_0^k \\ &= \sum_{k \in [K], k \ne k_0} \beta^{\lceil \frac{0}{K} \rceil - \lceil \frac{ite(k,0)}{K} \rceil} \hat{\eta}_{k,ite(k,0)} \mathbf{g}_{ite(k,0)}^k. \end{aligned}$$

Inductive hypothesis: for some arbitrary integer $t' - 1 \ge 0$, assume that (14) is true for $t = t' - 1$.

Inductive step: We will prove that (14) is true for $t = t'$. Firstly, we divide our discussion into two cases based on whether $t' - 1$ is divisible by $K$ and prove that

$$\begin{aligned} \hat{\mathbf{u}}_{t'+1} - \mathbf{u}_{t'+1} &= \sum_{k \in [K], k \ne k_{t'-1}} \beta^{\lceil \frac{t'}{K} \rceil - \lceil \frac{ite(k,t')}{K} \rceil} \hat{\eta}_{k,ite(k,t')} \mathbf{g}_{ite(k,t')}^k + \hat{\eta}_{k_{t'-1},t'} \mathbf{g}_{t'}^{k_{t'-1}} \\ &\quad - \beta^{\lceil \frac{t'}{K} \rceil - \lceil \frac{ite(k_{t'},t')}{K} \rceil} \hat{\eta}_{k_{t'},ite(k_{t'},t')} \mathbf{g}_{ite(k_{t'},t')}^{k_{t'}}. \end{aligned}$$

Case 1: $K \mid (t'-1)$

$$\hat{\mathbf{u}}_{t'+1} = \beta \hat{\mathbf{u}}_{t'} + \hat{\eta}_{k_{t'-1},t'} \mathbf{g}_{t'}^{k_{t'-1}},$$

$$\mathbf{u}_{t'+1} = \beta \mathbf{u}_{t'} + \beta^{\lceil \frac{t'}{K} \rceil - \lceil \frac{ite(k_{t'},t')}{K} \rceil} \hat{\eta}_{k_{t'},ite(k_{t'},t')} \mathbf{g}_{ite(k_{t'},t')}^{k_{t'}},$$

$$\hat{\mathbf{u}}_{t'+1} - \mathbf{u}_{t'+1} = \beta(\hat{\mathbf{u}}_{t'} - \mathbf{u}_{t'}) + \hat{\eta}_{k_{t'-1},t'} \mathbf{g}_{t'}^{k_{t'-1}} - \beta^{\lceil \frac{t'}{K} \rceil - \lceil \frac{ite(k_{t'},t')}{K} \rceil} \hat{\eta}_{k_{t'},ite(k_{t'},t')} \mathbf{g}_{ite(k_{t'},t')}^{k_{t'}}$$

$$= \sum_{k \in [K], k \neq k_{t'-1}} \beta^{\lceil \frac{t'-1}{K} \rceil + 1 - \lceil \frac{ite(k,t'-1)}{K} \rceil} \hat{\eta}_{k,ite(k,t'-1)} \mathbf{g}_{ite(k,t'-1)}^{k}$$

$$+ \hat{\eta}_{k_{t'-1},t'} \mathbf{g}_{t'}^{k_{t'-1}} - \beta^{\lceil \frac{t'}{K} \rceil - \lceil \frac{ite(k_{t'},t')}{K} \rceil} \hat{\eta}_{k_{t'},ite(k_{t'},t')} \mathbf{g}_{ite(k_{t'},t')}^{k_{t'}}$$

$$= \sum_{k \in [K], k \neq k_{t'-1}} \beta^{\lceil \frac{t'}{K} \rceil - \lceil \frac{ite(k,t'-1)}{K} \rceil} \hat{\eta}_{k,ite(k,t'-1)} \mathbf{g}_{ite(k,t'-1)}^{k}$$

$$+ \hat{\eta}_{k_{t'-1},t'} \mathbf{g}_{t'}^{k_{t'-1}} - \beta^{\lceil \frac{t'}{K} \rceil - \lceil \frac{ite(k_{t'},t')}{K} \rceil} \hat{\eta}_{k_{t'},ite(k_{t'},t')} \mathbf{g}_{ite(k_{t'},t')}^{k_{t'}}.$$

The second equation above holds because $\lceil \frac{t'}{K} \rceil > \lceil \frac{t'-1}{K} \rceil = I_{t'}$ when $K \mid (t'-1)$, $I_{t'+1} = \lceil \frac{t'}{K} \rceil$ and $\hat{\eta}_{k_{t'},ite(k_{t'},t')} = \eta_{next(k_{t'},ite(k_{t'},t'))} = \eta_{t'}$. The last equation above holds because $\lceil \frac{t'}{K} \rceil = \lceil \frac{t'-1}{K} \rceil + 1$.

Case 2: $K \nmid (t'-1)$

$$\hat{\mathbf{u}}_{t'+1} = \hat{\mathbf{u}}_{t'} + \hat{\eta}_{k_{t'-1},t'} \mathbf{g}_{t'}^{k_{t'-1}},$$

$$\mathbf{u}_{t'+1} = \mathbf{u}_{t'} + \beta^{\lceil \frac{t'}{K} \rceil - \lceil \frac{ite(k_{t'},t')}{K} \rceil} \hat{\eta}_{k_{t'},ite(k_{t'},t')} \mathbf{g}_{ite(k_{t'},t')}^{k_{t'}},$$

$$\hat{\mathbf{u}}_{t'+1} - \mathbf{u}_{t'+1} = (\hat{\mathbf{u}}_{t'} - \mathbf{u}_{t'}) + \hat{\eta}_{k_{t'-1},t'} \mathbf{g}_{t'}^{k_{t'-1}} - \beta^{\lceil \frac{t'}{K} \rceil - \lceil \frac{ite(k_{t'},t')}{K} \rceil} \hat{\eta}_{k_{t'},ite(k_{t'},t')} \mathbf{g}_{ite(k_{t'},t')}^{k_{t'}}$$

$$= \sum_{k \in [K], k \neq k_{t'-1}} \beta^{\lceil \frac{t'-1}{K} \rceil - \lceil \frac{ite(k,t'-1)}{K} \rceil} \hat{\eta}_{k,ite(k,t'-1)} \mathbf{g}_{ite(k,t'-1)}^{k}$$

$$+ \hat{\eta}_{k_{t'-1},t'} \mathbf{g}_{t'}^{k_{t'-1}} - \beta^{\lceil \frac{t'}{K} \rceil - \lceil \frac{ite(k_{t'},t')}{K} \rceil} \hat{\eta}_{k_{t'},ite(k_{t'},t')} \mathbf{g}_{ite(k_{t'},t')}^{k_{t'}}$$

$$= \sum_{k \in [K], k \neq k_{t'-1}} \beta^{\lceil \frac{t'}{K} \rceil - \lceil \frac{ite(k,t'-1)}{K} \rceil} \hat{\eta}_{k,ite(k,t'-1)} \mathbf{g}_{ite(k,t'-1)}^{k}$$

$$+ \hat{\eta}_{k_{t'-1},t'} \mathbf{g}_{t'}^{k_{t'-1}} - \beta^{\lceil \frac{t'}{K} \rceil - \lceil \frac{ite(k_{t'},t')}{K} \rceil} \hat{\eta}_{k_{t'},ite(k_{t'},t')} \mathbf{g}_{ite(k_{t'},t')}^{k_{t'}}.$$

The last equation above holds because $\lceil \frac{t'}{K} \rceil = \lceil \frac{t'-1}{K} \rceil$.

Since $ite(k,t') = ite(k,t'-1), \forall k \neq k_{t'-1}$, we can get the following equation for both cases above:

$$\hat{\mathbf{u}}_{t'+1} - \mathbf{u}_{t'+1} = \sum_{k \in [K], k \neq k_{t'-1}} \beta^{\lceil \frac{t'}{K} \rceil - \lceil \frac{ite(k,t')}{K} \rceil} \hat{\eta}_{k,ite(k,t')} \mathbf{g}_{ite(k,t')}^{k} + \hat{\eta}_{k_{t'-1},t'} \mathbf{g}_{t'}^{k_{t'-1}}$$

$$- \beta^{\lceil \frac{t'}{K} \rceil - \lceil \frac{ite(k_{t'},t')}{K} \rceil} \hat{\eta}_{k_{t'},ite(k_{t'},t')} \mathbf{g}_{ite(k_{t'},t')}^{k_{t'}}.$$

If $k_{t'} = k_{t'-1}$, then we have $ite(k_{t'},t') = t'$ and

$$\hat{\mathbf{u}}_{t'+1} - \mathbf{u}_{t'+1} = \sum_{k \in [K], k \neq k_{t'}} \beta^{\lceil \frac{t'}{K} \rceil - \lceil \frac{ite(k,t')}{K} \rceil} \hat{\eta}_{k,ite(k,t')} \mathbf{g}_{ite(k,t')}^{k}.$$

If $k_{t'} \neq k_{t'-1}$, then we have

$$
\begin{aligned}
\hat{\mathbf{u}}_{t'+1} - \mathbf{u}_{t'+1} &= \sum_{k\in[K], k\neq k_{t'-1}} \beta^{\lceil \frac{t'}{K}\rceil - \lceil \frac{ite(k,t')}{K}\rceil} \hat{\eta}_{k,ite(k,t')} \mathbf{g}^k_{ite(k,t')} + \hat{\eta}_{k_{t'-1},t'} \mathbf{g}^{k_{t'-1}}_{t'} \\
&\quad - \beta^{\lceil \frac{t'}{K}\rceil - \lceil \frac{ite(k_{t'},t')}{K}\rceil} \hat{\eta}_{k_{t'},ite(k_{t'},t')} \mathbf{g}^{k_{t'}}_{ite(k_{t'},t')} \\
&= \sum_{k\in[K], k\neq k_{t'-1}, k\neq k_{t'}} \beta^{\lceil \frac{t'}{K}\rceil - \lceil \frac{ite(k,t')}{K}\rceil} \hat{\eta}_{k,ite(k,t')} \mathbf{g}^k_{ite(k,t')} + \hat{\eta}_{k_{t'-1},t'} \mathbf{g}^{k_{t'-1}}_{t'} \\
&= \sum_{k\in[K], k\neq k_{t'-1}, k\neq k_{t'}} \beta^{\lceil \frac{t'}{K}\rceil - \lceil \frac{ite(k,t')}{K}\rceil} \hat{\eta}_{k,ite(k,t')} \mathbf{g}^k_{ite(k,t')} \\
&\quad + \beta^{\lceil \frac{t'}{K}\rceil - \lceil \frac{ite(k_{t'-1},t')}{K}\rceil} \hat{\eta}_{k_{t'-1},ite(k_{t'-1},t')} \mathbf{g}^{k_{t'-1}}_{ite(k_{t'-1},t')} \\
&= \sum_{k\in[K], k\neq k_{t'}} \beta^{\lceil \frac{t'}{K}\rceil - \lceil \frac{ite(k,t')}{K}\rceil} \hat{\eta}_{k,ite(k,t')} \mathbf{g}^k_{ite(k,t')}.
\end{aligned}
$$

We can conclude that $\hat{\mathbf{u}}_{t+1} - \mathbf{u}_{t+1} = \sum_{k\in[K], k\neq k_t} \beta^{\lceil \frac{t}{K}\rceil - \lceil \frac{ite(k,t)}{K}\rceil} \hat{\eta}_{k,ite(k,t)} \mathbf{g}^k_{ite(k,t)}$ is true for any $t \geq 0$. $\qquad\square$

Then, we define one auxiliary sequence $\{\hat{\mathbf{w}}_t\}_{t\geq 1}$ for the parameter: $\hat{\mathbf{w}}_1 = \mathbf{w}_0 - \sum_{k\in[K]} \hat{\eta}_{k,0} \mathbf{g}^k_0$, and

$$
\hat{\mathbf{w}}_{t+1} = \begin{cases} \hat{\mathbf{w}}_t - \beta \hat{\mathbf{u}}_t - \hat{\eta}_{k_{t-1},t} \mathbf{g}^{k_{t-1}}_t & K \mid (t-1), \\ \hat{\mathbf{w}}_t - \hat{\eta}_{k_{t-1},t} \mathbf{g}^{k_{t-1}}_t & K \nmid (t-1), \end{cases}
$$

for $t \geq 1$.

**Lemma 6.** *For any $t \geq 0$, the gap between $\mathbf{w}_{t+1}$ and $\hat{\mathbf{w}}_{t+1}$ can be formulated as follows:*

$$
\hat{\mathbf{w}}_{t+1} - \mathbf{w}_{t+1} = - \sum_{k\in[K], k\neq k_t} \frac{1 - \beta^{\lceil \frac{t}{K}\rceil - \lceil \frac{ite(k,t)}{K}\rceil + 1}}{1 - \beta} \hat{\eta}_{k,ite(k,t)} \mathbf{g}^k_{ite(k,t)}. \tag{15}
$$

*Proof.* Base case: for $t = 0$, $\hat{\mathbf{w}}_1 = \mathbf{w}_0 - \sum_{k\in[K]} \hat{\eta}_{k,0} \mathbf{g}^k_0$, $\mathbf{w}_1 = \mathbf{w}_0 - \hat{\eta}_{k_0,0} \mathbf{g}^{k_0}_0$, then we have

$$
\hat{\mathbf{w}}_1 - \mathbf{w}_1 = - \sum_{k\in[K], k\neq k_0} \frac{1 - \beta^{\lceil \frac{0}{K}\rceil - \lceil \frac{ite(k,0)}{K}\rceil + 1}}{1 - \beta} \hat{\eta}_{k,ite(k,0)} \mathbf{g}^k_{ite(k,0)}.
$$

Inductive hypothesis: for some arbitrary integer $t' - 1 \geq 0$, assume that (15) is true for $t = t' - 1$.

Inductive step: We will prove that (15) is true for $t = t'$. Firstly, we divide our discussion into two cases based on whether $t' - 1$ is divisible by $K$ and prove that

$$
\begin{aligned}
\hat{\mathbf{w}}_{t'+1} - \mathbf{w}_{t'+1} &= - \sum_{k\in[K], k\neq k_{t'-1}} \frac{1 - \beta^{\lceil \frac{t'}{K}\rceil - \lceil \frac{ite(k,t')}{K}\rceil + 1}}{1 - \beta} \hat{\eta}_{k,ite(k,t')} \mathbf{g}^k_{ite(k,t')} - \hat{\eta}_{k_{t'-1},t'} \mathbf{g}^{k_{t'-1}}_{t'} \\
&\quad + \frac{1 - \beta^{\lceil \frac{t'}{K}\rceil - \lceil \frac{ite(k_{t'},t')}{K}\rceil + 1}}{1 - \beta} \hat{\eta}_{k_{t'},ite(k_{t'},t')} \mathbf{g}^{k_{t'}}_{ite(k_{t'},t')}.
\end{aligned}
$$

Case 1: $K \mid (t' - 1)$

$$\hat{\mathbf{w}}_{t'+1} = \hat{\mathbf{w}}_{t'} - \beta \hat{\mathbf{u}}_{t'} - \hat{\eta}_{k_{t'-1},t'} \mathbf{g}_{t'}^{k_{t'-1}},$$

$$\mathbf{w}_{t'+1} = \mathbf{w}_{t'} - \beta \mathbf{u}_{t'} - \frac{1 - \beta^{\lceil \frac{t'}{K} \rceil - \lceil \frac{ite(k_{t'},t')}{K} \rceil + 1}}{1 - \beta} \hat{\eta}_{k_{t'},ite(k_{t'},t')} \mathbf{g}_{ite(k_{t'},t')}^{k_{t'}},$$

$$\hat{\mathbf{w}}_{t'+1} - \mathbf{w}_{t'+1} = \hat{\mathbf{w}}_{t'} - \mathbf{w}_{t'} - \beta \left( \hat{\mathbf{u}}_{t'} - \mathbf{u}_{t'} \right) - \hat{\eta}_{k_{t'-1},t'} \mathbf{g}_{t'}^{k_{t'-1}}$$

$$+ \frac{1 - \beta^{\lceil \frac{t'}{K} \rceil - \lceil \frac{ite(k_{t'},t')}{K} \rceil + 1}}{1 - \beta} \hat{\eta}_{k_{t'},ite(k_{t'},t')} \mathbf{g}_{ite(k_{t'},t')}^{k_{t'}}$$

$$= - \sum_{k \in [K], k \neq k_{t'-1}} \frac{1 - \beta^{\lceil \frac{t'-1}{K} \rceil - \lceil \frac{ite(k,t'-1)}{K} \rceil + 1}}{1 - \beta} \hat{\eta}_{k,ite(k,t'-1)} \mathbf{g}_{ite(k,t'-1)}^{k}$$

$$- \sum_{k \in [K], k \neq k_{t'-1}} \beta^{\lceil \frac{t'-1}{K} \rceil - \lceil \frac{ite(k,t'-1)}{K} \rceil + 1} \hat{\eta}_{k,ite(k,t'-1)} \mathbf{g}_{ite(k,t'-1)}^{k}$$

$$- \left( \hat{\eta}_{k_{t'-1},t'} \mathbf{g}_{t'}^{k_{t'-1}} - \frac{1 - \beta^{\lceil \frac{t'}{K} \rceil - \lceil \frac{ite(k_{t'},t')}{K} \rceil + 1}}{1 - \beta} \hat{\eta}_{k_{t'},ite(k_{t'},t')} \mathbf{g}_{ite(k_{t'},t')}^{k_{t'}} \right)$$

$$= - \sum_{k \in [K], k \neq k_{t'-1}} \frac{1 - \beta^{\lceil \frac{t'-1}{K} \rceil - \lceil \frac{ite(k,t'-1)}{K} \rceil + 2}}{1 - \beta} \hat{\eta}_{k,ite(k,t'-1)} \mathbf{g}_{ite(k,t'-1)}^{k}$$

$$- \hat{\eta}_{k_{t'-1},t'} \mathbf{g}_{t'}^{k_{t'-1}} + \frac{1 - \beta^{\lceil \frac{t'}{K} \rceil - \lceil \frac{ite(k_{t'},t')}{K} \rceil + 1}}{1 - \beta} \hat{\eta}_{k_{t'},ite(k_{t'},t')} \mathbf{g}_{ite(k_{t'},t')}^{k_{t'}}$$

$$= - \sum_{k \in [K], k \neq k_{t'-1}} \frac{1 - \beta^{\lceil \frac{t'}{K} \rceil - \lceil \frac{ite(k,t'-1)}{K} \rceil + 1}}{1 - \beta} \hat{\eta}_{k,ite(k,t'-1)} \mathbf{g}_{ite(k,t'-1)}^{k}$$

$$- \hat{\eta}_{k_{t'-1},t'} \mathbf{g}_{t'}^{k_{t'-1}} + \frac{1 - \beta^{\lceil \frac{t'}{K} \rceil - \lceil \frac{ite(k_{t'},t')}{K} \rceil + 1}}{1 - \beta} \hat{\eta}_{k_{t'},ite(k_{t'},t')} \mathbf{g}_{ite(k_{t'},t')}^{k_{t'}}.$$

The last equation above holds because $\lceil \frac{t'}{K} \rceil = \lceil \frac{t'-1}{K} \rceil + 1$.

Case 2: $K \nmid (t' - 1)$

$$\hat{\mathbf{w}}_{t'+1} = \hat{\mathbf{w}}_{t'} - \hat{\eta}_{k_{t'-1},t'} \mathbf{g}_{t'}^{k_{t'-1}},$$

$$\mathbf{w}_{t'+1} = \mathbf{w}_{t'} - \frac{1 - \beta^{\lceil \frac{t'}{K} \rceil - \lceil \frac{ite(k_{t'},t')}{K} \rceil + 1}}{1 - \beta} \hat{\eta}_{k_{t'},ite(k_{t'},t')} \mathbf{g}_{ite(k_{t'},t')}^{k_{t'}},$$

$$\hat{\mathbf{w}}_{t'+1} - \mathbf{w}_{t'+1} = \hat{\mathbf{w}}_{t'} - \mathbf{w}_{t'} - \left( \hat{\eta}_{k_{t'-1},t'} \mathbf{g}_{t'}^{k_{t'-1}} - \frac{1 - \beta^{\lceil \frac{t'}{K} \rceil - \lceil \frac{ite(k_{t'},t')}{K} \rceil + 1}}{1 - \beta} \hat{\eta}_{k_{t'},ite(k_{t'},t')} \mathbf{g}_{ite(k_{t'},t')}^{k_{t'}} \right)$$

$$= - \sum_{k \in [K], k \neq k_{t'-1}} \frac{1 - \beta^{\lceil \frac{t'-1}{K} \rceil - \lceil \frac{ite(k,t'-1)}{K} \rceil + 1}}{1 - \beta} \hat{\eta}_{k,ite(k,t'-1)} \mathbf{g}_{ite(k,t'-1)}^{k}$$

$$- \hat{\eta}_{k_{t'-1},t'} \mathbf{g}_{t'}^{k_{t'-1}} + \frac{1 - \beta^{\lceil \frac{t'}{K} \rceil - \lceil \frac{ite(k_{t'},t')}{K} \rceil + 1}}{1 - \beta} \hat{\eta}_{k_{t'},ite(k_{t'},t')} \mathbf{g}_{ite(k_{t'},t')}^{k_{t'}}$$

$$= - \sum_{k \in [K], k \neq k_{t'-1}} \frac{1 - \beta^{\lceil \frac{t'}{K} \rceil - \lceil \frac{ite(k,t'-1)}{K} \rceil + 1}}{1 - \beta} \hat{\eta}_{k,ite(k,t'-1)} \mathbf{g}_{ite(k,t'-1)}^{k}$$

$$- \hat{\eta}_{k_{t'-1},t'} \mathbf{g}_{t'}^{k_{t'-1}} + \frac{1 - \beta^{\lceil \frac{t'}{K} \rceil - \lceil \frac{ite(k_{t'},t')}{K} \rceil + 1}}{1 - \beta} \hat{\eta}_{k_{t'},ite(k_{t'},t')} \mathbf{g}_{ite(k_{t'},t')}^{k_{t'}}.$$

The last equation above holds because $\lceil \frac{t'}{K} \rceil = \lceil \frac{t'-1}{K} \rceil$.

Since $ite(k,t') = ite(k,t'-1), \forall k \neq k_{t'-1}$, we can get the following equation for both cases above:

$$\hat{\mathbf{w}}_{t'+1} - \mathbf{w}_{t'+1} = -\sum_{k\in[K],k\neq k_{t'-1}} \frac{1 - \beta^{\lceil\frac{t'}{K}\rceil - \lceil\frac{ite(k,t')}{K}\rceil+1}}{1-\beta}\hat{\eta}_{k,ite(k,t')}\mathbf{g}^k_{ite(k,t')} - \hat{\eta}_{k_{t'-1},t'}\mathbf{g}^{k_{t'-1}}_{t'}$$

$$+ \frac{1 - \beta^{\lceil\frac{t'}{K}\rceil - \lceil\frac{ite(k_{t'},t')}{K}\rceil+1}}{1-\beta}\hat{\eta}_{k_{t'},ite(k_{t'},t')}\mathbf{g}^{k_{t'}}_{ite(k_{t'},t')}.$$

If $k_{t'} = k_{t'-1}$, then we have $ite(k_{t'},t') = t'$ and

$$\hat{\mathbf{w}}_{t'+1} - \mathbf{w}_{t'+1} = -\sum_{k\in[K],k\neq k_{t'}} \frac{1 - \beta^{\lceil\frac{t'}{K}\rceil - \lceil\frac{ite(k,t')}{K}\rceil+1}}{1-\beta}\hat{\eta}_{k,ite(k,t')}\mathbf{g}^k_{ite(k,t')}.$$

If $k_{t'} \neq k_{t'-1}$, then we have

$$\hat{\mathbf{w}}_{t'+1} - \mathbf{w}_{t'+1} = -\sum_{k\in[K],k\neq k_{t'-1}} \frac{1 - \beta^{\lceil\frac{t'}{K}\rceil - \lceil\frac{ite(k,t')}{K}\rceil+1}}{1-\beta}\hat{\eta}_{k,ite(k,t')}\mathbf{g}^k_{ite(k,t')} - \hat{\eta}_{k_{t'-1},t'}\mathbf{g}^{k_{t'-1}}_{t'}$$

$$+ \frac{1 - \beta^{\lceil\frac{t'}{K}\rceil - \lceil\frac{ite(k_{t'},t')}{K}\rceil+1}}{1-\beta}\hat{\eta}_{k_{t'},ite(k_{t'},t')}\mathbf{g}^{k_{t'}}_{ite(k_{t'},t')}$$

$$= -\sum_{k\in[K],k\neq k_{t'-1},k\neq k_{t'}} \frac{1 - \beta^{\lceil\frac{t'}{K}\rceil - \lceil\frac{ite(k,t')}{K}\rceil+1}}{1-\beta}\hat{\eta}_{k,ite(k,t')}\mathbf{g}^k_{ite(k,t')} - \hat{\eta}_{k_{t'-1},t'}\mathbf{g}^{k_{t'-1}}_{t'}$$

$$= -\sum_{k\in[K],k\neq k_{t'}} \frac{1 - \beta^{\lceil\frac{t'}{K}\rceil - \lceil\frac{ite(k,t')}{K}\rceil+1}}{1-\beta}\hat{\eta}_{k,ite(k,t')}\mathbf{g}^k_{ite(k,t')}$$

We can conclude that $\hat{\mathbf{w}}_{t+1} - \mathbf{w}_{t+1} = -\sum_{k\in[K],k\neq k_t} \frac{1-\beta^{\lceil\frac{t}{K}\rceil - \lceil\frac{ite(k,t)}{K}\rceil+1}}{1-\beta}\hat{\eta}_{k,ite(k,t)}\mathbf{g}^k_{ite(k,t)}$ is true for any $t \geq 0$. $\qquad\square$

Then, we define another auxiliary sequence $\{\hat{\mathbf{y}}_t\}_{t\geq1}$: $\hat{\mathbf{y}}_1 = \mathbf{w}_0 - \frac{1}{1-\beta}\sum_{k\in[K]}\hat{\eta}_{k,0}\mathbf{g}^k_0$, and

$$\hat{\mathbf{y}}_{t+1} = \hat{\mathbf{y}}_t - \frac{1}{1-\beta}\hat{\eta}_{k_{t-1},t}\mathbf{g}^{k_{t-1}}_t,$$

for $t \geq 1$.

**Lemma 7.** *For any $t \geq 1$, the gap between $\hat{\mathbf{y}}_t$ and $\hat{\mathbf{w}}_t$ can be formulated as follows:*

$$\hat{\mathbf{y}}_t - \hat{\mathbf{w}}_t = -\frac{\beta}{1-\beta}\hat{\mathbf{u}}_t. \tag{16}$$

*Proof.* Base case: For $t = 1$, we have that

$$\hat{\mathbf{y}}_1 - \hat{\mathbf{w}}_1 = \left(\mathbf{w}_0 - \frac{1}{1-\beta}\sum_{k\in[K]}\hat{\eta}_{k,0}\mathbf{g}^k_0\right) - \left(\mathbf{w}_0 - \sum_{k\in[K]}\hat{\eta}_{k,0}\mathbf{g}^k_0\right)$$

$$= -\frac{\beta}{1-\beta}\sum_{k\in[K]}\hat{\eta}_{k,0}\mathbf{g}^k_0 = -\frac{\beta}{1-\beta}\hat{\mathbf{u}}_1.$$

Inductive hypothesis: for some arbitrary integer $t' \geq 1$, assume that $\hat{\mathbf{y}}_t - \hat{\mathbf{w}}_t = -\frac{\beta}{1-\beta}\hat{\mathbf{u}}_t$ is true for $t = t'$.

Inductive step: We will prove that $\hat{\mathbf{y}}_t - \hat{\mathbf{w}}_t = -\frac{\beta}{1-\beta}\hat{\mathbf{u}}_t$ is true for $t = t'+1$. We divide our discussion into two cases based on whether $t' - 1$ is divisible by $K$.

Case 1: $K \mid (t'-1)$

$$\hat{\mathbf{y}}_{t'+1} = \hat{\mathbf{y}}_{t'} - \frac{1}{1-\beta}\hat{\eta}_{k_{t'-1},t'}\mathbf{g}_{t'}^{k_{t'-1}}$$

$$\hat{\mathbf{w}}_{t'+1} = \hat{\mathbf{w}}_{t'} - \beta\hat{\mathbf{u}}_{t'} - \hat{\eta}_{k_{t'-1},t'}\mathbf{g}_{t'}^{k_{t'-1}}$$

$$\hat{\mathbf{y}}_{t'+1} - \hat{\mathbf{w}}_{t'+1} = \left(\hat{\mathbf{y}}_{t'} - \frac{1}{1-\beta}\hat{\eta}_{k_{t'-1},t'}\mathbf{g}_{t'}^{k_{t'-1}}\right) - \left(\hat{\mathbf{w}}_{t'} - \beta\hat{\mathbf{u}}_{t'} - \hat{\eta}_{k_{t'-1},t'}\mathbf{g}_{t'}^{k_{t'-1}}\right)$$

$$= \hat{\mathbf{y}}_{t'} - \hat{\mathbf{w}}_{t'} + \beta\hat{\mathbf{u}}_{t'} - \frac{\beta}{1-\beta}\hat{\eta}_{k_{t'-1},t'}\mathbf{g}_{t'}^{k_{t'-1}}$$

$$= -\frac{\beta^2}{1-\beta}\hat{\mathbf{u}}_{t'} - \frac{\beta}{1-\beta}\hat{\eta}_{k_{t'-1},t'}\mathbf{g}_{t'}^{k_{t'-1}}$$

$$= -\frac{\beta}{1-\beta}\hat{\mathbf{u}}_{t'+1}$$

Case 2: $K \nmid (t'-1)$

$$\hat{\mathbf{y}}_{t'+1} = \hat{\mathbf{y}}_{t'} - \frac{1}{1-\beta}\hat{\eta}_{k_{t'-1},t'}\mathbf{g}_{t'}^{k_{t'-1}}$$

$$\hat{\mathbf{w}}_{t'+1} = \hat{\mathbf{w}}_{t'} - \hat{\eta}_{k_{t'-1},t'}\mathbf{g}_{t'}^{k_{t'-1}}$$

$$\hat{\mathbf{y}}_{t'+1} - \hat{\mathbf{w}}_{t'+1} = \left(\hat{\mathbf{y}}_{t'} - \frac{1}{1-\beta}\hat{\eta}_{k_{t'-1},t'}\mathbf{g}_{t'}^{k_{t'-1}}\right) - \left(\hat{\mathbf{w}}_{t'} - \hat{\eta}_{k_{t'-1},t'}\mathbf{g}_{t'}^{k_{t'-1}}\right)$$

$$= \hat{\mathbf{y}}_{t'} - \hat{\mathbf{w}}_{t'} - \frac{\beta}{1-\beta}\hat{\eta}_{k_{t'-1},t'}\mathbf{g}_{t'}^{k_{t'-1}}$$

$$= -\frac{\beta}{1-\beta}\hat{\mathbf{u}}_{t'} - \frac{\beta}{1-\beta}\hat{\eta}_{k_{t'-1},t'}\mathbf{g}_{t'}^{k_{t'-1}}$$

$$= -\frac{\beta}{1-\beta}\hat{\mathbf{u}}_{t'+1}$$

We can conclude that $\hat{\mathbf{y}}_t - \hat{\mathbf{w}}_t = -\frac{\beta}{1-\beta}\hat{\mathbf{u}}_t$ is true for any $t \geq 1$. $\qquad\square$

**Lemma 8.** *For any $t \geq 1$, $\hat{\mathbf{u}}_t$ can be formulated as follows:*

$$\hat{\mathbf{u}}_t = \beta^{\lfloor\frac{t+K-2}{K}\rfloor}\left(\sum_{k\in[K]}\hat{\eta}_{k,0}\mathbf{g}_0^k\right) + \sum_{s=1}^{\lfloor\frac{t+K-2}{K}\rfloor}\beta^{\lfloor\frac{t+K-2}{K}\rfloor-s}\left(\sum_{j=(s-1)K+1}^{\min\{sK,t-1\}}\hat{\eta}_{k_{j-1},j}\mathbf{g}_j^{k_{j-1}}\right).$$

*Proof.* It's straightforward to get this conclusion from the definition of the sequence $\hat{\mathbf{u}}_t$ in (13). $\quad\square$

**Lemma 9.** *(descent lemma) With Assumptions 1 and 3, we have the following descent lemma for $t \geq 1$,:*

$$\mathbb{E}F(\hat{\mathbf{y}}_{t+1}) \leq F(\hat{\mathbf{y}}_t) + \left(\frac{L(\hat{\eta}_{k_{t-1},t})^2}{2(1-\beta)^2} - \frac{\hat{\eta}_{k_{t-1},t}}{2(1-\beta)}\right)\|\nabla F(\mathbf{w}_t)\|^2 + \frac{(\hat{\eta}_{k_{t-1},t})^2\sigma^2 L}{2(1-\beta)^2}$$

$$+ \frac{L^2\hat{\eta}_{k_{t-1},t}}{1-\beta}\|\hat{\mathbf{y}}_t - \hat{\mathbf{w}}_t\|^2 + \frac{L^2\hat{\eta}_{k_{t-1},t}}{1-\beta}\|\hat{\mathbf{w}}_t - \mathbf{w}_t\|^2.$$

*Proof.*

$$\hat{\mathbf{y}}_{t+1} = \hat{\mathbf{y}}_t - \frac{1}{1-\beta}\hat{\eta}_{k_{t-1},t}\mathbf{g}_t^{k_{t-1}}$$

$$\mathbb{E}F(\hat{\mathbf{y}}_{t+1}) \leq F(\hat{\mathbf{y}}_t) + \mathbb{E}\langle\nabla F(\hat{\mathbf{y}}_t), \hat{\mathbf{y}}_{t+1} - \hat{\mathbf{y}}_t\rangle + \frac{L}{2}\mathbb{E}\|\hat{\mathbf{y}}_{t+1} - \hat{\mathbf{y}}_t\|^2$$

$$= F(\hat{\mathbf{y}}_t) - \frac{1}{1-\beta}\mathbb{E}\langle\nabla F(\hat{\mathbf{y}}_t), \hat{\eta}_{k_{t-1},t}\mathbf{g}_t^{k_{t-1}}\rangle + \frac{L}{2(1-\beta)^2}\mathbb{E}\left[(\hat{\eta}_{k_{t-1},t})^2 \left\|\mathbf{g}_t^{k_{t-1}}\right\|^2\right]$$

$$= F(\hat{\mathbf{y}}_t) - \frac{1}{1-\beta}\langle\nabla F(\hat{\mathbf{y}}_t), \hat{\eta}_{k_{t-1},t}\nabla F(\mathbf{w}_t)\rangle + \frac{L}{2(1-\beta)^2}\mathbb{E}\left[(\hat{\eta}_{k_{t-1},t})^2 \left\|\mathbf{g}_t^{k_{t-1}}\right\|^2\right].$$

$$-\frac{\hat{\eta}_{k_{t-1},t}}{1-\beta}\langle\nabla F(\hat{\mathbf{y}}_t), \nabla F(\mathbf{w}_t)\rangle = -\frac{\hat{\eta}_{k_{t-1},t}}{1-\beta}\langle\nabla F(\hat{\mathbf{y}}_t) - \nabla F(\mathbf{w}_t) + \nabla F(\mathbf{w}_t), \nabla F(\mathbf{w}_t)\rangle$$

$$= -\frac{\hat{\eta}_{k_{t-1},t}}{1-\beta}\langle\nabla F(\hat{\mathbf{y}}_t) - \nabla F(\mathbf{w}_t), \nabla F(\mathbf{w}_t)\rangle - \frac{\hat{\eta}_{k_{t-1},t}}{1-\beta}\|\nabla F(\mathbf{w}_t)\|^2$$

$$\leq \frac{\hat{\eta}_{k_{t-1},t}}{2(1-\beta)}\|\nabla F(\hat{\mathbf{y}}_t) - \nabla F(\mathbf{w}_t)\|^2 - \frac{\hat{\eta}_{k_{t-1},t}}{2(1-\beta)}\|\nabla F(\mathbf{w}_t)\|^2$$

$$\leq \frac{\hat{\eta}_{k_{t-1},t}L^2}{2(1-\beta)}\|\hat{\mathbf{y}}_t - \mathbf{w}_t\|^2 - \frac{\hat{\eta}_{k_{t-1},t}}{2(1-\beta)}\|\nabla F(\mathbf{w}_t)\|^2$$

$$\frac{L}{2(1-\beta)^2}\mathbb{E}\left[(\hat{\eta}_{k_{t-1},t})^2 \left\|\mathbf{g}_t^{k_{t-1}}\right\|^2\right] = \frac{L(\hat{\eta}_{k_{t-1},t})^2}{2(1-\beta)^2}\mathbb{E}\left\|\mathbf{g}_t^{k_{t-1}} - \nabla F(\mathbf{w}_t) + \nabla F(\mathbf{w}_t)\right\|^2$$

$$\leq \frac{L(\hat{\eta}_{k_{t-1},t})^2}{2(1-\beta)^2}\left(\mathbb{E}\left\|\mathbf{g}_t^{k_{t-1}} - \nabla F(\mathbf{w}_t)\right\|^2 + \|\nabla F(\mathbf{w}_t)\|^2\right)$$

$$\leq \frac{(\hat{\eta}_{k_{t-1},t})^2 L}{2(1-\beta)^2}\left(\sigma^2 + \|\nabla F(\mathbf{w}_t)\|^2\right)$$

$$\mathbb{E}F(\hat{\mathbf{y}}_{t+1}) \leq F(\hat{\mathbf{y}}_t) + \left(\frac{L(\hat{\eta}_{k_{t-1},t})^2}{2(1-\beta)^2} - \frac{\hat{\eta}_{k_{t-1},t}}{2(1-\beta)}\right)\|\nabla F(\mathbf{w}_t)\|^2 + \frac{(\hat{\eta}_{k_{t-1},t})^2\sigma^2 L}{2(1-\beta)^2}$$

$$+ \frac{L^2\hat{\eta}_{k_{t-1},t}}{2(1-\beta)}\|\hat{\mathbf{y}}_t - \mathbf{w}_t\|^2$$

$$\leq F(\hat{\mathbf{y}}_t) + \left(\frac{L(\hat{\eta}_{k_{t-1},t})^2}{2(1-\beta)^2} - \frac{\hat{\eta}_{k_{t-1},t}}{2(1-\beta)}\right)\|\nabla F(\mathbf{w}_t)\|^2 + \frac{(\hat{\eta}_{k_{t-1},t})^2\sigma^2 L}{2(1-\beta)^2}$$

$$+ \frac{L^2\hat{\eta}_{k_{t-1},t}}{1-\beta}\|\hat{\mathbf{y}}_t - \hat{\mathbf{w}}_t\|^2 + \frac{L^2\hat{\eta}_{k_{t-1},t}}{1-\beta}\|\hat{\mathbf{w}}_t - \mathbf{w}_t\|^2.$$

$\square$

**Lemma 10.** *With Assumption 1, the gap between $\hat{\mathbf{y}}_t$ and $\hat{\mathbf{w}}_t$ in OrMo-DA can be bounded as follows:*

$$\sum_{t=1}^{T-1}\mathbb{E}\left(\hat{\eta}_{k_{t-1},t}\|\hat{\mathbf{y}}_t - \hat{\mathbf{w}}_t\|^2\right) \leq \frac{2\beta^2\eta K^2}{(1-\beta)^4}\left[\sum_{k\in[K]}(\hat{\eta}_{k,0})^2\|\nabla F(\mathbf{w}_0)\|^2 + \sum_{t=1}^{T-1}(\hat{\eta}_{k_{t-1},t})^2\mathbb{E}\|\nabla F(\mathbf{w}_t)\|^2\right]$$

$$+ \frac{2\beta^2\eta K\sigma^2}{(1-\beta)^3}\left[\sum_{k\in[K]}(\hat{\eta}_{k,0})^2 + \sum_{t=1}^{T-1}(\hat{\eta}_{k_{t-1},t})^2\right].$$

*Proof.*

$$
\mathbb{E}\left\|\hat{\mathbf{u}}_t\right\|^2 = \mathbb{E}\left\|\beta^{\lfloor\frac{t+K-2}{K}\rfloor}\left(\sum_{k\in[K]}\hat{\eta}_{k,0}\mathbf{g}_0^k\right) + \sum_{s=1}^{\lfloor\frac{t+K-2}{K}\rfloor}\beta^{\lfloor\frac{t+K-2}{K}\rfloor-s}\left(\sum_{j=(s-1)K+1}^{\min\{sK,t-1\}}\hat{\eta}_{k_{j-1},j}\mathbf{g}_j^{k_{j-1}}\right)\right\|^2
$$

$$
\leq 2\mathbb{E}\left\|\beta^{\lfloor\frac{t+K-2}{K}\rfloor}\left(\sum_{k\in[K]}\hat{\eta}_{k,0}\left(\mathbf{g}_0^k - \nabla F(\mathbf{w}_0)\right)\right)\right.
$$
$$
\left. + \sum_{s=1}^{\lfloor\frac{t+K-2}{K}\rfloor}\beta^{\lfloor\frac{t+K-2}{K}\rfloor-s}\left(\sum_{j=(s-1)K+1}^{\min\{sK,t-1\}}\hat{\eta}_{k_{j-1},j}\left(\mathbf{g}_j^{k_{j-1}} - \nabla F(\mathbf{w}_j)\right)\right)\right\|^2
$$
$$
+ 2\mathbb{E}\left\|\beta^{\lfloor\frac{t+K-2}{K}\rfloor}\left(\sum_{k\in[K]}\hat{\eta}_{k,0}\nabla F(\mathbf{w}_0)\right) + \sum_{s=1}^{\lfloor\frac{t+K-2}{K}\rfloor}\beta^{\lfloor\frac{t+K-2}{K}\rfloor-s}\left(\sum_{j=(s-1)K+1}^{\min\{sK,t-1\}}\hat{\eta}_{k_{j-1},j}\nabla F(\mathbf{w}_j)\right)\right\|^2
$$

$$
\leq 2\sum_{k\in[K]}\beta^{2\lfloor\frac{t+K-2}{K}\rfloor}\left(\hat{\eta}_{k,0}\right)^2\sigma^2 + 2\sum_{s=1}^{\lfloor\frac{t+K-2}{K}\rfloor}\sum_{j=(s-1)K+1}^{\min\{sK,t-1\}}\beta^{2\lfloor\frac{t+K-2}{K}\rfloor-2s}\left(\hat{\eta}_{k_{j-1},j}\right)^2\sigma^2
$$
$$
+ 2\mathbb{E}\left\|\beta^{\lfloor\frac{t+K-2}{K}\rfloor}\left(\sum_{k\in[K]}\hat{\eta}_{k,0}\nabla F(\mathbf{w}_0)\right) + \sum_{s=1}^{\lfloor\frac{t+K-2}{K}\rfloor}\beta^{\lfloor\frac{t+K-2}{K}\rfloor-s}\left(\sum_{j=(s-1)K+1}^{\min\{sK,t-1\}}\hat{\eta}_{k_{j-1},j}\nabla F(\mathbf{w}_j)\right)\right\|^2
$$

Let $q_t = \sum_{k\in[K]}\beta^{\lfloor\frac{t+K-2}{K}\rfloor} + \sum_{s=1}^{\lfloor\frac{t+K-2}{K}\rfloor}\sum_{j=(s-1)K+1}^{\min\{sK,t-1\}}\beta^{\lfloor\frac{t+K-2}{K}\rfloor-s}$, then we have

$$
\mathbb{E}\left\|\beta^{\lfloor\frac{t+K-2}{K}\rfloor}\left(\sum_{k\in[K]}\hat{\eta}_{k,0}\nabla F(\mathbf{w}_0)\right) + \sum_{s=1}^{\lfloor\frac{t+K-2}{K}\rfloor}\beta^{\lfloor\frac{t+K-2}{K}\rfloor-s}\left(\sum_{j=(s-1)K+1}^{\min\{sK,t-1\}}\hat{\eta}_{k_{j-1},j}\nabla F(\mathbf{w}_j)\right)\right\|^2
$$

$$
= q_t^2\mathbb{E}\left\|\left(\sum_{k\in[K]}\frac{\beta^{\lfloor\frac{t+K-2}{K}\rfloor}}{q_t}\hat{\eta}_{k,0}\nabla F(\mathbf{w}_0)\right) + \sum_{s=1}^{\lfloor\frac{t+K-2}{K}\rfloor}\left(\sum_{j=(s-1)K+1}^{\min\{sK,t-1\}}\frac{\beta^{\lfloor\frac{t+K-2}{K}\rfloor-s}}{q_t}\hat{\eta}_{k_{j-1},j}\nabla F(\mathbf{w}_j)\right)\right\|^2
$$

$$
\leq q_t\left[\sum_{k\in[K]}\beta^{\lfloor\frac{t+K-2}{K}\rfloor}\left(\hat{\eta}_{k,0}\right)^2\left\|\nabla F(\mathbf{w}_0)\right\|^2\right.
$$
$$
\left. + \sum_{s=1}^{\lfloor\frac{t+K-2}{K}\rfloor}\sum_{j=(s-1)K+1}^{\min\{sK,t-1\}}\beta^{\lfloor\frac{t+K-2}{K}\rfloor-s}\left(\hat{\eta}_{k_{j-1},j}\right)^2\mathbb{E}\left\|\nabla F(\mathbf{w}_j)\right\|^2\right]
$$

$$
\leq \frac{K}{1-\beta}\left[\sum_{k\in[K]}\beta^{\lfloor\frac{t+K-2}{K}\rfloor}\left(\hat{\eta}_{k,0}\right)^2\left\|\nabla F(\mathbf{w}_0)\right\|^2\right.
$$
$$
\left. + \sum_{s=1}^{\lfloor\frac{t+K-2}{K}\rfloor}\sum_{j=(s-1)K+1}^{\min\{sK,t-1\}}\beta^{\lfloor\frac{t+K-2}{K}\rfloor-s}\left(\hat{\eta}_{k_{j-1},j}\right)^2\mathbb{E}\left\|\nabla F(\mathbf{w}_j)\right\|^2\right].
$$

Thus, we get that

$$\mathbb{E}\left\|\hat{\mathbf{u}}_t\right\|^2$$

$$\leq 2\mathbb{E}\left\|\beta^{\lfloor\frac{t+K-2}{K}\rfloor}\left(\sum_{k\in[K]}\hat{\eta}_{k,0}\nabla F(\mathbf{w}_0)\right)+\sum_{s=1}^{\lfloor\frac{t+K-2}{K}\rfloor}\beta^{\lfloor\frac{t+K-2}{K}\rfloor-s}\left(\sum_{j=(s-1)K+1}^{\min\{sK,t-1\}}\hat{\eta}_{k_{j-1},j}\nabla F(\mathbf{w}_j)\right)\right\|^2$$

$$+2\sum_{k\in[K]}\beta^{2\lfloor\frac{t+K-2}{K}\rfloor}(\hat{\eta}_{k,0})^2\sigma^2+2\sum_{s=1}^{\lfloor\frac{t+K-2}{K}\rfloor}\sum_{j=(s-1)K+1}^{\min\{sK,t-1\}}\beta^{2\lfloor\frac{t+K-2}{K}\rfloor-2s}\left(\hat{\eta}_{k_{j-1},j}\right)^2\sigma^2$$

$$\leq\frac{2K}{1-\beta}\left[\sum_{k\in[K]}\beta^{\lfloor\frac{t+K-2}{K}\rfloor}(\hat{\eta}_{k,0})^2\left\|\nabla F(\mathbf{w}_0)\right\|^2+\sum_{s=1}^{\lfloor\frac{t+K-2}{K}\rfloor}\sum_{j=(s-1)K+1}^{\min\{sK,t-1\}}\beta^{\lfloor\frac{t+K-2}{K}\rfloor-s}\left(\hat{\eta}_{k_{j-1},j}\right)^2\mathbb{E}\left\|\nabla F(\mathbf{w}_j)\right\|^2\right]$$

$$+2\sum_{k\in[K]}\beta^{2\lfloor\frac{t+K-2}{K}\rfloor}(\hat{\eta}_{k,0})^2\sigma^2+2\sum_{s=1}^{\lfloor\frac{t+K-2}{K}\rfloor}\sum_{j=(s-1)K+1}^{\min\{sK,t-1\}}\beta^{2\lfloor\frac{t+K-2}{K}\rfloor-2s}\left(\hat{\eta}_{k_{j-1},j}\right)^2\sigma^2.$$

$$\sum_{t=1}^{T-1}\mathbb{E}\left(\hat{\eta}_{k_{t-1},t}\left\|\hat{\mathbf{y}}_t-\hat{\mathbf{w}}_t\right\|^2\right)\leq\frac{\beta^2\eta}{(1-\beta)^2}\sum_{t=1}^{T-1}\mathbb{E}\left\|\hat{\mathbf{u}}_t\right\|^2$$

$$\leq\frac{2\beta^2K\eta}{(1-\beta)^3}\sum_{t=1}^{T-1}\sum_{k\in[K]}\beta^{\lfloor\frac{t+K-2}{K}\rfloor}(\hat{\eta}_{k,0})^2\left\|\nabla F(\mathbf{w}_0)\right\|^2$$

$$+\frac{2\beta^2K\eta}{(1-\beta)^3}\sum_{t=1}^{T-1}\sum_{s=1}^{\lfloor\frac{t+K-2}{K}\rfloor}\sum_{j=(s-1)K+1}^{\min\{sK,t-1\}}\beta^{\lfloor\frac{t+K-2}{K}\rfloor-s}\left(\hat{\eta}_{k_{j-1},j}\right)^2\mathbb{E}\left\|\nabla F(\mathbf{w}_j)\right\|^2$$

$$+\frac{2\beta^2\eta\sigma^2}{(1-\beta)^2}\sum_{t=1}^{T-1}\left[\sum_{k\in[K]}\beta^{2\lfloor\frac{t+K-2}{K}\rfloor}(\hat{\eta}_{k,0})^2+\sum_{s=1}^{\lfloor\frac{t+K-2}{K}\rfloor}\sum_{j=(s-1)K+1}^{\min\{sK,t-1\}}\beta^{2\lfloor\frac{t+K-2}{K}\rfloor-2s}\left(\hat{\eta}_{k_{j-1},j}\right)^2\right]$$

$$\leq\frac{2\beta^2\eta K^2}{(1-\beta)^4}\left[\sum_{k\in[K]}(\hat{\eta}_{k,0})^2\left\|\nabla F(\mathbf{w}_0)\right\|^2+\sum_{t=1}^{T-1}\left(\hat{\eta}_{k_{t-1},t}\right)^2\mathbb{E}\left\|\nabla F(\mathbf{w}_t)\right\|^2\right]$$

$$+\frac{2\beta^2\eta K\sigma^2}{(1-\beta)^3}\left[\sum_{k\in[K]}(\hat{\eta}_{k,0})^2+\sum_{t=1}^{T-1}\left(\hat{\eta}_{k_{t-1},t}\right)^2\right].$$

$\square$

**Lemma 11.** *With Assumption 1, letting $\eta\leq\frac{1}{8KL}$, the gap between $\hat{\mathbf{w}}_t$ and $\mathbf{w}_t$ in OrMo-DA can be bounded as follows:*

$$\sum_{t=1}^{T-1}\mathbb{E}\left(\hat{\eta}_{k_{t-1},t}\left\|\hat{\mathbf{w}}_t-\mathbf{w}_t\right\|^2\right)\leq\frac{\eta K}{2L(1-\beta)^2}\left[\sum_{t=1}^{T-1}\hat{\eta}_{k_{t-1},t}\mathbb{E}\left\|\nabla F(\mathbf{w}_t)\right\|^2+\left(\sum_{k\in[K]}\hat{\eta}_{k,0}\right)\left\|\nabla F(\mathbf{w}_0)\right\|^2\right]$$

$$+\frac{\eta\sigma^2}{2L(1-\beta)^2}\left(\sum_{t=1}^{T-1}\hat{\eta}_{k_{t-1},t}+\sum_{k\in[K]}\hat{\eta}_{k,0}\right).$$

*Proof.*

$$\sum_{t=1}^{T-1} \mathbb{E}\left(\hat{\eta}_{k_{t-1},t} \|\hat{\mathbf{w}}_t - \mathbf{w}_t\|^2\right) \leq \eta \sum_{t=1}^{T-1} \mathbb{E} \|\hat{\mathbf{w}}_t - \mathbf{w}_t\|^2$$

$$= \eta \sum_{t=1}^{T-1} \mathbb{E} \left\| \sum_{k\in[K], k\neq k_{t-1}} \frac{1 - \beta^{\lceil \frac{t-1}{K} \rceil - \lceil \frac{ite(k,t-1)}{K} \rceil + 1}}{1-\beta} \hat{\eta}_{k,ite(k,t-1)} \mathbf{g}^k_{ite(k,t-1)} \right\|^2$$

$$\leq 2\eta \sum_{t=1}^{T-1} \mathbb{E} \left\| \sum_{k\in[K], k\neq k_{t-1}} \frac{1 - \beta^{\lceil \frac{t-1}{K} \rceil - \lceil \frac{ite(k,t-1)}{K} \rceil + 1}}{1-\beta} \hat{\eta}_{k,ite(k,t-1)} \left(\mathbf{g}^k_{ite(k,t-1)} - \nabla F(\mathbf{w}_{ite(k,t-1)})\right) \right\|^2$$

$$+ 2\eta \sum_{t=1}^{T-1} \mathbb{E} \left\| \sum_{k\in[K], k\neq k_{t-1}} \frac{1 - \beta^{\lceil \frac{t-1}{K} \rceil - \lceil \frac{ite(k,t-1)}{K} \rceil + 1}}{1-\beta} \hat{\eta}_{k,ite(k,t-1)} \nabla F(\mathbf{w}_{ite(k,t-1)}) \right\|^2$$

$$\leq \frac{2\eta}{(1-\beta)^2} \sum_{t=0}^{T-2} \sum_{k\in[K], k\neq k_t} \left(\hat{\eta}_{k,ite(k,t)}\right)^2 \left(\sigma^2 + K\mathbb{E} \|\nabla F(\mathbf{w}_{ite(k,t)})\|^2\right)$$

$$= \frac{2\eta}{(1-\beta)^2} \sum_{j=0}^{T-2} \sum_{t=0}^{T-2} \sum_{k\in[K]} \left(\hat{\eta}_{k,ite(k,t)}\right)^2 \left(\sigma^2 + K\mathbb{E} \|\nabla F(\mathbf{w}_{ite(k,t)})\|^2\right) \mathbb{1}\,(k \neq k_t)\, \mathbb{1}\,(j = ite(k,t))$$

$$= \frac{2\eta}{(1-\beta)^2} \sum_{j=1}^{T-2} \sum_{t=0}^{T-2} \sum_{k\in[K]} \left(\hat{\eta}_{k,ite(k,t)}\right)^2 \left(\sigma^2 + K\mathbb{E} \|\nabla F(\mathbf{w}_{ite(k,t)})\|^2\right) \mathbb{1}\,(k \neq k_t)\, \mathbb{1}\,(j = ite(k,t))$$

$$+ \frac{2\eta}{(1-\beta)^2} \sum_{t=0}^{T-2} \sum_{k\in[K]} \left(\hat{\eta}_{k,ite(k,t)}\right)^2 \left(\sigma^2 + K\mathbb{E} \|\nabla F(\mathbf{w}_{ite(k,t)})\|^2\right) \mathbb{1}\,(k \neq k_t)\, \mathbb{1}\,(0 = ite(k,t))$$

$$\leq \frac{2\eta}{(1-\beta)^2} \sum_{j=1}^{T-2} \left(\hat{\eta}_{k_{j-1},j}\right)^2 \left(\sigma^2 + K\mathbb{E} \|\nabla F(\mathbf{w}_j)\|^2\right) (next(k_{j-1},j) - j)$$

$$+ \frac{2\eta}{(1-\beta)^2} \sum_{k\in[K]} (\hat{\eta}_{k,0})^2 \left(\sigma^2 + K \|\nabla F(\mathbf{w}_0)\|^2\right) (next(k,0) - 0)$$

$$\sum_{t=1}^{T-1} \mathbb{E}\left(\hat{\eta}_{k_{t-1},t} \|\hat{\mathbf{w}}_t - \mathbf{w}_t\|^2\right) \leq \frac{2\eta}{(1-\beta)^2} \sum_{t=1}^{T-1} (\hat{\eta}_{k_{t-1},t})^2 \left(\sigma^2 + K\mathbb{E} \|\nabla F(\mathbf{w}_t)\|^2\right) \hat{\tau}_{k_{t-1},next(k_{t-1},t)}$$

$$+ \frac{2\eta}{(1-\beta)^2} \sum_{k\in[K]} (\hat{\eta}_{k,0})^2 \left(\sigma^2 + K \|\nabla F(\mathbf{w}_0)\|^2\right) \hat{\tau}_{k,next(k,0)}$$

$$\sum_{t=1}^{T-1} \mathbb{E}\left(\hat{\eta}_{k_{t-1},t} \|\hat{\mathbf{w}}_t - \mathbf{w}_t\|^2\right)$$

$$\leq \frac{\eta}{2L(1-\beta)^2} \left[ \sum_{t=1}^{T-1} \hat{\eta}_{k_{t-1},t} \left[\sigma^2 + K\mathbb{E} \|\nabla F(\mathbf{w}_t)\|^2\right] + \sum_{k\in[K]} \hat{\eta}_{k,0} \left[\sigma^2 + K \|\nabla F(\mathbf{w}_0)\|^2\right] \right]$$

$$\leq \frac{\eta K}{2L(1-\beta)^2} \left[ \sum_{t=1}^{T-1} \hat{\eta}_{k_{t-1},t} \mathbb{E} \|\nabla F(\mathbf{w}_t)\|^2 + \left( \sum_{k\in[K]} \hat{\eta}_{k,0} \right) \|\nabla F(\mathbf{w}_0)\|^2 \right]$$

$$+ \frac{\eta \sigma^2}{2L(1-\beta)^2} \left( \sum_{t=1}^{T-1} \hat{\eta}_{k_{t-1},t} + \sum_{k\in[K]} \hat{\eta}_{k,0} \right).$$

$\square$

**Theorem 3.** *With Assumptions 1, 3 and 4, letting*

$$\eta_t = \begin{cases} \eta & \tau_t \leq 2K, \\ \min\{\eta, \dfrac{1}{4L\tau_t}\} & \tau_t > 2K, \end{cases}$$

*and* $\eta = \min\{\frac{(1-\beta)^2}{8KL}, \sqrt{\frac{(1-\beta)^3\Delta}{TL\sigma^2}}\}$, *Algorithm 5 has the following convergence rate:*

$$\mathbb{E}\left\|\nabla F(\bar{\mathbf{w}}_T)\right\|^2 \leq \mathcal{O}(\sqrt{\frac{L\sigma^2}{T}} + \frac{KL}{T}),$$

*where* $\Delta = F(\mathbf{w}_0) - F^*$ *and* $\bar{\mathbf{w}}_T$ *is randomly chosen from* $\{\mathbf{w}_0, \mathbf{w}_1, \cdots, \mathbf{w}_{T-1}\}$ *according to a probability distribution which is related to the delay-adaptive learning rates as shown in (17).*

*Proof.* According to Lemma 9, we can get that

$$
\begin{aligned}
&\mathbb{E}F(\hat{\mathbf{y}}_T) - \mathbb{E}F(\hat{\mathbf{y}}_1) \\
&\leq \sum_{t=1}^{T-1}\left[\left(\frac{L(\hat{\eta}_{k_{t-1},t})^2}{2(1-\beta)^2} - \frac{\hat{\eta}_{k_{t-1},t}}{2(1-\beta)}\right)\mathbb{E}\left\|\nabla F(\mathbf{w}_t)\right\|^2\right] + \frac{L\sigma^2}{2(1-\beta)^2}\sum_{t=1}^{T-1}(\hat{\eta}_{k_{t-1},t})^2 \\
&\quad + \mathbb{E}\left[\frac{L^2}{1-\beta}\sum_{t=1}^{T-1}\left(\hat{\eta}_{k_{t-1},t}\left\|\hat{\mathbf{w}}_t - \mathbf{w}_t\right\|^2 + \hat{\eta}_{k_{t-1},t}\left\|\hat{\mathbf{y}}_t - \hat{\mathbf{w}}_t\right\|^2\right)\right] \\
&\leq \sum_{t=1}^{T-1}\left[\left(\frac{L(\hat{\eta}_{k_{t-1},t})^2}{2(1-\beta)^2} - \frac{\hat{\eta}_{k_{t-1},t}}{2(1-\beta)}\right)\mathbb{E}\left\|\nabla F(\mathbf{w}_t)\right\|^2\right] + \frac{L\sigma^2}{2(1-\beta)^2}\sum_{t=1}^{T-1}(\hat{\eta}_{k_{t-1},t})^2 \\
&\quad + \frac{\eta L\sigma^2}{2(1-\beta)^3}\left(\sum_{t=1}^{T-1}\hat{\eta}_{k_{t-1},t} + \sum_{k\in[K]}\hat{\eta}_{k,0}\right) \\
&\quad + \frac{\eta KL}{2(1-\beta)^3}\left[\sum_{t=1}^{T-1}\hat{\eta}_{k_{t-1},t}\mathbb{E}\left\|\nabla F(\mathbf{w}_t)\right\|^2 + \left(\sum_{k\in[K]}\hat{\eta}_{k,0}\right)\left\|\nabla F(\mathbf{w}_0)\right\|^2\right] \\
&\quad + \frac{2\beta^2\eta K^2 L^2}{(1-\beta)^5}\left[\sum_{k\in[K]}(\hat{\eta}_{k,0})^2\left\|\nabla F(\mathbf{w}_0)\right\|^2 + \sum_{t=1}^{T-1}(\hat{\eta}_{k_{t-1},t})^2\mathbb{E}\left\|\nabla F(\mathbf{w}_t)\right\|^2\right] \\
&\quad + \frac{2\beta^2\eta KL^2\sigma^2}{(1-\beta)^4}\left[\sum_{k\in[K]}(\hat{\eta}_{k,0})^2 + \sum_{t=1}^{T-1}(\hat{\eta}_{k_{t-1},t})^2\right] \\
&\leq \sum_{t=1}^{T-1}\left[\left(\frac{L(\hat{\eta}_{k_{t-1},t})^2}{2(1-\beta)^2} - \frac{\hat{\eta}_{k_{t-1},t}}{2(1-\beta)}\right)\mathbb{E}\left\|\nabla F(\mathbf{w}_t)\right\|^2\right] + \frac{L\sigma^2}{2(1-\beta)^2}\sum_{t=1}^{T-1}(\hat{\eta}_{k_{t-1},t})^2 \\
&\quad + \left(\frac{\eta L\sigma^2}{2(1-\beta)^3} + \frac{\eta\beta^2 L\sigma^2}{4(1-\beta)^2}\right)\left(\sum_{t=1}^{T-1}\hat{\eta}_{k_{t-1},t} + \sum_{k\in[K]}\hat{\eta}_{k,0}\right) \\
&\quad + \frac{\eta KL}{2(1-\beta)^3}\left[\sum_{t=1}^{T-1}\hat{\eta}_{k_{t-1},t}\mathbb{E}\left\|\nabla F(\mathbf{w}_t)\right\|^2 + \sum_{k\in[K]}\hat{\eta}_{k,0}\left\|\nabla F(\mathbf{w}_0)\right\|^2\right] \\
&\quad + \frac{\beta^2\eta KL}{4(1-\beta)^3}\left[\sum_{k\in[K]}\hat{\eta}_{k,0}\left\|\nabla F(\mathbf{w}_0)\right\|^2 + \sum_{t=1}^{T-1}\hat{\eta}_{k_{t-1},t}\mathbb{E}\left\|\nabla F(\mathbf{w}_t)\right\|^2\right]
\end{aligned}
$$

$$\leq \sum_{t=1}^{T-1}\left[\left(\frac{L(\hat{\eta}_{k_{t-1},t})^2}{2(1-\beta)^2}-\frac{\hat{\eta}_{k_{t-1},t}}{2(1-\beta)}\right)\mathbb{E}\left\|\nabla F(\mathbf{w}_t)\right\|^2\right]+\frac{L\sigma^2}{2(1-\beta)^2}\sum_{t=1}^{T-1}(\hat{\eta}_{k_{t-1},t})^2$$

$$+\frac{\eta L\sigma^2}{(1-\beta)^3}\left(\sum_{t=1}^{T-1}\hat{\eta}_{k_{t-1},t}+\sum_{k\in[K]}\hat{\eta}_{k,0}\right)$$

$$+\frac{\eta KL}{(1-\beta)^3}\left[\sum_{t=1}^{T-1}\hat{\eta}_{k_{t-1},t}\mathbb{E}\left\|\nabla F(\mathbf{w}_t)\right\|^2+\sum_{k\in[K]}\hat{\eta}_{k,0}\left\|\nabla F(\mathbf{w}_0)\right\|^2\right]$$

$$\leq \sum_{t=1}^{T-1}\left[\left(\frac{L(\hat{\eta}_{k_{t-1},t})^2}{2(1-\beta)^2}-\frac{\hat{\eta}_{k_{t-1},t}}{2(1-\beta)}\right)\mathbb{E}\left\|\nabla F(\mathbf{w}_t)\right\|^2\right]+\frac{L\sigma^2}{2(1-\beta)^2}\sum_{t=1}^{T-1}(\hat{\eta}_{k_{t-1},t})^2$$

$$+\frac{\eta L\sigma^2}{(1-\beta)^3}\left(\sum_{t=1}^{T-1}\hat{\eta}_{k_{t-1},t}+\sum_{k\in[K]}\hat{\eta}_{k,0}\right)$$

$$+\frac{1}{4(1-\beta)}\left[\sum_{t=1}^{T-1}\hat{\eta}_{k_{t-1},t}\mathbb{E}\left\|\nabla F(\mathbf{w}_t)\right\|^2+\sum_{k\in[K]}\hat{\eta}_{k,0}\left\|\nabla F(\mathbf{w}_0)\right\|^2\right]$$

$$\mathbb{E}F(\hat{\mathbf{y}}_1)\leq F(\mathbf{w}_0)-\frac{1}{1-\beta}\mathbb{E}\langle\nabla F(\mathbf{w}_0),\left(\sum_{k\in[K]}\hat{\eta}_{k,0}\mathbf{g}_0^k\right)\rangle+\frac{L}{2(1-\beta)^2}\mathbb{E}\left\|\sum_{k\in[K]}\hat{\eta}_{k,0}\mathbf{g}_0^k\right\|^2$$

$$=F(\mathbf{w}_0)-\frac{\sum_{k\in[K]}\hat{\eta}_{k,0}}{1-\beta}\left\|\nabla F(\mathbf{w}_0)\right\|^2$$

$$+\frac{L}{2(1-\beta)^2}\mathbb{E}\left\|\sum_{k\in[K]}\hat{\eta}_{k,0}\left(\mathbf{g}_0^k-\nabla F(\mathbf{w}_0)+\nabla F(\mathbf{w}_0)\right)\right\|^2$$

$$\leq F(\mathbf{w}_0)+\left[\frac{L\left(\sum_{k\in[K]}\hat{\eta}_{k,0}\right)^2}{2(1-\beta)^2}-\frac{\sum_{k\in[K]}\hat{\eta}_{k,0}}{1-\beta}\right]\left\|\nabla F(\mathbf{w}_0)\right\|^2+\frac{L\sigma^2}{2(1-\beta)^2}\sum_{k\in[K]}(\hat{\eta}_{k,0})^2$$

$$\mathbb{E}F(\hat{\mathbf{y}}_T)-F(\mathbf{w}_0)$$

$$\leq \sum_{t=1}^{T-1}\left[\left(\frac{L(\hat{\eta}_{k_{t-1},t})^2}{2(1-\beta)^2}-\frac{\hat{\eta}_{k_{t-1},t}}{2(1-\beta)}\right)\mathbb{E}\left\|\nabla F(\mathbf{w}_t)\right\|^2\right]+\frac{L\sigma^2}{2(1-\beta)^2}\sum_{t=1}^{T-1}(\hat{\eta}_{k_{t-1},t})^2$$

$$+\frac{\eta L\sigma^2}{(1-\beta)^3}\left(\sum_{t=1}^{T-1}\hat{\eta}_{k_{t-1},t}+\sum_{k\in[K]}\hat{\eta}_{k,0}\right)$$

$$+\frac{1}{4(1-\beta)}\left[\sum_{t=1}^{T-1}\hat{\eta}_{k_{t-1},t}\mathbb{E}\left\|\nabla F(\mathbf{w}_t)\right\|^2+\sum_{k\in[K]}\hat{\eta}_{k,0}\left\|\nabla F(\mathbf{w}_0)\right\|^2\right]$$

$$+\left[\frac{L\left(\sum_{k\in[K]}\hat{\eta}_{k,0}\right)^2}{2(1-\beta)^2}-\frac{\sum_{k\in[K]}\hat{\eta}_{k,0}}{1-\beta}\right]\left\|\nabla F(\mathbf{w}_0)\right\|^2+\frac{L\sigma^2}{2(1-\beta)^2}\sum_{k\in[K]}(\hat{\eta}_{k,0})^2$$

$$\leq \sum_{t=1}^{T-1}\left[\left(\frac{L(\hat{\eta}_{k_{t-1},t})^2}{2(1-\beta)^2}-\frac{\hat{\eta}_{k_{t-1},t}}{4(1-\beta)}\right)\mathbb{E}\left\|\nabla F(\mathbf{w}_t)\right\|^2\right]+\frac{2\eta L\sigma^2}{(1-\beta)^3}\left(\sum_{t=1}^{T-1}\hat{\eta}_{k_{t-1},t}+\sum_{k\in[K]}\hat{\eta}_{k,0}\right)$$

$$+\left(\frac{L\left(\sum_{k\in[K]}\hat{\eta}_{k,0}\right)^2}{2(1-\beta)^2}-\frac{\sum_{k\in[K]}\hat{\eta}_{k,0}}{4(1-\beta)}\right)\left\|\nabla F(\mathbf{w}_0)\right\|^2$$

$$\mathbb{E}F(\hat{\mathbf{y}}_T) - F(\mathbf{w}_0) \leq -\frac{1}{8\left(1-\beta\right)} \left[ \sum_{t=1}^{T-1} \hat{\eta}_{k_{t-1},t} \mathbb{E}\left\|\nabla F(\mathbf{w}_t)\right\|^2 + \sum_{k\in[K]} \hat{\eta}_{k,0}\left\|\nabla F(\mathbf{w}_0)\right\|^2 \right]$$

$$+ \frac{2\eta L\sigma^2}{(1-\beta)^3}\left( \sum_{t=1}^{T-1} \hat{\eta}_{k_{t-1},t} + \sum_{k\in[K]} \hat{\eta}_{k,0} \right)$$

Thus, we can get that

$$\frac{1}{\sum_{t=0}^{T-1}\bar{\eta}_t}\mathbb{E}\left( \sum_{t=0}^{T-1} \bar{\eta}_t\left\|\nabla F(\mathbf{w}_t)\right\|^2 \right) \leq \frac{8\left(1-\beta\right)\left(F(\mathbf{w}_0)-F^*\right)}{\sum_{t=0}^{T-1}\bar{\eta}_t} + \frac{16\eta L\sigma^2}{(1-\beta)^2},$$

where $\bar{\eta}_0 = \sum_{k\in[K]} \hat{\eta}_{k,0}$ and $\bar{\eta}_t = \hat{\eta}_{k_{t-1},t}(t\geq 1)$.

Then, we analyse the lower bound of $\sum_{t=0}^{T-1}\bar{\eta}_t$.

It's easy to verify that

$$\hat{\tau}_{k,t+1} = t + 1 - ite(k,t+1) = \begin{cases} t - ite(k,t) + 1 = \hat{\tau}_{k,t} + 1 & k \neq k_t, \\ 0 = \hat{\tau}_{k,t} - \tau_t & k = k_t. \end{cases}$$

Since $\sum_{k\in[K]} \hat{\tau}_{k,0} = 0$, we have $\sum_{k\in[K]} \hat{\tau}_{k,T} + \sum_{t=0}^{T-1} \tau_t = (K-1)T$. Moreover, $\hat{\tau}_{k_{T-1},T} = T - ite(k_{T-1},T) = 0$. We get that $\sum_{k\in[K],k\neq k_{T-1}} \hat{\tau}_{k,T} + \sum_{t=0}^{T-1} \tau_t = (K-1)T$. Thus, at least $\frac{T}{2}$ delays are smaller than $2K$. $\sum_{t=0}^{T} \bar{\eta}_t = \sum_{t=1}^{T-1} \hat{\eta}_{k_{t-1},t} + \sum_{k\in[K]} \hat{\eta}_{k,0} \geq \frac{T\eta}{2}$.

$$\frac{1}{\sum_{t=0}^{T-1}\bar{\eta}_t} \sum_{t=0}^{T-1} \bar{\eta}_t\mathbb{E}\left\|\nabla F(\mathbf{w}_t)\right\|^2 \leq \frac{16(1-\beta)}{T\eta}\left(F(\mathbf{w}_0)-F^*\right) + \frac{16\eta L\sigma^2}{(1-\beta)^2}.$$

If we choose an output $\bar{\mathbf{w}}_T$ from $\{\mathbf{w}_0, \mathbf{w}_1, \cdots, \mathbf{w}_{T-1}\}$ according to

$$\mathbb{P}(\bar{\mathbf{w}}_T = \mathbf{w}_t) \propto \bar{\eta}_t, \tag{17}$$

where $t \in [T]$ and let $\eta = \min\{\frac{(1-\beta)^2}{8KL}, \sqrt{\frac{(1-\beta)^3(F(\mathbf{w}_0)-F^*)}{TL\sigma^2}}\}$, we have that

$$\mathbb{E}\left\|\nabla F(\bar{\mathbf{w}}_T)\right\|^2 \leq \mathcal{O}\left( \sqrt{\frac{L\sigma^2}{T}} + \frac{KL}{T} \right).$$

$\square$

