# OpenReview forum: "Ordered Momentum for Asynchronous SGD"
_NeurIPS.cc/2024/Conference — NeurIPS 2024 poster_

### Official Review · Reviewer_dA6E · 2024-06-29

**Soundness:** 2
**Presentation:** 3
**Contribution:** 3
**Rating:** 5
**Confidence:** 3

**Summary:**

In distributed learning environments, asynchronous SGD (ASGD) and its variants are often used to deal with computing nodes with uneven computing power. Momentum methods are beneficial for both optimization and generalization in deep model training, but directly applying momentum to ASGD may hinder its convergence. The paper proposes the OrMo method to integrate momentum into ASGD by sorting gradients based on iteration index. The paper provides a theoretical proof of the convergence of OrMo on non-convex problems.

**Strengths:**

1. A new asynchronous stochastic gradient descent optimization method (OrMo) is proposed, and a theoretical proof of convergence on non-convex problems is provided, which is the first time that it does not rely on the bounded delay assumption.
2. Experiments show the advantages of the OrMo method in convergence performance.
3. The paper provides detailed algorithm implementation details and open access links to the experimental code, which enhances the reproducibility.
4. The experiment setting is clear, and the wall-clock time shows that the OrMo outperforms baseline methods significantly.

**Weaknesses:**

1. Although experimental validation was performed on the CIFAR dataset, the generalization ability of the OrMo method may need to be further tested in more types of datasets and different application scenarios.
2. The paper does not discuss in detail the computational resource requirements of the OrMo method on data and models of different scales. Specifically, the ordered momentum need to be cached on the server, the extra storage costs should be considerred.
3. The proposed re-ordering method inherently is like the staleness control. The difference is that the OrMo proposes using the exponentioal term on \beta to control the staleness of the incoming gradients.

**Questions:**

Figure 3 shows that the curve is very unstable. So, on which epoch, the model is used as the final model for testing?

---

> ### Author Rebuttal · Authors · 2024-08-07
>
> We sincerely thank the reviewer for the constructive comments and the support of our work. Below, we respond to the raised concerns and questions point by point.
>
> **Response to Weakness 1:**
>
> Thank you for your valuable feedback and suggestions regarding the generalization ability of the OrMo method. We acknowledge the importance of testing OrMo on a wider range of datasets and in various application scenarios to fully understand its generalization capabilities.
>
> We conduct an additional experiment by training a Vision Transformer [[1](https://github.com/lucidrains/vit-pytorch/blob/main/vit_pytorch/vit_for_small_dataset.py)] model on CIFAR10 dataset. The experiments are conducted on TITAN XP GPUs. The hyperparameter settings are following that in the submitted paper. Due to current constraints in time and computational resources, the experiments are still ongoing.  More experimental results will be uploaded as soon as possible. The following table shows the test accuracy of OrMo and SMEGA$^2$ in the homogeneous setting when there are $32$ workers.
>
> |                   | SMEGA$^2$                 | OrMo  |
> | -- |--|--|
> |K=32 (homogeneous) | 60.67\%  | 61.32\%  |
>
>
> We appreciate your understanding and hope that the current results still offer valuable insights into the potential of OrMo.
>
> **Response to Weakness 2:**
>
> Compared with vanilla ASGD, the only additional computational resource for OrMo is the storage cost for the momentum on the server. As a momentum method, the extra storage cost for the momentum in OrMo is inevitable. The total storage complexity for the momentum in OrMo is $\mathcal{O}(d)$, where $d$ is the dimension of the momentum.
>
> Moreover, OrMo only maintains one momentum on the server. In contrast, shifted momentum [A] maintains one momentum on each worker. Therefore, the total storage complexity $\mathcal{O}(d)$ for the momentum in OrMo is much less than the $\mathcal{O}(Kd)$ complexity of shifted momentum [A], where $d$ is the dimension of the momentum and $K$ is the number of workers.
>
> **Response to Weakness 3:**
>
> The update rule in OrMo is motivated by that in SSGDm. The key insight of the update rule of the momentum and parameter in OrMo lies in tracking the sequences $\hat{\bf w}_t$ and $\hat{\bf u}_t$ defined in Subsection 3.3, which are updated in a manner similar to SSGDm. OrMo cannot be simply regarded as a staleness control method. Due to limited space here, please refer to paragraphs 2-5 in our Author Rebuttal for the details.
>
> **Response to Question 1:**
>
> The main reason for the instability of the curves in Figure $3$ is that naive ASGDm occasionally fails to converge, as shown in Table $1$. In contrast, our proposed method OrMo demonstrates stable convergence across all experiments. We use the model obtained after the last epoch (i.e., $160$-th epoch for CIFAR10 and $200$-th epoch for CIFAR100) as the final model for testing.
>
> We hope that we have addressed the reviewer's concerns, and we are always willing to respond to any further concerns. Meanwhile, we would greatly appreciate it if the reviewer could re-evaluate our work based on our response.
>
> [A] Giladi et al., At Stability's Edge: How to Adjust Hyperparameters to Preserve Minima Selection in Asynchronous Training of Neural Networks? ICLR 2020.

---

> > ### Comment · Reviewer_dA6E · 2024-08-09
> > **Thanks for responses**
> >
> > I have two follow-up comments:
> > - 1. Why do you train ViT on CIFAR-10, instead of ResNet? The ResNet 18 trained with SGD momentum can obtain round 94% test accuracy, which should be the most standard baseline to compare optimizers with SGD momentum.
> > - 2. While the derivation of how to adjust the momentum is complex, the core of it is still like staleness control and a penalty function.

---

> > > ### Author Response · Authors · 2024-08-11
> > >
> > > We sincerely thank you for your timely response. We would like to answer the questions point by point as follows:
> > >
> > > 1. In the maintext of our submission, we have presented the empirical results of training a ResNet20 model on the CIFAR-10 dataset. More experimental results about ResNet20 are given in the response to Reviewer NjVh.
> > > Meanwhile, we promise to add the experimental results of training a Vision Transformer model on CIFAR-10 in our initial responses. The number of workers is set as $32$. The additional experiments are finished now. We would like to present the results below. Each experiment is repeated 3 times.
> > >
> > >     |Test Accuracy      |  (hom.)  | (het.)      |
> > >     | -------|--------|---------|
> > >     |ASGD             | 56.09\% $\pm$ 0.41\% | 56.33\% $\pm$  0.56\% |
> > >     |naive ASGDm      | 55.66\% $\pm$ 0.15\% | 55.74\% $\pm$  0.35\% |
> > >     |shifted momentum | 57.43\% $\pm$ 0.28\% | 57.39\% $\pm$  0.47\% |
> > >     |SMEGA$^2$        | 60.54\% $\pm$ 0.05\% | 60.66\% $\pm$  0.12\% |
> > >     |OrMo             | **61.21\% $\pm$ 0.17\%** | **61.30\% $\pm$  0.11\%** |
> > >
> > >
> > >     We have also tested the performance of the OrMo method on the ResNet18 model, as you mentioned, over the past three days. Compared to the above Vision Transformer model which has about $0.5$ million parameters, ResNet18 has $10$ million parameters. OrMo achieves around 94% test accuracy on the ResNet18 model, as shown in the table below. Please note that while shifted momentum performs well in terms of final test accuracy, it converges more slowly than OrMo. Specifically, OrMo requires approximately 30 epochs to reach 85% test accuracy, whereas shifted momentum requires 60 epochs. Due to format constraints in the discussion period, we are unable to include figures in this response. We will provide more details (e.g., training curve figures) in the final version. These additional results are consistent with those in the main text and further support the conclusions of our work.
> > >
> > >     |Test Accuracy      |  (hom.)  | (het.)      |
> > >     | -------|--------|---------|
> > >     |ASGD             | 91.45\% | 91.52\% |
> > >     |naive ASGDm      | 93.74\% | 93.10\% |
> > >     |shifted momentum | 94.02\% | 94.20\% |
> > >     |SMEGA$^2$        | 93.72\% | 93.36\% |
> > >     |OrMo             | **94.32\%** | **94.26\%** |
> > >
> > > 2. We sincerely thank you for the valuable comment, which makes us rethink about our method. We have provided a discussion about the differences between OrMo and existing penalty function methods in the general response, and we will add the discussion in the final version. Meanwhile, we would like to briefly summarize the differences between OrMo and existing methods that use a penalty function below.
> > >
> > >     +  For penalty function methods, the key insight is to reduce the contribution of gradients with larger delays to the parameter update.
> > >     For OrMo, the key insight lies in tracking the sequences $\hat{\bf w}_t$ and $\hat{\bf u}_t$ defined in Subsection 3.3, which are updated in a manner similar to SSGDm.
> > >
> > >     + In penalty function methods, gradients with larger delays are given smaller weights when updating the parameter.
> > >     In OrMo, gradients with larger delays are given smaller weights when updating the momentum, but larger weights when updating the parameter. Both theoretical derivations and empirical ablation studies highlight the importance of giving larger weights to gradients with larger delays when updating the parameter
> > >
> > > For more details, please refer to our general responses.
> > >
> > > Thank you again for letting us know your remaining concerns. We are always willing to answer any further questions.

---

> ### Comment · Reviewer_dA6E · 2024-08-13
> **Thanks for explanations**
>
> Thanks for the efforts and explanations. As illustrated by authors,
> > In penalty function methods, gradients with larger delays are given smaller weights when updating the parameter. In OrMo, gradients with larger delays are given smaller weights when updating the momentum, but larger weights when updating the parameter.
>
> Thus, the main differences between the penalty and the OrMo is the updating object, i.e., parameter v.s. momentum. Inherently, the core idea is similar, as adjusting the weights according to the staleness. In light of this, I think the claim that the OrMo is not a staleness-control method is not appropriate. Maybe authors should reconsider this claim. Nevertheless, I appreciate the idea of integrating momentum into ASGD by sorting gradients based on iteration index. Thus, I'd like to keep the positive score.

---

> > ### Author Response · Authors · 2024-08-13
> >
> > We greatly appreciate the insightful follow-up comments, which helps us rethink about our work. We agree that OrMo can be considered as another type of staleness-control methods since the weights in OrMo are depended on the staleness. We will add the discussion in the final version and sincerely thank you for your support of our work.

---

### Official Review · Reviewer_5XzJ · 2024-07-02

**Soundness:** 3
**Presentation:** 2
**Contribution:** 3
**Rating:** 5
**Confidence:** 4

**Summary:**

This paper proposed a new ordered momentum for asynchronous SGD based on the delayed update characteristic of asynchronous training, which weights the momentum according to the actual iteration index. The authors proved the convergence of the algorithm both theoretically and experimentally.

**Strengths:**

* This paper proposes a new momentum algorithm for ASGD and proves that the algorithm is convergent.

* Experiments are conducted in this paper to verify the effectiveness of the OrMo algorithm.

**Weaknesses:**

* This paper studies asynchronous training algorithms but does not explore the effect of asynchronous delays on convergence.

* The theoretical results in this paper are limited by the number of distributed workers $K$, i.e., the algorithm does not scale well to large-scale distributed training systems.

**Questions:**

1. The theoretical results in this paper do not contain asynchronous delay information and fail to reveal the effect of asynchronous delay on the convergence of the algorithm. This may be due to rough gradient upper bounds that obscure the delay gradient information. However, the reviewer believes that analyzing the effect of asynchronous delay is essential in a work that studies ASGD algorithms. Also, the experimental section did not test the empirical effect of different delays on the algorithm.

2. Theorem 1 shows that the convergence of the algorithm is linearly dependent on the number of distributed workers $K$, which implies that the algorithm will fail to converge when $K$ is large. In light of this theoretical result, can't the algorithm proposed in this paper be applied to large-scale distributed training systems?

3. The algorithm design and theoretical analysis in this paper relies on a fixed constant learning rate. Can this be extended to adaptive learning rates or decreasing learning rates that are more common in practice?

4. The result of Lemma 4 in the main text is inconsistent with the proof in the Appendix.

**Limitations:**

Please refer to Weaknesses and Questions.

---

> ### Author Rebuttal · Authors · 2024-08-07
>
> We sincerely thank the reviewer for the constructive comments and the support of our work. Below, we respond to the raised concerns and questions point by point.
>
> **Response to Question 1 (Weakness 1):**
>
> This is a very good question. In fact, a core contribution of this paper is exactly the theoretical guarantee for OrMo, which doesn't contain the asynchronous delay information $\tau_{max}$ (the maximum delay). In contrast, previous works, such as [B], show that ASGD has a convergence rate of $\mathcal{O}(\frac{\sigma}{\sqrt{T}}+\frac{\tau_{max}}{T})$. However, this convergence result containing the maximum delay $\tau_{max}$ is not consistent with the actual convergence performance, since vanilla ASGD can converge well when $\tau_{max}$ is extremely large in practice.
>
> The most closely related work to ours is [A], which analyzes vanilla ASGD with unbounded gradient delays. We extend the analysis for vanilla ASGD in [A] to OrMo. Our analysis shows that the convergence of OrMo doesn't depend on the maximum delay $\tau_{max}$. Our experimental results offer additional insights into the convergence rate. In the following table, we show the maximum delays of two different settings in CIFAR10 training when $K=64$. In heterogeneous settings, the existence of slow workers results in a maximum delay that exceeds that of the homogeneous setting by about 100 times. Our experiments show that OrMo can achieve comparable performances in these two settings, which aligns with our theoretical analysis results.
>
> |   | homogeneous |heterogeneous|
> | --- |--| --|
> | Maximum Delay       |    343  | 30911 |
> | Training Loss      |    0.15 $\pm$ 0.02  | 0.16 $\pm$ 0.03 |
> | Test Accuracy      |    88.03% $\pm$ 0.28%  | 87.76% $\pm$ 0.57% |
>
> Our analysis provides an upper bound for the convergence rate of OrMo and shows that the maximum delay doesn't affect the convergence rate.
>
> **Response to Question 2 (Weakness 2):**
>
> In our analysis, OrMo achieves a convergence rate of $\mathcal{O} (\frac{\sigma}{\sqrt{T}}+ (\frac{K}{T})^{\frac{2}{3}} + \frac{K}{T})$. Our convergence result aligns with the theoretical results in Theorem $2$ of [A], which are currently the tightest convergence analyses for vanilla ASGD. This convergence rate indicates that OrMo has a comparable convergence rate to SGD for non-convex problems when $T$ is sufficiently large, as the term $\frac{\sigma}{\sqrt{T}}$ becomes dominant and the non-dominant terms $(\frac{K}{T})^{\frac{2}{3}}+\frac{K}{T}$ have a minimal impact on the overall convergence rate. Thus, OrMo can be applied for large-scale algorithms.
>
> In existing distributed optimization works (e.g., [C][D][E]), the non-dominant terms in their theoretical results are also constrained by the number of distributed workers. We take the convergence analysis for Gossip-PGA in [E] as an example. According to Subsection $1.1$ of [E], Gossip-PGA has a convergence rate of $\mathcal{O} (\frac{\sigma}{\sqrt{K\tilde{T}}} + \frac{1}{\tilde{T}^{\frac{2}{3}}}+\frac{1}{\tilde{T}})$, where $\tilde{T}$ is the number of iterations. Please note that Gossip-PGA is a synchronous algorithm, which requires $K$ gradients on all workers for one update iteration of the parameter. In contrast, one update iteration of the parameter in OrMo requires only one gradient from a single worker. For fair comparison, we rewrite the convergence rate of Gossip-PGA in terms of the number of the gradient computations: $\mathcal{O} (\frac{\sigma}{\sqrt{C}} + \frac{K^{\frac{2}{3}}}{C^{\frac{2}{3}}}+\frac{K}{C})$, where $C$ is the number of the gradient computations and $C = K\tilde{T}$. The convergence rate of OrMo in terms of the number of the gradient computations can be written as $\mathcal{O} (\frac{\sigma}{\sqrt{C}} + \frac{K^{\frac{2}{3}}}{C^{\frac{2}{3}}}+\frac{K}{C})$. We can observe that the non-dominant terms $\mathcal{O}(\frac{K^{\frac{2}{3}}}{C^{\frac{2}{3}}}+\frac{K}{C})$ in [E] and OrMo are both constrained by the number of the workers.
>
> **Response to Question 3:**
>
> We sincerely thank you for the constructive comment. We agree that incorporating adaptive learning rates into asynchronous methods is an important research direction. Due to limited space here, please refer to paragraphs 5-6 of our Author Rebuttal for more discussion about the adaptive learning rates. The theoretical analysis for a constant learning rate is a common setting in existing distributed optimization works [C][D][E]. Actually, the stage-wise decreasing learning rate in our experiment can be considered as a combination of constant learning rates and the widely-used restarting technique [F][G].
>
> We promise to add the statements above in the final version.
>
> **Response to Question 4:**
>
> We sincerely thank you for your careful review. There is a typo in Lemma 4 in the maintext. The right-hand side of the inequality in Lemma 4 should be $2\frac{\beta^2}{(1-\beta)^4}\eta^2K^2G^2$. We will fix the typo in the final version.
>
> We sincerely thank the reviewers for their valuable time and their support of our work again. Meanwhile, we would greatly appreciate it if the reviewer could re-evaluate our work in light of our response.
>
> [A] Mishchenko et al., Asynchronous SGD Beats Minibatch SGD Under Arbitrary Delays. NeurIPS 2022.
>
> [B] Stich et al., The Error-Feedback framework: SGD with Delayed Gradients. JMLR 2020.
>
> [C] Xie et al., CSER: Communication-efficient SGD with Error Reset. NeurIPS 2020.
>
> [D] Yu et al., On the Linear Speedup Analysis of Communication Efficient Momentum SGD for Distributed Non-Convex Optimization. ICML 2019.
>
> [E] Chen et al., Accelerating Gossip SGD with Periodic Global Averaging. ICML 2021.
>
> [F] Powell et al., Restart procedures for the conjugate gradient method. Mathematical Programming 1977.
>
> [G] Li et al., Restarted Nonconvex Accelerated Gradient Descent: No More Polylogarithmic Factor in the in the O(epsilon^(-7/4)) Complexity. JMLR 2023.

---

> > ### Comment · Reviewer_5XzJ · 2024-08-11
> >
> > Thanks to the authors for their responses. I currently have no further questions.

---

> > > ### Author Response · Authors · 2024-08-14
> > >
> > > Thank you for the response. We are grateful for your support of our work.

---

### Official Review · Reviewer_M3Nx · 2024-07-11

**Soundness:** 2
**Presentation:** 1
**Contribution:** 2
**Rating:** 3
**Confidence:** 2

**Summary:**

The paper introduces Ordered Momentum (OrMo), a novel method enhancing the performance of ASGD by systematically incorporating momentum based on the iteration indexes of gradients. The authors provide theoretical proofs demonstrating the convergence of OrMo for non-convex problems, marking an advancement as this is purportedly the first such convergence analysis for ASGD with momentum that does not rely on the bounded delay assumption. Empirical results further validate that OrMo outperforms standard ASGD and other asynchronous variants with momentum in terms of convergence rates.

**Strengths:**

1. The proposed OrMo algorithm addresses the common problem of momentum integration in asynchronous settings, which is often associated with convergence issues.
2. The paper provides a convergence analysis of OrMo in non-convex settings without the need for a bounded delay assumption.

**Weaknesses:**

1. The writing and presentation of the paper seems to require substantial improvement. Especially for Section 3, it is hard to fully follow the algorithmic and theoretical details, making it laborious to check the correctness.
2. The paper considers only scenarios with homogeneous data distributions, where the workers access to a shared dataset $\mathcal{D}$.
3. Assumption 1 includes a bounded gradient assumption, which is rather demanding.
4. The experiment setup does not quite match the theoretical analysis. In the experiment, the learning rate is multiplied by a factor after a specific number of epochs, while Theorem 1 is based on a constant learning rate. There also lacks discussion on how the multiplication factors and the intervals are tuned.

**Questions:**

1. What is the impact of the size of the waiting set $\mathcal{C}$? How to choose the size of $\mathcal{C}$?
2. What are the key insights that the convergence can be guaranteed without bounded delay?
3. What is the complexity and cost in terms of the ordering of gradients?

**Limitations:**

The authors have partially addressed the limitations.

---

> ### Author Rebuttal · Authors · 2024-08-07
>
> We sincerely thank the reviewer for the constructive comments. Below, we respond to the raised concerns and questions point by point.
>
> **Response to Weakness 1:**
>
> In Section 3, we first propose a new reformulation of SSGDm, which serves as the inspiration for designing OrMo for ASGD. We then present the details of OrMo for ASGD, including the algorithm and convergence analysis. Thanks for your valuable feedback. We will refine the writing and presentation of this section.
>
> **Response to Weakness 2:**
>
> Thank you for the constructive comment. Our analysis can be easily extended to heterogeneous data distributions with modified assumptions:
>
> Assumption 1' : For $\forall {\bf w} \in \mathbb{R}^d, \forall k \in [K]$,
> $\mathbb{E}\_{\xi^k}[\nabla f({\bf w}; \xi^k)]=\nabla F\_k({\bf w}),$ $\mathbb{E}\_{\xi^k}\|\nabla f({\bf w}; \xi^k)-\nabla F\_k({\bf w})\|^2 \leq \sigma^2,$ and $ \mathbb{E}\_{\xi^k}\\|\nabla f({\bf w};\xi^k)\\|^2 \leq G^2.$
>
> Assumption 4 (heterogeneity): There exists $\zeta \geq 0$ such that $\\|\nabla F\_k({\bf w})-F({\bf w})\\|\leq \zeta^2, \forall {\bf w} \in \mathbb{R}^d, \forall k \in [K]$.
>
> OrMo has the following convergence rate for heterogeneous data distributions:
> $$\frac{1}{T}\sum\_{t=1}^T \mathbb{E}\\|\nabla F({\bf{w}}\_t)\\|^2 \leq \mathcal{O} (\frac{\sigma}{\sqrt{T}}+ (\frac{K}{T})^{\frac{2}{3}} + \frac{K}{T} + \zeta^2).$$
>
> Compared to the homogeneous setting, the convergence rate in the heterogeneous setting includes an additive term $\zeta^2$, where the dependence on $\zeta^2$ is unavoidable without additional assumptions in asynchronous methods. This result is also consistent with existing works [A].
>
>
> **Response to Weakness 3:**
>
> Thank you for pointing out the demanding nature of the bounded gradient assumption in Assumption $1$. We agree that the bounded gradient assumption is a little stronger compared to the other assumptions. However, we think it is acceptable since the bounded gradient assumption is widely used in theoretical analyses in existing distributed optimization works (e.g., [B] [C]). And the bounded gradient assumption is also aligned with our empirical experience during model training. We will add the discussion above in the final version and thank you for the valuable comment again.
>
> **Response to Weakness 4:**
>
> Thank you for the valuable comments. Actually, the stage-wise decreasing learning rate in our experiment can be considered as a combination of constant learning rates and the widely-used restarting technique [D] [E].
>
> In our experiments, we only tune the initial learning rate while the multiplication factors and the intervals are fixed. In particular, the settings of the multiplication factors and intervals in our experiment are exactly the same as those in previous works ([F] for CIFAR10 and [B] for CIFAR100). These are commonly used configurations for CIFAR training.
>
> We sincerely apologize for not making it clear in the submission and promise to add the statements above in the final version.
>
> **Response to Question 1:**
>
> Thank you for the question. The waiting set $\mathcal{C}$ is the set of worker indexes that are waiting for the server to send the parameter. The size of the waiting set indicates the number of workers that are idle and waiting for the server to send the parameter. In other words, the size of the waiting set $\mathcal{C}$ is not a hyper-parameter and thus does not need to be set manually. This concept is introduced to unify SSGD and ASGD into a single framework in Algorithm $1$.
>
> **Response to Question 2:**
>
> We use the example in [A] to describe the insights. Suppose that there are two parallel workers: one fast worker that takes only $10^{-6}$ seconds to compute a stochastic gradient and one slow worker that takes $1$ second. If we implement ASGD with these two workers, the delay of the slow worker’s gradients will be $1$ million, since in the $1$ second that the server waits for the slower worker, the gradients from the fast worker will produce $1$ million updates. Consequently, analyses based on $\tau\_{max}$ degrades by a factor of $10^6$. However, ASGD actually performs very well in this scenario, since $99.9999\\%$ of the updates on the server use the stochastic gradients with no delay.
>
> Briefly speaking, slower workers which typically send heavily delayed gradients will engage in fewer training iterations.  And existing analyses for ASGD based on $\tau\_{max}$ aren't aligned with the actual performance.
>
> **Response to Question 3:**
>
> OrMo organizes the gradients into the momentum in order based on their iteration indexes. Since the momentum is a weighted sum of the gradients with predefined weights, each gradient is incorporated into momentum by multiplying it with a weight $\beta^{b\_{t+1} - \lceil \frac{ite(k\_t, t)}{K}\rceil}$. The complexity related to the ordering of the gradient is $\mathcal{O}(1)$, making the cost negligible.
>
> We hope that we have addressed the reviewer's concerns, and we are always willing to respond to any further concerns. Meanwhile, we would greatly appreciate it if the reviewer could re-evaluate our work based on our response.
>
> [A] Mishchenko et al., Asynchronous SGD Beats Minibatch SGD Under Arbitrary Delays. NeurIPS 2022.
>
> [B] Xie et al., CSER: Communication-efficient SGD with Error Reset. NeurIPS 2020.
>
> [C] Xu et al., Detached Error Feedback for Distributed SGD with Random Sparsification. ICML 2022.
>
> [D] Powell et al., Restart procedures for the conjugate gradient method. Mathematical Programming 1977.
>
> [E] Li et al., Restarted Nonconvex Accelerated Gradient Descent: No More Polylogarithmic Factor in the in the O(epsilon^(-7/4)) Complexity. JMLR 2023.
>
> [F] He et al., Deep Residual Learning for Image Recognition. CVPR 2016.

---

> > ### Comment · Reviewer_M3Nx · 2024-08-14
> >
> > I thank the authors for the response.
> >
> > The paper integrates momentum into ASGD. Compared to vanilla ASGD [1], this paper requires stronger assumption, while not improving the convergence rate. I am inclined to change my score to 4. As I have not carefully checked the technical details of the analysis, my confidence remains limited.
> >
> > [1] Mishchenko el al., Asynchronous SGD Beats Minibatch SGD Under Arbitrary Delays, NeurIPS 2022.

---

> > > ### Author Response · Authors · 2024-08-14
> > >
> > > We sincerely thank the reviewer for the willingness to increase the rating and the follow-up comments. Meanwhile, we would like to address the remaining concern below.
> > >
> > > **Q: The paper integrates momentum into ASGD. Compared to vanilla ASGD [1], this paper requires stronger assumption, while not improving the convergence rate.**
> > >
> > > + We would like to first clarify that momentum is widely used in distributed machine learning and has a good performance in many practical applications. There are several existing works that integrate momentum into ASGD [2][3][4]. However, as far as we know, there is almost no convergence analysis of these methods in existing works. In view of this challenge, we introduce the ordered momentum to ASGD and present the method OrMo, which is theoretically proven to be convergent and empirically performs better than existing methods.
> > > + Although the analysis of ASGD for adaptive learning rates in [1] does not require the bounded gradient assumption, the analysis of ASGD for constant learning rates in [1] is based on the bounded gradient assumption. Therefore, the analyses in our work and in [1] for constant learning rates are under the same assumptions ($L$-Lipschitz smoothness, lower-bounded objective, and bounded gradient).
> > > +  As presented in Remark 6 at the end of Section 3, OrMo can also be used with adaptive learning rates. We have discussed this point in the general response, and will further explore it in future work.
> > >
> > > Given the reasons above, we carefully think that there are adequate theoretical contributions in our work. We hope that our response can address the reviewer's concern. Meanwhile, we would greatly appreciate it if the reviewer could re-evaluate our work based on our response.
> > >
> > >
> > > [1] Mishchenko el al., Asynchronous SGD Beats Minibatch SGD Under Arbitrary Delays, NeurIPS 2022.
> > >
> > > [2] Giladi et al., At stability’s edge: How to adjust hyperparameters to preserve minima selection in asynchronous training of neural networks?, ICLR 2020.
> > >
> > > [3] Cohen et al., SMEGA2: distributed asynchronous deep neural network training with a single momentum buffer, ICPP 2022.
> > >
> > > [4]  Mitliagkas et al., Asynchrony begets momentum, with an application to deep learning. In Proceedings of the Annual Allerton Conference on Communication, Control, and Computing, 2016.

---

### Official Review · Reviewer_NjVh · 2024-07-18

**Soundness:** 3
**Presentation:** 3
**Contribution:** 2
**Rating:** 6
**Confidence:** 3

**Summary:**

This paper introduces OrMo, a novel method to weight stale gradients received on the server in asynchronous SGD with momentum. The algorithm is based on the idea of organizing the sequence of gradients received on the server into "buckets" to approximate standard minibatch SGD with momentum. The method is shown to converge under arbitrary (possibly unbounded) delays, with experiments on CIFAR-10 and CIFAR-100 validating the strength of OrMo compared to some baseline.

**Strengths:**

* **Convergence without bounded delay assumption:** Convergence is proved without relying on the bounded delay assumption, extending recent analysis of ASGD [[Mishchenko et al., 2022]](https://arxiv.org/pdf/2206.07638 ), [[Koloskova et al., 2022]](https://arxiv.org/pdf/2206.08307 ) to ASGD with momentum.
* **Clear explanation of the "momentum as a sum of weighted buckets" idea:** Time is spent to clearly lay down the ideas of "buckets" and re-writing the momentum updates so that OrMo comes naturally, which is helped by Fig.1 \& 2.
* **Promising experimental results:** Compared to naive implementation (e.g., plain ASGD with momentum) and some chosen baselines, OrMo is shown to perform better in practice, especially in cases where the workers speed is heterogeneous.

**Weaknesses:**

* **Lack of discussion between OrMo and its approximation of standard synchronous SGD with momentum:** While the idea of OrMo may seem natural when we visualize standard momentum as a "sum of weighted buckets of gradients", the update performed by OrMo simply *approximate* synchronous momentum. Indeed, as the parameters hosted on the server are updated at each stochastic gradient in the asynchronous setting, the *points* $w_t$ on which the gradients are computed all differ even inside a given "bucket", which is not the case for synchronous SGDm. Yet, this is not discussed.
* **Standard baselines are lacking in the experiments:** In Table 1\&2, and in Fig. 3\& 4, a comparison *(for reference only)* to synchronous SGDm could be interesting to have (to see whether or not there is still a gap to fill with synchronous methods in terms of "performance per iteration"). Moreover, it seems that the baselines "tuning the momentum value" for asynchronous SGD with momentum described in [[Zhang and Mitliagkas, 2018]]( https://arxiv.org/pdf/1706.03471 ) and [[Mitliagkas et al., 2016]]( https://arxiv.org/pdf/1605.09774 ) are lacking. Finally, we can see OrMo as a method to penalize stale gradients received by the server, using some sort of "exponential weight". Empirically, there have been other "penalty functions" that have been tried in Asynchronous Federated Learning, such as the ones described Part 5.2 of [[Xie et al., 2020]]( https://www.opt-ml.org/papers/2020/paper_28.pdf ). Have you compared OrMo to them?

**Questions:**

* In Theorem 1, the quantities $L,G$ do not appear in the convergence rate (contrary to what is displayed in Theorem 2 of [[Mishchenko et al., 2022]](https://arxiv.org/pdf/2206.07638 ) ). Is it normal?
* Fig.5: As the Parameter Server is known to become a bottleneck for communications at scale, how does OrMo fair compared to synchronous SGDm at larger scale than $K=16$? (for instance, $K=64$)

**Limitations:**

.

---

> ### Author Rebuttal · Authors · 2024-08-07
>
> We sincerely thank the reviewer for the constructive comments and the support of our work. Below, we respond to the raised concerns and questions point by point.
>
> **Response to Weakness 1:**
>
> As you pointed out, the gradients in SSGDm and OrMo are computed at different points due to the asynchronous nature of OrMo. In SSGDm, the set of the computed gradients can be expressed as $\\{{\bf g}\_0^0, \cdots, {\bf g}\_0^{K-1}, {\bf g}\_K^0, \cdots, {\bf g}\_K^{K-1}, {\bf g}\_{2K}^0, \cdots, {\bf g}\_{2K}^{K-1}, \cdots\\}$. In contrast, the set of the computed gradients in OrMo can be formulated as $\\{{\bf g}\_0^1, \cdots, {\bf g}\_0^{K-1}, {\bf g}\_1^{k\_0}, {\bf g}\_2^{k\_1}, {\bf g}\_3^{k\_2}, \cdots, {\bf g}\_K^{k\_{K-1}},\cdots\\}$. This difference between SSGDm and OrMo makes it challenging to directly measure how closely OrMo approximates SSGDm. To address this, in the theoretical analysis presented in Subsection 3.3, we define two auxiliary sequences, $\hat{\bf w}\_t$ and $\hat{\bf u}\_t$, for the parameter and the momentum, respectively. These sequences are updated in a manner similar to SSGDm but use the gradient set from OrMo instead of SSGDm's gradient set. *Lemma $1$ and Lemma $2$ rigorously formulate the differences between OrMo and the auxiliary sequences, providing a theoretical foundation for understanding the relationship between OrMo and SSGDm despite their differences in gradient computation.*
>
> Thanks for your very insightful comment which makes us think more deeply about the relationship and difference between our OrMo and its approximation of standard synchronous SGD with momentum. We will clarify this approximation in the final version.
>
>
> **Response to Weakness 2:**
>
> **2.1: A comparison (for reference only) to synchronous SGDm could be interesting to have.**
>
> The results are presented in the following table, and we will refine the figures in the final version.
>
> |CIFAR10            | K=16             | K=64  |
> | -- |--|--|
> |OrMo (hom.)        | 90.95\% $\pm$ 0.27\% | 88.03\% $\pm$ 0.28\% |
> |OrMo (het.)        | 91.01\% $\pm$ 0.10\% | 87.76\% $\pm$ 0.57\% |
> |SSGDm              | 90.85\% $\pm$ 0.26\% | 88.96\% $\pm$ 0.18\% |
>
> |CIFAR100            | K=16             | K=64  |
> | -|-|-|
> |OrMo (hom.)        | 67.56\% $\pm$ 0.34\% | 65.48\% $\pm$ 0.17\% |
> |OrMo (het.)        | 67.71\% $\pm$ 0.33\% | 65.43\% $\pm$ 0.35\% |
> |SSGDm              | 67.86\% $\pm$ 0.32\% | 66.43\% $\pm$ 0.50\% |
>
> **2.2:  Moreover, it seems that the baselines "tuning the momentum value" for asynchronous SGD with momentum are lacking.**
>
> Thanks for your insightful advice. Following the suggestion in [A], we conducted experiments to tune the momentum value $\beta$ for naive ASGDm and present the results on CIFAR10 here. While tuning the momentum value can enhance the performance of naive ASGDm, hyperparameter tuning is quite time-consuming and costly. In contrast, our method achieves better performance using the commonly used momentum value of 0.9, without requiring extensive tuning. We will include the momentum value tuning experiments in the final version.
> |Algorithm ($\beta$)           | K=16 (hom.)        | K=64 (hom.)        | K=16 (het.)        | K=64  (het.)       |
> | -|-|-|-|-|
> |naive ASGDm (0.1)             | 89.85\% $\pm$ 0.24\% | 83.93\% $\pm$ 0.25\% | 89.95\% $\pm$ 0.19\% | 83.76\% $\pm$ 0.34\% |
> |naive ASGDm (0.3)             | 89.91\% $\pm$ 0.20\% | 84.23\% $\pm$ 0.49\% | 90.26\% $\pm$ 0.05\% | 84.43\% $\pm$ 0.22\% |
> |naive ASGDm (0.6)             | 90.39\% $\pm$ 0.24\% | 83.87\% $\pm$ 0.28\% | 90.56\% $\pm$ 0.13\% | 84.07\% $\pm$ 0.38\% |
> |naive ASGDm (0.9)             | 88.15\% $\pm$ 1.70\% | 82.39\% $\pm$ 1.79\% | 73.23\% $\pm$ 31.61\%| 68.75\% $\pm$ 29.51\%|
> |OrMo (0.9)                    | **90.95\% $\pm$ 0.27\%** | **88.03\% $\pm$ 0.28\%** | **91.01\% $\pm$ 0.10\%** | **87.76\% $\pm$ 0.57\%** |
>
> **2.3: Finally, we can see OrMo as a method to penalize stale gradients $ \cdots $**
>
> The update rule in OrMo is motivated by that in SSGDm. The key insight of the update rule of the momentum and parameter in OrMo lies in tracking the sequences $\hat{\bf w}\_t$ and $\hat{\bf u}\_t$ defined in Subsection 3.3, which are updated in a manner similar to SSGDm. Hence, OrMo cannot be simply regarded as a method to penalize stale gradients received by the server. Due to limited space here, please refer to paragraphs 2-5 in our Author Rebuttal for the details.
>
> **Response to Question 1:**
>
> For simplicity, we omit some quantities (e.g., $L$ and $G$) in Theorem $1$ due to the $\mathcal{O}(\cdot)$ notation. When considering $L$ and $G$ and setting $\eta = \min \\{\frac{1}{2KL}, \frac{1}{\sigma \sqrt{TL}}, \frac{1}{{(KLG)}^{\frac{2}{3}} T^{\frac{1}{3}}}\\}$, the convergence rate in Theorem 1 can be expressed as:
>   $$\frac{1}{T}\sum\_{t=1}^T \mathbb{E} \\|\nabla F({\bf{w}}\_t) \\| ^2 \leq \mathcal{O} (\sqrt{\frac{L\sigma^2}{T}}+ (\frac{KLG}{T})^{\frac{2}{3}} + \frac{KL}{T}).$$
> We will refine the presentation of Theorem $1$ using this convergence rate. Thanks a lot for pointing out this.
>
> **Response to Question 2:**
>
> Both OrMo and SSGDm are implemented based on the Parameter Server framework. The implementation details of OrMo are outlined in Algorithm 2. For SSGDm, the only modification is to replace the asynchronous communication scheduler in line $15$ of Algorithm 2 with a synchronous communication scheduler (i.e., only when $\mathcal{C}=[K]$, the server sends the parameter ${\bf w}\_{t+1}$ and its iteration index $t+1$ to all workers in $\mathcal{C}$ and sets $\mathcal{C} = \emptyset$), as described in Remark 4.  This ensures a fair comparison between OrMo and SSGDm.
>
> We sincerely thank the reviewers for their valuable time and support of our work. Meanwhile, we would greatly appreciate it if the reviewer could re-evaluate our work in light of our response.
>
> [A] Mitliagkas et al., Asynchrony begets Momentum, with an Application to Deep Learning. arXiv:1605.09774v2, 2016.

---

> > ### Comment · Reviewer_NjVh · 2024-08-14
> >
> > I thank the authors for their rebuttals, and their thorough discussion of the points I raised. I hope the authors will include these additional experimental results, the more "standard" presentation of convergence results, as well as the discussions (especially the one about the difference with penalty function methods which I found particularly instructive) in their final version of the paper. Given that my concerns were answered, I raise my score.

---

> > > ### Author Response · Authors · 2024-08-14
> > >
> > > We greatly appreciate the insightful comments, which help us rethink our work. The additional results, the more "standard" presentation of convergence results, and the discussions will be included in the final version of the paper, as we promised. Thank you deeply for the support of our work.

---

### Author Rebuttal · Authors · 2024-08-07

We sincerely thank all the reviewers for the comments concerning our manuscript entitled "Ordered Momentum for Asynchronous SGD". These comments are valuable and very helpful. We have read through the comments carefully and responded to the comments point by point. Based on the comments of all the reviewers, we provide additional clarification on our proposed method OrMo.

The update rule in OrMo is motivated by that in SSGDm. The key insight of the update rule of the momentum and parameter in OrMo lies in tracking the sequences $\hat{\bf w}\_t$ and $\hat{\bf u}\_t$ defined in Subsection 3.3, which are updated in a manner similar to SSGDm.

Moreover, OrMo cannot be simply regarded as a penalty function method, as described in Subsection 5.2 of [A] (or a staleness control method, as described in Subsection 3.2 of [B]). The key insight of both the penalty function methods and the staleness control methods is to reduce the contribution of gradients (or parameters in federated learning algorithms) with larger delays to the parameter update. Concretely, these two methods typically assign smaller weights to the gradients (or parameters in federated learning algorithms) with larger delays when updating the parameter. OrMo differs significantly from these methods.

In OrMo, the update rule of the momentum is formulated as
$${\bf u}\_{t+1} = {\bf u}\_{t+\frac{1}{2}} + \eta \beta^{b\_{t+1} - \lceil \frac{ite(k\_t, t)}{K}\rceil}{\bf g}\_{ite(k\_t, t)}^{k\_t},$$
where the gradient ${\bf g}\_{ite(k\_t, t)}^{k\_t}$ is incorporated into momentum by multiplying it with an exponential weight $\beta^{b\_{t+1} - \lceil \frac{ite(k\_t, t)}{K}\rceil}$, as shown in line $12$ of Algorithm $2$. The exponential weight $\beta^{b\_{t+1} - \lceil \frac{ite(k\_t, t)}{K}\rceil}$ indeed decreases as the delay of the gradient increases. However, when it comes to the update rule of the parameter, the situation changes. As shown in line $13$ of Algorithm $2$, the update rule of the parameter is $${\bf w}\_{t+1} = {\bf w}\_{t+\frac{1}{2}} - \eta \frac{1-\beta^{b\_{t+1} - \lceil \frac{ite(k\_t, t)}{K}\rceil + 1}}{1-\beta}{\bf g}\_{ite(k\_t, t)}^{k\_t},$$
where the gradient ${\bf g}\_{ite(k\_t, t)}^{k\_t}$ is used to update the parameter with the weight $\frac{1-\beta^{b\_{t+1} - \lceil \frac{ite(k\_t, t)}{K}\rceil + 1}}{1-\beta}$. For a given $t$, an increase of the delay $t-ite(k\_t, t)$ implies a decrease in $ite(k\_t, t)$, typically causing the weight $\frac{1-\beta^{b\_{t+1} - \lceil \frac{ite(k\_t, t)}{K}\rceil + 1}}{1-\beta}$ to increase. This is in direct contrast to the insight behind both the penalty function methods and the staleness control methods. The design of the parameter update rule with respect to the momentum value $\beta$ is crucial for formulating the gap between $\hat{\bf w}\_t$ and ${\bf w}\_t$ as outlined in Lemma $2$ in Subsection 3.3. An ablation study was also conducted to justify the parameter update rule in line $13$ of Algorithm $2$. We replaced the update rule in line $13$ of Algorithm 2 with a vanilla SGD step, ${\bf w}\_{t+1} = {\bf w}\_{t+\frac{1}{2}} - \eta {\bf g} \_{ite(k\_t, t)}^{k\_t}$ and name it OrMo (vanilla SGD step). The comparison between OrMo and OrMo (vanilla SGD step) is presented below.

|Test Accuracy (CIFAR10)       | K=16 (hom.)        | K=64 (hom.)        | K=16 (het.)        | K=64  (het.)       |
| ---|---|-----|-----|----|
|OrMo (vanilla SGD step)       | 90.32\% $\pm$ 0.45\% | 86.08\% $\pm$ 1.33\% | 90.23\% $\pm$ 0.32\% | 86.10\% $\pm$ 1.71\% |
|OrMo                          | **90.95\% $\pm$ 0.27\%** | **88.03\% $\pm$ 0.28\%** | **91.01\% $\pm$ 0.10\%** | **87.76\% $\pm$ 0.57\%** |

The relationship between OrMo and the penalty function methods should be considered orthogonal. An adaptive learning rate, multiplied by a "penalty function" related to the delay of the gradient, can be incorporated into OrMo. For example, replace the learning rate $\eta$ in Algorithm 2 with $\eta\_t = \eta \times penalty(\tau\_t)$, where $penalty$ is the penalty function related to the gradient delay. In this way, the update rule in lines $12$ and $13$ will be
$${\bf u}\_{t+1} = {\bf u}\_{t+\frac{1}{2}} + \eta\_t \beta^{b\_{t+1} - \lceil \frac{ite(k\_t, t)}{K}\rceil}{\bf g}\_{ite(k\_t, t)}^{k\_t},$$  $${\bf w}\_{t+1} = {\bf w}\_{t+\frac{1}{2}} - \eta\_t \frac{1-\beta^{b\_{t+1} - \lceil \frac{ite(k\_t, t)}{K}\rceil + 1}}{1-\beta}{\bf g}\_{ite(k\_t, t)}^{k\_t}.$$
Actually, incorporating adaptive learning rates into asynchronous methods is an important research direction. Some works, such as [C], provide convergence analysis for adaptive learning rates, but these analyses rely on demanding assumptions, such as bounded delays. Some works, such as [D] and [E], offer rigorous theoretical guarantees but lack sufficient empirical verification. Designing adaptive learning rates for asynchronous methods that provide both strong theoretical guarantees and promising empirical results remains an open challenge. Our focus in this paper is on incorporating momentum into ASGD, which is orthogonal to the adaptive learning rate schemes. We plan to explore integrating OrMo with an adaptive learning rate in future work.

We hope that we have addressed the reviewer's concerns, and we are always willing to answer any further questions. Meanwhile, we would greatly appreciate it if the reviewer could re-evaluate our work in light of our response.

[A] Xie et al., Asynchronous Federated Optimization. OPT 2020.

[B] Wu et al., HiFlash: Communication-Efficient Hierarchical Federated Learning with Adaptive Staleness Control and Heterogeneity-aware Client-Edge Association. IEEE TPDS 2023.

[C] Barkai et al., Gap Aware Mitigation of Gradient Staleness. ICLR 2020.

[D] Koloskova et al., Sharper Convergence Guarantees for Asynchronous SGD for Distributed and Federated Learning. NeurIPS 2022.

[E] Mishchenko et al., Asynchronous SGD Beats Minibatch SGD Under Arbitrary Delays. NeurIPS 2022.

---

> ### Comment · Area_Chair_ZQwB · 2024-08-11
> **comparison to existing results**
>
> How is the theoretical convergence rate in this paper comparable to existing results such as
>
> vanilla asynchronous SGD:
> Asynchronous parallel stochastic gradient for nonconvex optimization.
>
> momentum asynchronous SGD:
> 1-bit Adam: Communication Efficient Large-Scale Training with Adam’s Convergence Speed

---

> > ### Author Response · Authors · 2024-08-12
> >
> > Dear Area Chair,
> >
> > We greatly appreciate your valuable time on our submission and the insightful comments, which make us rethink our method. We would like to briefly summarize the theoretical results in our work and then compare our results with those in existing works [1][2].
> >
> > **Theoretical results in our work:**
> > In our submission, we propose an asynchronous momentum method called OrMo and give a theoretical analysis of the convergence rate of OrMo without relying on the maximum delay of the gradients. In Theorem $1$ of our submission, OrMo has the following convergence rate:
> > $$ \frac{1}{T}\sum_{t=1}^T \mathbb{E}\\|\nabla F({\bf w}_t)\\|^2\leq \mathcal{O} (\frac{\sigma}{\sqrt{T}}+ \frac{K^{\frac{2}{3}}}{T^{\frac{2}{3}}} + \frac{K}{T}), $$
> > where $T$ is the number of iterations and $K$ is the number of workers. Below, we will make a comparison between the theoretical convergence rate of OrMo and those of the existing works in [1][2].
> >
> > **Comparison with [1]:**
> > [1] provides a theoretical analysis for vanilla asynchronous SGD (ASGD). As shown in Theorem $1$ and Corollary $2$ in [1], for smooth non-convex problems, ASGD has a convergence rate of $\frac{1}{T}\sum\_{t=1}^T \mathbb{E}\\|\nabla F({\bf w}\_t)\\|^2\leq \mathcal{O} (\frac{\sigma}{\sqrt{T}})$ when $T \geq \Omega(\tau_{max}^2)$. For OrMo, the same convergence rate can be achieved when $T \geq \Omega(K^4)$.
> >
> > Although the convergence rate are the same, our analysis is based on weaker assumptions since we do not assume a bounded delay. To the best of our knowledge, this is the first work to establish the convergence analysis of ASGD with momentum without relying on the bounded delay assumption.
> >
> > **Comparison with [2]:**
> > [2] proposes 1-bit Adam, which is a communication efficient momentum SGD algorithm pre-conditioned with Adam optimizer. As shown in Theorem $1$ and Corollary $1$ in [2], 1-bit Adam has the following convergence rate:
> > $$ \frac{1}{\tilde{T}}\sum_{t=1}^{\tilde{T}} \mathbb{E}\\|\nabla F({\bf w}_t)\\|^2\leq \mathcal{O} (\frac{\sigma}{\sqrt{K\tilde{T}}}+ \frac{1}{\tilde{T}^{\frac{2}{3}}} + \frac{1}{\tilde{T}}), $$
> > where $K$ is the number of workers and $\tilde{T}$ is the number of iteration. We notice that the server in 1-bit Adam needs to aggregate the information from all the $K$ workers at each iteration and every worker executes a gradient computation for each iteration. For fair comparison, we can rewrite the convergence rate of 1-bit Adam in terms of the number of gradient computations as $\mathcal{O} (\frac{\sigma}{\sqrt{C}}+ \frac{K^{\frac{2}{3}}}{C^{\frac{2}{3}}} + \frac{K}{C})$, where $C$ is the number of gradient computations and $C = K \tilde{T}$. At each iteration of OrMo, the server only needs one stochastic gradient. Thus, the convergence rate of OrMo in terms of the number of  gradient computations is $\mathcal{O} (\frac{\sigma}{\sqrt{C}}+ \frac{K^{\frac{2}{3}}}{C^{\frac{2}{3}}} + \frac{K}{C})$. We can find that OrMo and 1-bit Adam have the same convergence rate in terms of gradient computations.
> >
> > Meanwhile, we would like to clarify that although the two methods have the same convergence rate, the application scenarios of 1-bit Adam and OrMo are quite different. The communication compression technique in 1-bit Adam can greatly reduce the communication cost while the asynchronous updating technique in OrMo is mainly to reduce the synchronization overhead when the computing capabilities of workers are heterogeneous. Additionally, combining OrMo with the communication compression technique used 1-bit Adam can reduce its communication overhead, which can be considered in future work.
> >
> > ---
> >
> > We sincerely thank you again for the valuable comments, which greatly help us improve our work. We promise that the discussion and the references [1][2] will be added in the final version.
> >
> > [1] Lian et al., Asynchronous Parallel Stochastic Gradient for Nonconvex Optimization. NeurIPS 2015.
> >
> > [2] Tang et al., 1-bit Adam: Communication Efficient Large-Scale Training with Adam’s Convergence Speed. ICML 2021.

---

### Decision · Program_Chairs · 2024-09-25

**Decision:**

Accept (poster)

**Comment:**

Given the momentum information is a critical component in DL training, this paper proposed a novel method -- ordered momentum (OrMo) and focused on studying the convergence efficiency of its asynchronous parallelization. This paper provided a reasonable convergence efficiency proof. Considering the value of the marriage of the practical (asynchronous) parallel system and the practical (momentum based) optimization algorithm, we decide to accept this paper.

But in the meantime, this final version of this paper needs to be significantly improved and polished as suggested by reviewers, especially the following two aspects:
- The boundedness assumption is over strong, which is not essential in this proof. Please refer to [1] and [2] how to remove it.
- The comparison to existing work about asynchronous/parallel momentum SGD [1] is missing. The discussion of differentiation needs to be more clear.

[1] momentum asynchronous SGD: 1-bit Adam: Communication Efficient Large-Scale Training with Adam’s Convergence Speed
[2] DISTRIBUTED LEARNING SYSTEMS WITH FIRST-ORDER METHODS.